# Single-cell analysis of chromatin accessibility in the adult mouse brain

Songpeng Zu[1,10], Yang Eric Li[1,2,10], Kangli Wang[1,10], Ethan J. Armand[1], Sainath Mamde[1], Maria Luisa Amaral[1], Yuelai Wang[1], Andre Chu[1], Yang Xie[1], Michael Miller[3], Jie Xu[1], Zhaoning Wang[1], Kai Zhang[1], Bojing Jia[1], Xiaomeng Hou[3], Lin Lin[3], Qian Yang[3], Seoyeon Lee[1], Bin Li[1], Samantha Kuan[1], Hanqing Liu[4], Jingtian Zhou[4], Antonio Pinto-Duarte[5], Jacinta Lucero[5], Julia Osteen[5], Michael Nunn[6], Kimberly A. Smith[7], Bosiljka Tasic[7], Zizhen Yao[7], Hongkui Zeng[7], Zihan Wang[8], Jingbo Shang[8], M. Margarita Behrens[5], Joseph R. Ecker[6], Allen Wang[3], Sebastian Preissl[3,9] & Bing Ren[1,3✉]

Recent advances in single-cell technologies have led to the discovery of thousands of brain cell types; however, our understanding of the gene regulatory programs in these cell types is far from complete[1–4]. Here we report a comprehensive atlas of candidate *cis*-regulatory DNA elements (cCREs) in the adult mouse brain, generated by analysing chromatin accessibility in 2.3 million individual brain cells from 117 anatomical dissections. The atlas includes approximately 1 million cCREs and their chromatin accessibility across 1,482 distinct brain cell populations, adding over 446,000 cCREs to the most recent such annotation in the mouse genome. The mouse brain cCREs are moderately conserved in the human brain. The mouse-specific cCREs—specifically, those identified from a subset of cortical excitatory neurons—are strongly enriched for transposable elements, suggesting a potential role for transposable elements in the emergence of new regulatory programs and neuronal diversity. Finally, we infer the gene regulatory networks in over 260 subclasses of mouse brain cells and develop deep-learning models to predict the activities of gene regulatory elements in different brain cell types from the DNA sequence alone. Our results provide a resource for the analysis of cell-type-specific gene regulation programs in both mouse and human brains.

The Brain Initiative Cell Census Network aims to achieve a comprehensive understanding of the cellular and molecular composition of the mammalian brain[1]. As an experimental model, the laboratory mouse has a critical role in the investigation of gene function in vivo as well as in the development and safety evaluation of various therapeutics. A detailed catalogue of cell types in the mouse brain along with their spatial distribution and functional connections would therefore greatly facilitate the study of the complex neurocircuits and gene pathways as well as help in the development of treatments for neurological disorders. Single-cell transcriptomics studies[2–7] have identified hundreds of subclasses and thousands of cell types across the brain. This considerable cellular and spatial complexity underscores the need for a better understanding of the *cis*-regulatory elements (CREs) that are responsible for the identity and gene expression patterns in each cell type.

CREs control spatiotemporal gene expression through the binding of sequence-specific transcription factors (TFs) and the recruitment of chromatin remodeller proteins and/or transcription machinery to their target genes[8–10]. These elements, including promoters, enhancers,

insulators, silencers and other less-well-characterized regulatory sequences work together to drive cell-type-specific gene expression in development[11,12], differentiation and disease[13,14]. Comprehensive mapping of CREs in mouse brain cells will provide mechanistic insights into gene regulation and function in different brain cell types and advance our understanding of brain development and neurological disorders.

Previous catalogues of cCREs in mouse brain cells were derived through epigenomic profiling of a limited number of brain regions and are therefore incomplete[2,15–22]. To more comprehensively delineate the cCREs in the mouse brain cells, we used the single-nucleus assay for transposase-accessible chromatin followed by sequencing (snATAC–seq) to profile chromatin accessibility at the single-cell resolution across the entire adult mouse brain. In a previous study[19] that focused on the mouse cerebrum, we reported the delineation of 160 cell types comprising approximately 800,000 brain cells across 45 anatomic dissections, and the annotation of 491,818 cCREs that are probably deployed in one or more of these cell types. Here we report the analysis of an additional 1.5 million brain cells from the rest of mouse

[1]Department of Cellular and Molecular Medicine, University of California San Diego, School of Medicine, La Jolla, CA, USA. [2]Department of Neurosurgery and Genetics, Washington University School of Medicine, St Louis, MO, USA. [3]Center for Epigenomics, University of California San Diego, School of Medicine, La Jolla, CA, USA. [4]Genomic Analysis Laboratory, The Salk Institute for Biological Studies, La Jolla, CA, USA. [5]The Salk Institute for Biological Studies, La Jolla, CA, USA. [6]Howard Hughes Medical Institute, The Salk Institute for Biological Studies, La Jolla, CA, USA. [7]Allen Institute for Brain Science, Seattle, WA, USA. [8]Department of Computer Science and Engineering, University of California San Diego, La Jolla, CA, USA. [9]Institute of Experimental and Clinical Pharmacology and Toxicology, Faculty of Medicine, University of Freiburg, Freiburg, Germany. [10]These authors contributed equally: Songpeng Zu, Yang Eric Li, Kangli Wang. ✉e-mail: biren@health.ucsd.edu

brain regions, including 72 new anatomical dissections. Through integrative analysis of a total of 2.3 million mouse brain cells, we provide a comprehensive map of cCREs representing 1,482 brain cell types. Our results not only provide independent evidence to support the complexity and diversity of cell types across brain regions, but also double the annotated mouse brain cCREs to 1 million.

A large fraction of the mouse brain cCREs has sequence homology in the human genome, and displays chromatin accessibility in the human brain cells[23], suggesting conserved gene regulatory functions. Consistent with previous reports[10,24,25], mouse-specific brain cCREs, especially those found in the subclasses of excitatory neurons, are strongly enriched for transposable elements (TEs) including LINE-1 and endogenous retrotransposons, highlighting a potential role of TEs in the evolution of neuronal functions in the mammalian brain. We also predict gene regulatory networks (GRNs) in over 260 subclasses of brain cell types and develop deep-learning-based models to predict cell-type-specific use of cCREs from DNA sequence information.

## Single-cell analysis of the mouse brain

We dissected 117 brain regions from the isocortex, olfactory bulb (OLF), hippocampal formation (HPF), striatum (STR), pallidum (PAL), amygdala (AMY), thalamus (TH), hypothalamus (HY), midbrain (MB), pons (P), medulla (MY) and cerebellum (CB) in 8-week-old male mice (Fig. 1a, Extended Data Fig. 1 and Supplementary Table 1), including 45 dissections from the isocortex, OLF, HPF, STR and PAL reported previously[19]. The dissections were performed on 600-µm-thick coronal brain slices according to the Allen Brain Reference Atlas[26] (Extended Data Fig. 1) with two replicates obtained from pools of the same region dissected from at least two brains (Fig. 1a and Methods). We performed snATAC–seq for all of the 234 samples using an automated single-cell combinatorial indexing ATAC–seq[27] protocol. The sequencing reads corresponding to each nucleus were then deconvoluted on the basis of nucleus-specific DNA barcode combinations (Extended Data Fig. 2a–e). High correlations between biological replicates (median, 0.99; range, 0.96–1.0) and between datasets from similar brain regions (ranges: 0.97–0.99 (AMY); 0.94–0.98 (CB); 0.89–0.99 (HPF); 0.97–0.99 (HY); 0.93–0.99 (isocortex); 0.94–0.99 (MB); 0.98–0.99 (MY); 0.89–0.99 (OLF); 0.95–0.99 (PAL); 0.94–0.99 (P); 0.83–0.98 (STR); and 0.92–0.99 (TH)) support the high reliability and robustness of the assays (Extended Data Fig. 2f). We confirmed the high quality of all of the datasets ($n = 234$: 117 dissections with 2 replicates) using a set of quality-control metrics (Methods and Extended Data Fig. 2a–f). For the subsequent analyses, we focused on the nuclei with at least 1,000 sequenced fragments and the transcriptional start site (TSS) enrichment above 10 (Extended Data Fig. 3a). We next removed potential doublets in each dataset based on a modified Scrublet[28] procedure using SnapATAC2[29]. As Scrublet was originally designed for single-cell RNA-sequencing (scRNA-seq) doublet removal, we compared it using another method, AMULET[30], which was recently published for doublet detection and removal in snATAC–seq data. We found that it achieved similar results for our data based on a simulation study, in which the doublets were simulated from several samples from our data (Extended Data Fig. 3b). After removing 7% of nuclei that were deemed to be potential doublets (Extended Data Fig. 3c,d), we retained the chromatin accessibility profiles from 2,355,842 nuclei, with a median 4,368 DNA fragments per nucleus (Supplementary Table 2). Among them, 817,655 were from the isocortex (including 370,841 from previous study), 201,113 from the OLF (including 137,209 from previous study), 155,952 from the STR (including 114,743 from previous study), 81,834 from the PAL (including 38,960 from previous study), 271,933 from the HPF (including 164,568 from previous study), 65,958 from the AMY, 142,890 from the TH, 83,321 from the HY, 243,137 from the MB, 82,488 from the MY, 103,147 from the pons and 106,414 from the CB (Fig. 1a,b and Extended Data Fig. 3e,f). This dataset represents a

considerable number of single-cell chromatin accessibility profiles for the mammalian brain.

## Clustering and cell type annotation

We performed iterative clustering using SnapATAC2[29] to classify the 2.3 million nuclei into distinct cell groups on the basis of their pairwise similarity of chromatin accessibility profiles (Methods, Extended Data Figs. 4 and 5 and Supplementary Table 3). Before clustering, we first visualized the data using uniform manifold approximation and projection (UMAP; Fig. 1c) with a 5 kb resolution for genomic bin features in SnapATAC[31] for a global view. In the UMAP, we marked the nuclei into three major divisions, including 998,000 nuclei predominantly comprising glutamatergic (Glut) neurons (based on the neurotransmitter genes *Slc17a7*, *Slc17a6*, *Slc17a8*); 384,000 nuclei predominantly comprising GABAergic neurons (GABA, based on the neurotransmitter gene *Slc32a1*) and 959,000 nuclei consisting of primarily non-neuronal cell types. We performed four rounds of iterative clustering to further classify the cells into subclasses and cell subtypes (Extended Data Fig. 4a). During clustering, we used a 500 bp resolution for genomic bin features. After the first iteration (hereafter, L1-level clustering), we divided the 2.3 million nuclei into 37 groups for L2-level clustering, using over 4 million chromatin features. For each group, we then performed a second and a third round of clustering (L2-level and L3-level clustering) sequentially with the top 500,000 genomic bin features and identified a total of 248 subgroups and 899 subtypes of brain cells, respectively (Extended Data Fig. 4a). A total of 291 out of 899 L3-level subtypes consisted of more than 400 cells per subtype and, in total, they captured 1.8 million cells. For these 291 L3-level subtypes, we also performed a fourth round of clustering (L4-level clustering) to further classify them into a total of 874 clusters. In summary, we identified a total of 1,482 cell clusters (874 L4-level clusters and 608 L3-level clusters without L4-level clustering). The number of nuclei in each cluster ranges from 34 to 48,694, with a median number of 484 nuclei per cluster (Supplementary Tables 3 and 4). We used the term subtypes to represent the 1,482 clusters in the latter part of this Article.

To annotate the cell type identity of the 1,482 subtypes, we performed integration analysis using the data reported in a companion single-cell RNA-seq study of 2 million cells (over 5,300 clusters) from adult male mouse brains[5]. We first calculated the gene expression scores in each nucleus using SnapATAC2 with the fragments mapped to the gene promoter (up to 2 kb to TSSs) and gene body regions as described previously[31,32]. We next performed integration analysis using the Seurat[32,33] separately for neuronal cells and non-neuronal cells (Methods). The co-embedding of both the scRNA-seq and the snATAC–seq neuronal cells showed excellent overlap between the two modalities (Fig. 1d) and the mouse brain major regions (Extended Data Fig. 6a,b). We also observed the same result for non-neuronal cells (Extended Data Fig. 6c–e). The consensus matrix calculated on the basis of the ratio of transferred labels from the scRNA-seq data to our snATAC–seq data showed excellent correspondence between the two datasets, suggesting the robustness of the cell type identification based on either transcriptome or chromatin accessibility (Fig. 1e, Extended Data Fig. 6f–h and Supplementary Table 5). For each snATAC–seq-based subtype, we used the top-ranked cluster label transferred from the scRNA-seq data to represent its scRNA-seq cluster-level annotation. In total, 1,267 neuronal subtypes in the snATAC–seq data were mapped to 965 scRNA-seq clusters. In the scRNA-seq data, the 5,300 clusters were grouped into 338 cell subclasses, the most representative layer for cell type analysis. To annotate our data more robustly, we next mapped our cell subtypes into this layer using the hierarchical relationship between cell cluster and cell subclass defined in the scRNA-seq data. The heat map of the consensus matrix between our subtypes and the scRNA-seq subclasses showed excellent correspondence (Fig. 1e and Supplementary Table 5). To reduce the potential annotation bias induced by different numbers

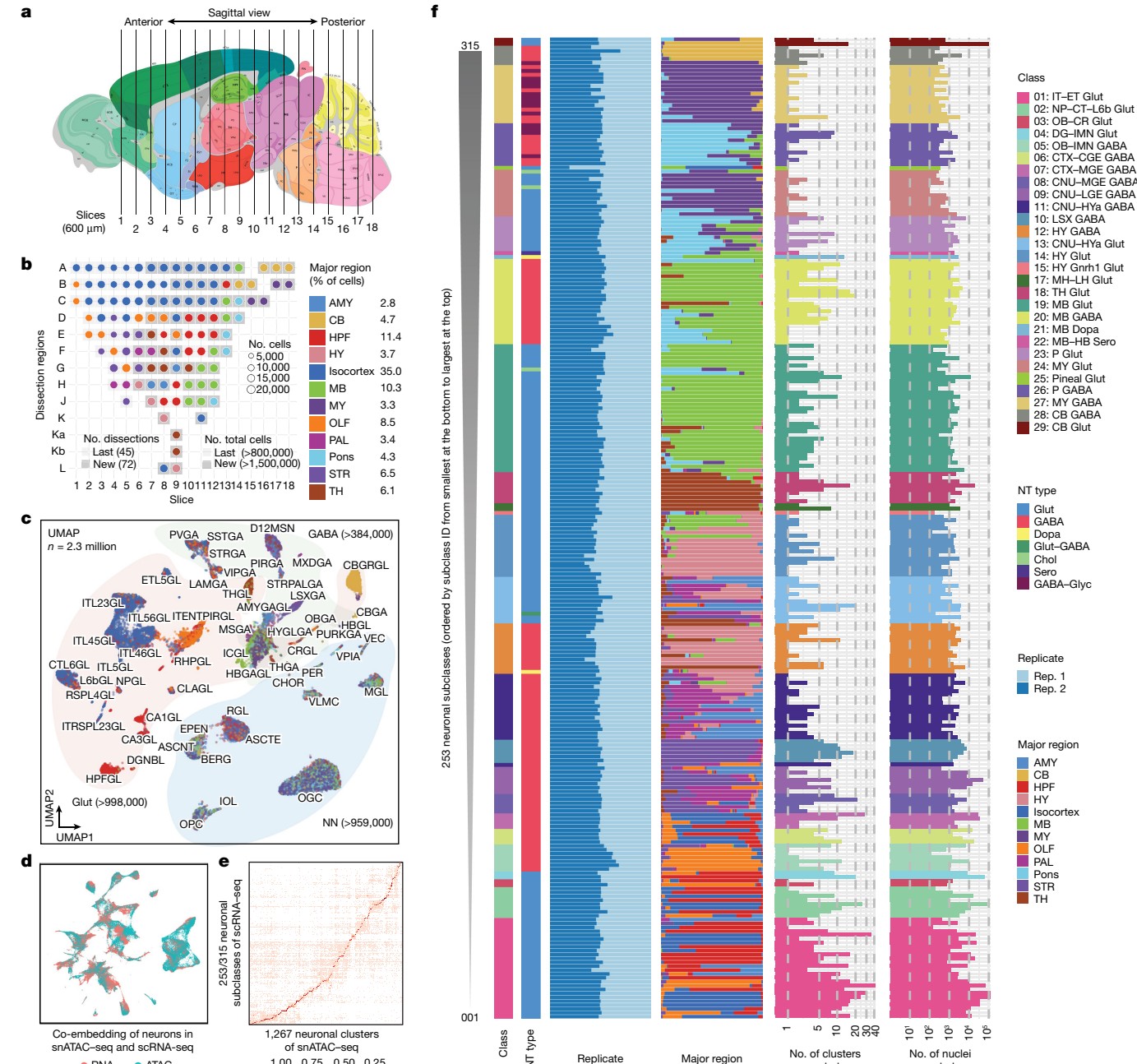

**Fig. 1 | Single-cell analysis of chromatin accessibility in the adult whole mouse brain. a**, Schematic of the sample dissection strategy. The brain map was generated using coordinates from the Allen Mouse Brain Common Coordinate Framework (CCF) v.3 (ref. 26). **b**, The number of nuclei for 117 dissections after quality control and doublet removal. The dot size is proportional to the size of cells and the dissections that were not covered by our previous study[19] are shown in grey. A to L on the left were used as the dissection region labels on each slice (details are provided in Extended Data Fig. 1). The number of dissections represents the number of dissections covered by our previous study (last) and updated in the current study (new). The total number of cells represents the number of cells covered by our previous study (last) and updated in the current study (new). **c**, UMAP[81] embedding and clustering analysis of snATAC–seq data. The light colours denote major cell classes. NN, non-neuronal cells. Cells are coloured on the basis of major regions as in **b**. **d**, The co-embedding UMAP embedding of the neuronal cells from scRNA-seq data[5] and the snATAC–seq data on the same space coloured by the two modalities. **e**, The consensus score between neuronal subclasses from the scRNA-seq data above and L4-level

neuronal clusters from our snATAC–seq data. **f**, The 253 neuronal subclasses in our snATAC–seq data matched to neuronal subclasses in the scRNA-seq above, and ordered on the basis of the subclass IDs (for all of the following figures, the order was kept the same unless otherwise mentioned). From left to right, the bar plots represent the class, major neurotransmitter (NT) type, biological replicate distribution of nuclei, major region distribution of nuclei, number of clusters and number of nuclei. Detailed information about class, neurotransmitter type and subclass is reported in the companion paper[5]. A list of full names of the subclasses is provided in Supplementary Table 3. CTX, cerebral cortex; HYa, anterior hypothalamus; L6b, layer 6b; LSX, lateral septal complex; IT, intratelencephalic; ET, extratelencephalic; NP, near-projecting; CT, corticothalamic; OB, olfactory bulb; CR, Cajal-Retzius; DG, dentate gyrus; IMN, immature neurons; CGE, caudal ganglionic eminence; MGE, medial ganglionic eminence; CNU, cerebral nuclei; LGE, lateral ganglioniceminence; MH, medial habenula; LH, lateral habenula; Chol, cholinergic neurons; Dopa, dopaminergic neurons; Glyc, glycinergic neurons; Sero, serotonergic neurons.

of cells in the clusters, for each of our 1,482 subtypes, we manually checked the major regions of the top three cluster-related subclasses, and the gene markers for some subclasses using the bigwig data and gene expression scores (Extended Data Fig. 7) generated using Sna-pATAC2. Finally, 275 out of 338 subclasses were annotated to the 1,482 subtypes. This includes 253 out of 315 neuronal subclasses, covering 28 neuronal classes and 7 neurotransmitter types, as well as 22 out of 23 non-neuronal subclasses, covering 5 non-neuronal classes (Supplementary Table 4). We confirmed that the matched subclasses in our snATAC–seq data were robust to variations in the sequencing depth, signal-to-noise ratio between brain regions and replicates (Extended Data Fig. 4b,c) by performing the $k$-nearest-neighbour batch effect test[34] and local inverse Simpson's index analysis[35] (Extended Data Fig. 4d,e) and by comparing the ratio of biological replicates across multiple subclasses (Extended Data Fig. 5). The unmatched 63 subclasses correspond mainly to rare cell populations, accounting for a total of 1.7% of the scRNA-seq data. For example, the only unmatched non-neuronal subclass is monocytes, with 21 cells. Other unmatched subclasses correspond to rare cell subclasses mainly from the MB, pons and MY regions, in which the subtle differences between cell types may hinder their identification using chromatin accessibility profiles alone[5]. Nevertheless, the general agreement between the open-chromatin-based clustering and transcriptomics-based clustering laid the foundation for integrative analysis of cell-type-specific gene regulatory programs in the mouse brain, as for the mouse cerebral region[19]. In the text below, we focus on the snATAC–seq subclasses and the subtypes within each subclass based on the above integrative analysis.

Most neuronal cell types and some non-neuronal cell types showed strong regional specificity (Fig. 1f and Extended Data Fig. 8). For example, in the CB region, we identified 15 subtypes consisting of 97,000 nuclei that were annotated as CB granule Glut neurons; and two Bergmann glial subtypes including about 1,600 nuclei. In the HY region, one subtype with 297 nuclei specifically showed the imputed gene expression of the neuropeptide gene *Pmch*, which integrated well with the lateral hypothalamic area *Pmch*-positive Glut neurons from the scRNA-seq data. A series of astrocyte-related cells were identified with region specificity, such as astrocytes in the telencephalon region, astrocytes in non-telencephalon regions, choroid plexus cells and tanycytes, which were integrated well with the corresponding subclasses in the scRNA-seq data (Extended Data Fig. 6i).

### Identification and annotation of cCREs

To identify the cCREs in each of the 1,482 subtypes, we aggregated the DNA-sequence reads from cells in the subtype and determined peaks of open chromatin signals using MACS2[36] (Extended Data Fig. 9a). When the number of cells of a subtype was fewer than 200, we combined it with other subtypes that were within the same L3-level subtype and mapped to the same cluster in the scRNA-seq data. Only 19 subtypes were affected by this step. Finally, we performed the peak calling on the resulting 1,463 clusters. We selected the genomic regions mapped as accessible chromatin in both biological replicates. To account for potential biases introduced by factors such as sequencing depth and/or number of nuclei in individual clusters, we retained only the reproducible peaks based on a modified MACS2 score (hereafter, score per million (SPM))[37] (Methods and Extended Data Fig. 9a). The peaks with SPM ≥ 5 were retained. For each subtype, we retained the peaks that were determined to be open chromatin regions in a significant fraction of the cells (false-discovery rate (FDR) < 0.01, zero-inflated $\beta$-model; Extended Data Fig. 9b). In total, we identified a union of 1,053,811 open chromatin regions (500 bp extension surrounding the peak summit) or cCREs (Supplementary Table 6), which together make up 19% of the mouse genome (Supplementary Tables 7 and 8). This list includes 98% of the cCREs reported in our previous study on the

mouse cerebral regions[19] (Extended Data Fig. 9c), and further expands it by an additional 446,606 cCREs. They are also enriched for active chromatin states or potential insulator-protein-binding sites mapped in bulk mouse brain tissues (Extended Data Fig. 9d). Nearly all of the frequently interacting regions previously identified from the mouse cortex region[38] (3,158 out of 3,169) overlap with our cCREs (Methods and Extended Data Fig. 9e,f). Only 2.3% were in promoter regions (defined as 1.5 kb upstream and 500 bp downstream of the TSS) of protein-coding and long non-coding RNA genes, while 34.2% were in intron regions, 35.9% in intergenic regions and 22.8% in TEs, including long terminal repeats (LTRs), long interspersed nuclear elements (LINEs), short interspersed nuclear element (SINEs) and other repeats (Fig. 2a). We found an average of 45,303 (range, between 4,947 and 177,906) peaks (501 bp in length) in each cell cluster (Extended Data Fig. 9g).

The list of cCREs greatly expands the previous catalogue of mouse cCREs defined by bulk chromatin accessibility data. Importantly, 44% of the mouse brain cCREs (Supplementary Table 9) did not overlap with the DNase-hypersensitive sites (DHSs) mapped in a broad spectrum of mouse tissues (not limited to brain) and multiple developmental stages[39,40] (Fig. 2b). Several lines of evidence indicate that these cCREs probably participate in regulatory functions. First, they display higher levels of sequence conservation compared with random genomic regions with similar GC content (Fig. 2c). Second, they feature cell-type-restricted accessibility, a potential factor in their lack of detection in previous bulk tissue assays. More than 62% of the cCREs are active in less than ten subtypes, and more than 19% of them are accessible in only one cell subtype (Fig. 2d,e and Extended Data Fig. 9h). Third, the cell-type-specific chromatin accessibility profiles of these cCREs strongly correlate with DNA hypomethylation[41] (Fig. 2f, Methods and Extended Data Fig. 9i). The cCREs were organized on the basis of the non-negative matrix factorization (NMF)[42] using the matrix of normalized chromatin accessibility of the cCREs (all of the cCREs and the cCREs with no overlaps with the DHSs separately) across the 275 cell subclasses (Methods and Supplementary Tables 10 and 11). Notably, two subclasses show DNA hypomethylation across most of the cCREs (Extended Data Fig. 9j).

### Inferring GRNs

To further dissect the gene regulatory programs in each of the 275 subclasses on the basis of the subtype-specific cCREs identified previously, we first assessed the relationship between the chromatin accessibility at the cCREs with transcription levels of putative target genes across the cell subclasses, and we then constructed cell-subclass-specific GRNs[43]. We performed the analysis at the subclass level because cell clusters are sufficiently resolved and the open-chromatin landscapes align strongly with scRNA-seq dataset.

We began with detecting pairs of co-accessible cCREs within 500 kb for each cell subclass using Cicero[44] and inferred candidate target promoters for each distal cCRE located more than 1 kb away from the annotated TSSs in the mouse genome (Fig. 3a and Methods). We determined hundreds of thousands of cCRE–cCRE pairs within 500 kb of each other in 274 out of 275 cell subclasses (Supplementary Table 12). This set included the promoter-distal cCRE combinations between 502,704 distal cCREs and 24,414 promoters of protein-coding and long non-coding RNA genes (Extended Data Fig. 10a,b). The median distance between all of the promoter-distal cCREs pairs is 156 kb (Extended Data Fig. 10c).

To link potential enhancers to their putative target genes, we looked for the subsets of distal cCREs showing positive correlations between their chromatin accessibility and RNA expression of the putative target genes across the 275 cell subclasses. We computed Pearson correlation coefficients (PCCs) between the normalized chromatin accessibility signals and the RNA expression for each pair of distal cCRE and the corresponding genes of the proximal cCRE (Fig. 3a). As a control, we randomly shuffled the cCREs and the putative target genes, then

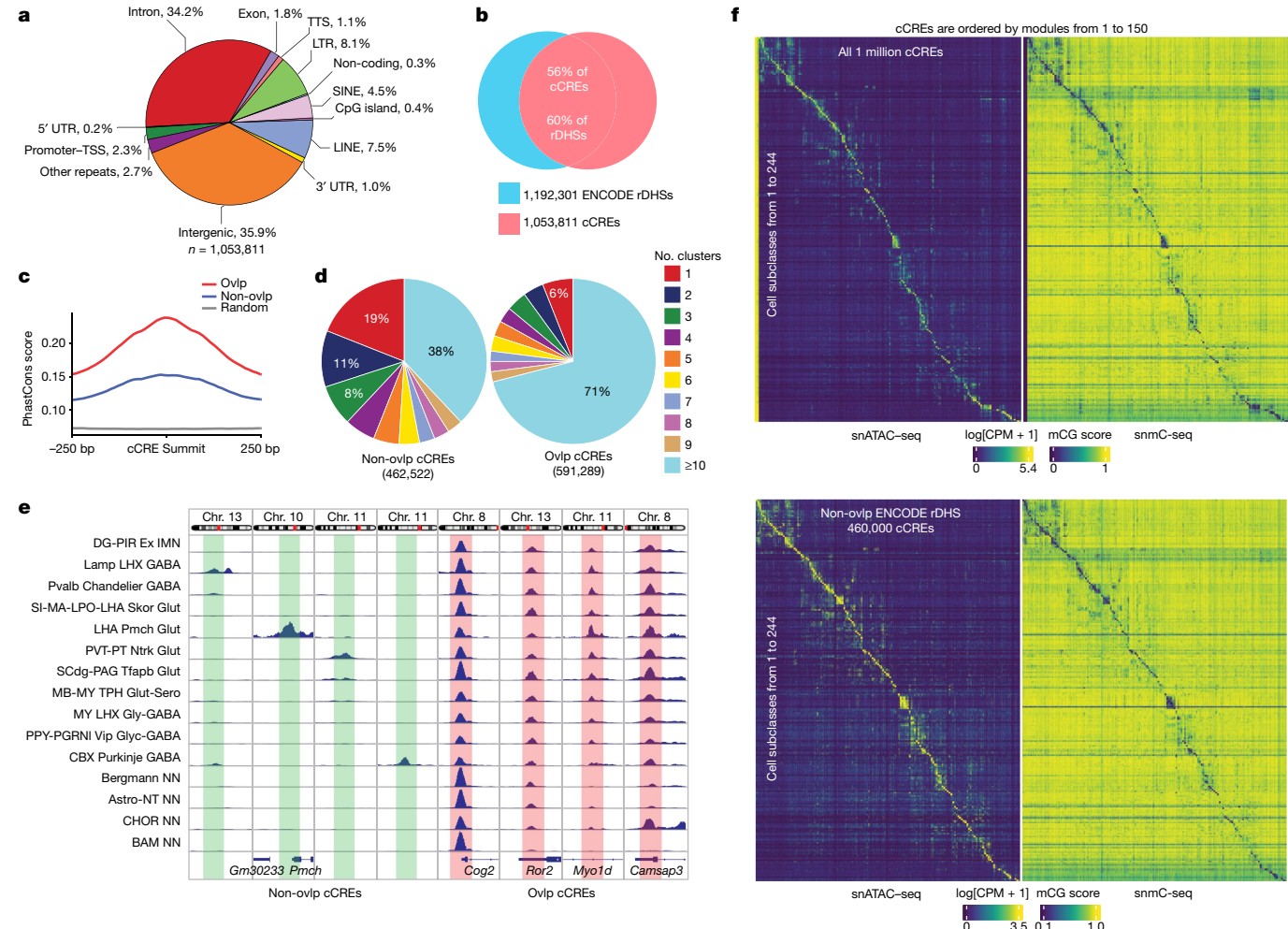

**Fig. 2 | Identification and characterization of cCREs across mouse brain cell types. a**, The fraction of cCREs that overlaps with annotated sequences in the mouse genome was determined using HOMER[45]. TTS, transcription termination site; UTR, untranslated region. **b**, The overlaps between the cCREs in this study (red) and the representative DHSs (rDHSs; blue) from the SCREEN database[18]. **c**, The average PhastCons conservation scores of cCREs (red) overlapping (ovlp) with rDHSs, cCREs (blue) with no overlaps with rDHSs, and random genomic background (grey) were determined using deepTools[82]. **d**, The fraction of cCREs captured by different cell subtypes for peak calling. Left, the cCREs with no overlaps with rDHSs. Right, the cCREs with overlaps with rDHSs. **e**, Genome browser tracks of the two types of cCREs. Left, cCREs with no overlaps with rDHSs. Right, the cCREs with overlaps with rDHSs. The subclass names were the same as for the scRNA-seq data in the companion paper[5]. **f**, The chromatin accessibility at 150 *cis*-regulatory modules across the 244 shared cell subclasses in the snATAC–seq data for all of the 1 million cCREs (top left). Rows represent subclasses, and columns are representative cCREs sampled from each module. Right, heat map showing the snDNA-methylation signals from the snmC-seq[41] analysis at the genomic locations of the corresponding cCREs for the same subclasses. Bottom, heat maps similar to those above but for only the 460,000 cCREs with no overlaps with the ENCODE rDHSs.

computed the PCCs of the shuffled cCRE–gene pairs (Fig. 3b and Methods). This analysis revealed a total of 613,485 positively correlated distal cCRE (putative enhancer)–gene pairs and 107,413 negatively correlated distal cCRE–gene pairs at an empirically defined significance threshold of FDR < 0.01 (Extended Data Fig. 10d and Supplementary Table 13). The median distance between the potential enhancers and the target promoters was 133 kb (Extended Data Fig. 10e). Each promoter region was assigned to a median of 24 putative enhancers (Extended Data Fig. 10f). The top proximal–distal cCRE pairs and positive pairs showed enrichment signals using the chromatin conformation data from the companion study[41] (Methods and Extended Data Fig. 10g,h). For the subsequent analysis, we focused mainly on the positively correlated pairs, including 281,200 potential enhancers and 20,703 putative target genes. To investigate how the putative enhancer may regulate cell-type-specific gene expression, we further classified them into 54 modules using the NMF[42] on the matrix of normalized chromatin accessibility across the cell subclasses based on the integration analysis with the scRNA-seq data, and organized the distal cCREs

based on the modules (Fig. 3c and Supplementary Tables 14 and 15). The putative enhancers in each module showed cell-subclass-specific chromatin accessibility profiles co-occurring with the RNA expression of their putative target genes (Fig. 3c). We next performed the motif-enrichment analysis for each module using HOMER[45] with a threshold of $P < 10^{-10}$ (Fig. 3c and Supplementary Table 16). The known motifs showed a similar cell-subclass-specific pattern, which indicated cell-subclass-specific regulatory programs. For example, EBF transcription factor 1 (EBF1), which is important for B cell development, was expressed in the pericytes from human brain tissues[46]. We found that EBF1 motifs are enriched in the cCREs from pericytes in the mouse brain (Fig. 3c). For example, motifs for both the TF PU.1 and interferon regulatory factor 8 (IRF8) were enriched in border-associated macrophages (BAMs) and microglia (Fig. 3c and Supplementary Tables 15 and 16). IRF8 is critical to transform microglia into a reactive phenotype[47,48]. PU.1 is especially expressed in microglia and can regulate genes associated with Alzheimer's disease in primary human microglia[49]. PU.1 and IRF8 also have essential roles in macrophages[50,51].

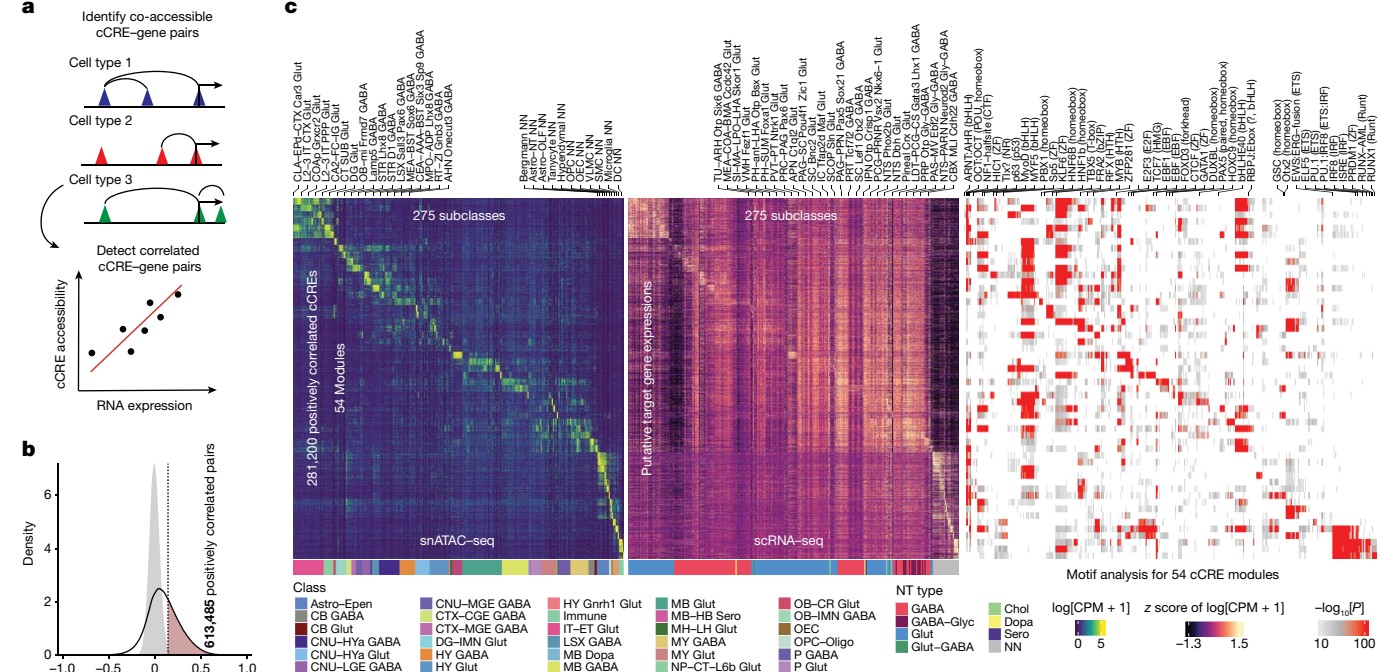

**Fig. 3 | Integrative analysis to identify the potential enhancer–gene connections across the whole mouse brain. a**, Schematic of the computational strategy used to identify cCREs that are positively correlated with the mRNA expression of the target genes; PCCs were calculated across 275 cell subclasses between the snATAC–seq and scRNA-seq data. Co-accessible *cis*-regulatory DNA interactions were predicted using Cicero[44] for each cell subclass. **b**, In total, 613,485 pairs (red) of positively correlated cCRE–gene pairs were identified (FDR < 0.01). The grey-filled curve shows the distribution of PCCs for randomly shuffled cCRE–gene pairs. **c**, The chromatin accessibility of putative enhancers (left); mRNA expression of the linked genes in the 275 cell subclasses across the whole mouse brain (middle); and the enrichment of known TF motifs in distinct enhancer gene modules (right). A total of 428 out of 440 known motifs from HOMER[45] with enrichment *P* < 10⁻¹⁰ is shown. The unadjusted *P* values were calculated using two-sided Fisher's exact tests.

We next applied CellOracle[52] to the snATAC–seq and scRNA-seq data (Methods and Extended Data Fig. 11a,b) for GRN analysis. To achieve this, the subclass-specific distal cCREs detected using Cicero above were first mapped to mouse TFs based on TF-binding motifs using the tool gimmemotifs[53]. A regularized linear regression model was then used to predict the gene expression at the single-cell level on the basis of the mapped TF-motif instances surrounding each gene promoter and generate GRNs for each subclass. The 3,000 most variable genes across all of the subclasses from the scRNA-seq data using Seurat and 499 TFs reported to have essential roles in defining cell subclasses in the scRNA-seq data[5] were included for this analysis. Finally, we successfully inferred GRNs for 267 out of 275 cell subclasses (one example of GRN from the subclass ASC-TE_NN, that is, astrocytes from the telencephalon region, is shown in Fig. 4a). The resulting GRNs contained a total of 403 TFs and 2,628 non-TF genes (Methods and Supplementary Table 17). As expected, the connectivity of the nodes follows a power-law distribution[54] (Fig. 4b) in 266 of 267 of them (Extended Data Fig. 11c). On average, each GRN owned 312 TFs and 681 genes (Fig. 4c).

Recurring network motifs are a common feature of GRNs[55]. We compared the 17 common network motifs[56] in each of the above GRNs (Methods and Supplementary Table 18) across different cell classes defined in the scRNA-seq data (Extended Data Figs. 11d,e and 12a) and across different brain regions (Methods, Extended Data Fig. 12b and Supplementary Table 19). We first mapped the 267 subclasses to five main regions, that is, the telencephalon (isocortex, OLF, AMY, STR, PAL), diencephalon (TH, HY), hindbrain (pons, MY), MB and CB, only if at least 60% (248 subclasses left) of the cells in the subclass could be mapped to these regions, and identified regulated double-positive motifs (TF A increases the expression of both TF B and TF C, and TF B and TF C can positively regulate each other) (Fig. 4d and Supplementary Table 20). The GRN from BAMs (BAM_NN; Fig. 4e) includes a regulated double-positive motif composed of activating transcription factor 3 (ATF3), KLF4 and TAL1, indicating that the three factors may positively regulate each other in the BAM subclass. ATF3 is an inflammatory mediator and a key regulator of interferon response in macrophages[57]. KLF4 from the Kruppel-like family of factors has an essential role in monocyte differentiation[58], and is a mediator of proinflammatory signals in macrophages[59]. The *Tal1* gene, which encodes a basic helix-loop-helix TF, is expressed during monocyte–macrophage lineage differentiation and has an important role in cell cycle progression and proliferation during monocytopoiesis[60,61]. Using the Cistrome Data Browser[62] as a resource for chromatin immunoprecipitation followed by sequencing data, we noticed that ATF3 binds to putative enhancers near both *Tal1* and *Klf4* in bone-marrow-derived macrophages (Gene Expression Omnibus: GSE99895; Extended Data Fig. 12c,d). Overall, non-neuronal cells showed higher numbers on several network motifs (such as the regulated double-positive motif) compared with Glut neurons and GABAergic neurons (Fig. 4e and Extended Data Figs. 11d and 12a).

Furthermore, we highlighted the importance of key TFs within these networks by calculating their eigenvector centrality scores using CellOracle. In Fig. 4f, the 267 subclasses and 226 TFs were ordered in the same manner as described in the companion paper[5] (Supplementary Table 21). Notably, we observed a similar pattern of importance scores for the TFs as seen in the scRNA-seq data, where normalized gene expression was shown. This consistency of the TF signatures across modalities reinforced the fidelity of our GRN inferences. It also demonstrated how regulatory codes of TFs across the whole mouse brain could be revealed through integrated analysis of snATAC–seq and scRNA-seq data.

TFs such as JUN, JUNB and FOS have high importance scores across multiple neuronal and non-neuronal subclasses. TFs of the bHLH family such as NEUROD1, NEUROD2, NEUROD6 and BHLHE22 have

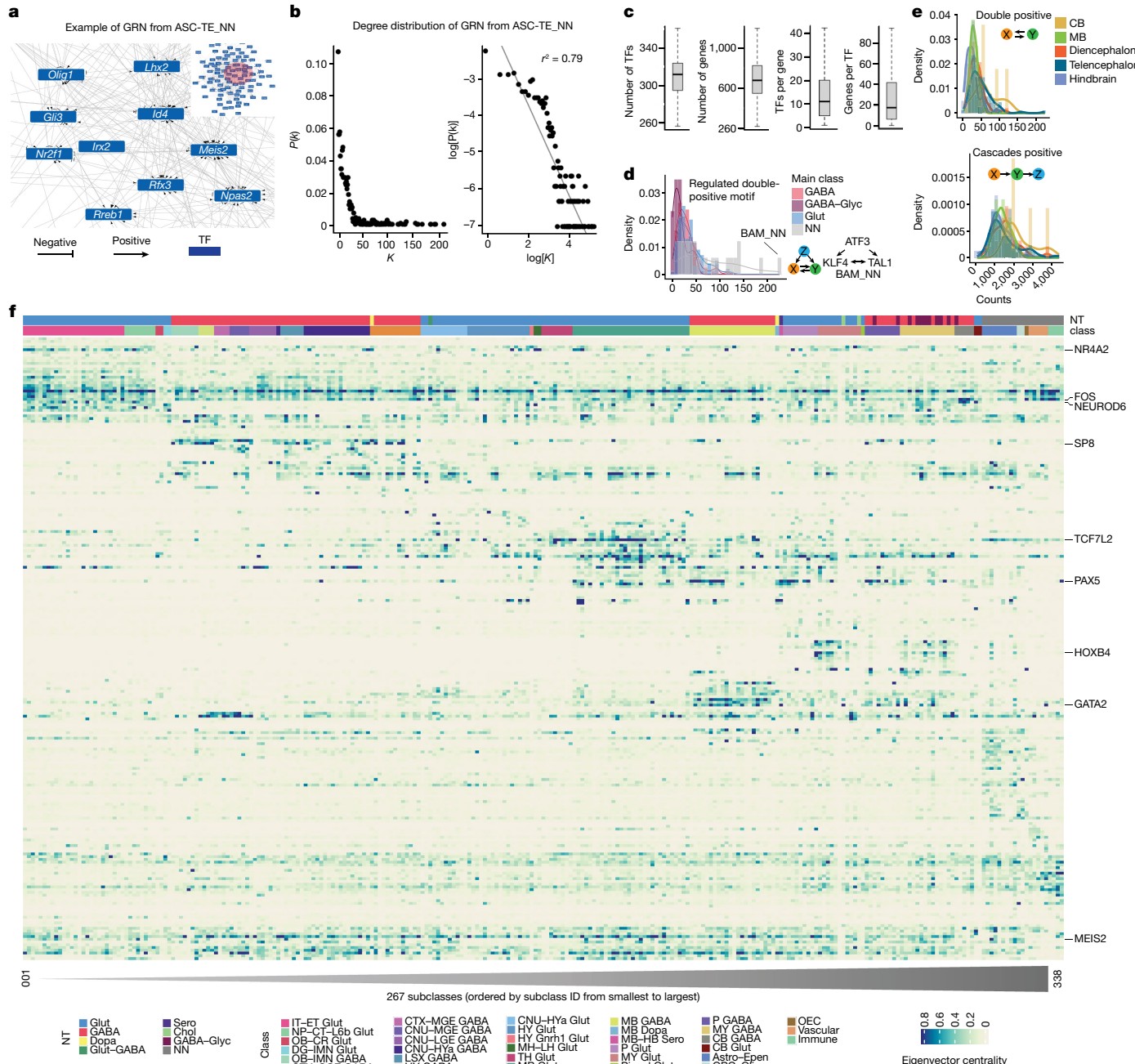

**Fig. 4 | Inference of subclass-specific GRNs across the whole mouse brain.** **a**, Example of the GRN inferred in telencephalon-region astrocyte (ASC-TE_NN) using CellOracle[52]. Edges are weighted and directed to reflect the putative regulation strength and mode (inhibition or activation). **b**, The degree distribution of the GRN in **a**. $P(k)$, the probability of a node having $k$ degree in the GRN. The degree of one node is the number of other nodes with links to it. **c**, The number of TFs, the number of genes, the number of regulated TFs per gene and the number of genes regulated by the TFs among the GRNs for each of 267 cell subclasses. The numbers of dots in each box plot from left to right are as follows: 267, 267, 185,000 and 82,000. For the latter two plots, treat TFs and genes from different subclasses as different ones. For the box plots in **c**, the box limits span the first to third quartiles, the centre line denotes the median and the whiskers show 1.5× the interquartile range. **d**, Normalized histograms of the number of the regulated double-positive[56] network motifs for each main cell class. The lines are the kernel-based density curves fitted for different histograms. **e**, Histograms of the two network motifs for five mouse brain regions: telencephalon (isocortex, OLF, HPF, STR, PAL and AMY), diencephalon (TH and HY), MB, hindbrain (MY and pons) and CB. **f**, Heat map of eigenvector-based centralities or importance scores of TFs in each of the subclass-specific GRNs. Each row represents a TF, and each column a subclass. The orders of the TFs and subclasses are based on the companion paper[5] for the similar heat map but using the scRNA-seq data. The names of the rows and columns are listed in Supplementary Table 18.

high importance scores for many types of neurons such as the Glut neurons in the isocortex region. Our analysis also indicated potential regulation of gene expression in GABAergic neurons by TFs such as ARX, SP8 and SP9 in the telencephalon regions, whereas TFs such as GATA2, TAL1 and GATA3 showed high importance scores for GABAergic

neurons in the MB and pons regions. TCF7L2, SHOX2 and EBF1 had high importance scores associated with Glut neurons specifically in the TH region. Moreover, TCF7L2 exhibited high importance in the MB region. Next, we observed that the TFs FOXA1 and FOXA2 had a specific association with the Glut neurons in the MB region. HOX-family TFs

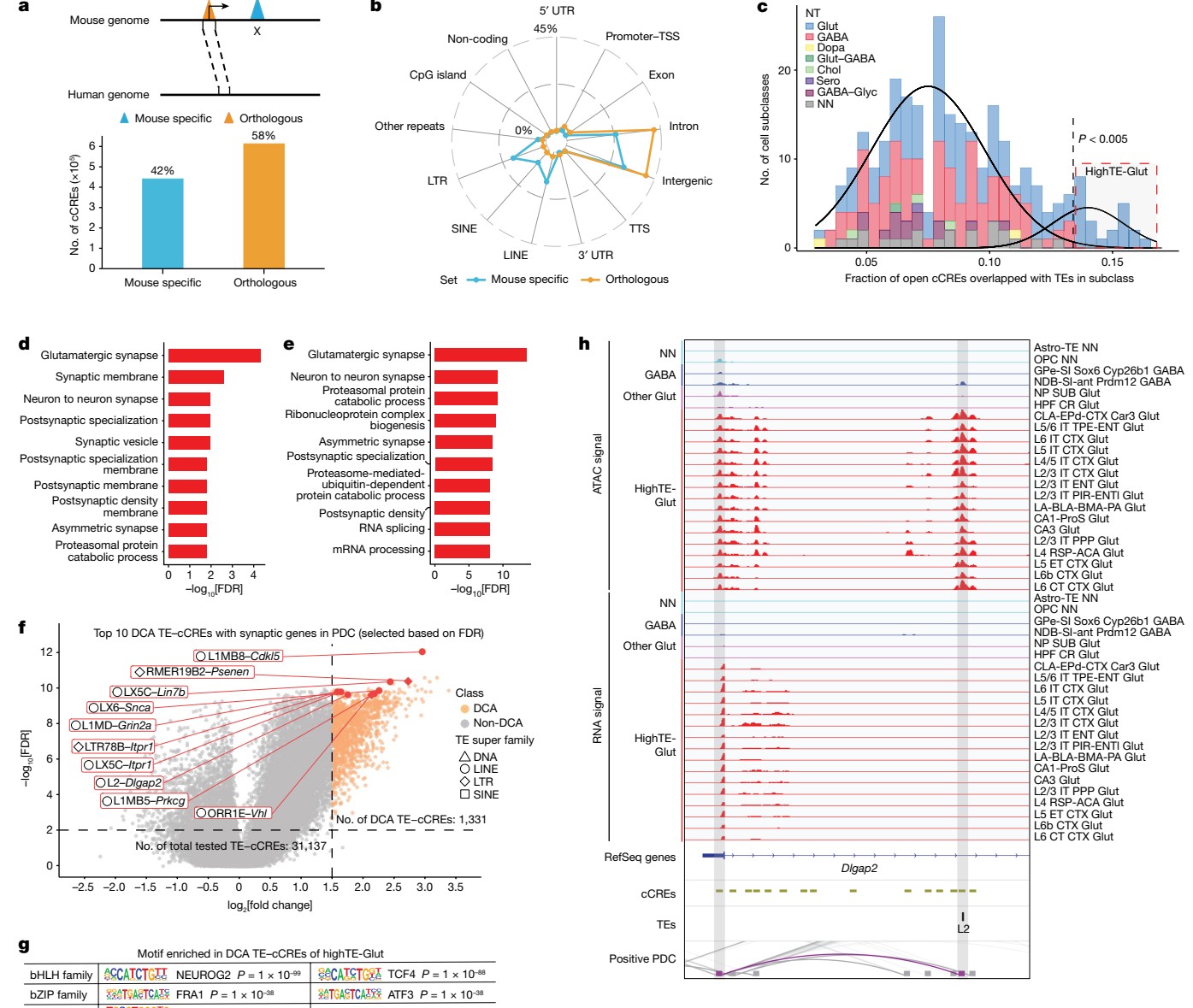

**Fig. 5 | Analyses of chromatin accessibility at TEs of cCREs. a**, Schematic of mouse-specific and orthologous cCREs. The bar plot shows the numbers of mouse-specific and orthologous cCREs. **b**, The fraction of the genomic distribution of mouse-specific and orthologous cCREs. **c**, The fraction of cCREs overlapping with TEs in each subclass of Glut neurons, GABAergic neurons, dopaminergic neurons, cholinergic neurons, serotonergic neurons, glycinergic neurons and non-neurons. The two curves show the Gaussian distribution from the mixture model. highTE-Glut refers to the Glut neuron subclasses with a high percentage of their cCREs overlapping with TEs. **d**, Gene Ontology (GO) analysis revealing an enrichment of neuronal-specific functions among genes that exhibited positive correlations with TE-cCREs (TE-related cCREs) in highTE-Glut subclasses, compared with genes positively correlated with TE-cCREs in all subclasses. **e**, GO analysis revealing an enrichment of neuronal-specific functions among genes that exhibited positive correlations with TE-cCREs in highTE-Glut subclasses, compared with genes positively correlated with all cCREs in highTE-Glut subclasses. **f**, DCA at TE-cCREs in highTE-Glut subclasses compared with other subclasses. The top ten DCA TE-cCREs correlating with synaptic-related genes are shown. The top ten DCA TE-cCRE–gene pairs (such as L1MB8–*Cdkl5*) are indicated by red boxes. The super family of the top ten DCA TE-cCREs are indicated by different shapes. **g**, The top three motif families enriched in the DCA TE-cCREs in highTE-Glut neurons. The unadjusted *P* values were calculated using two-sided Fisher's exact tests. **h**, Genome browser tracks of aggregate chromatin accessibility profiles for NN, GABA, highTE-Glut and other Glut subclasses at selected DCA TE-cCREs and gene pairs. RNA signals shown here were collected from the previous study[2]. PDC, proximal–distal connections.

displayed high importance scores in both GABAergic and Glut neurons in the MY region. Last, MAF and MAFB showed high importance scores in GABAergic neurons in the cortex region.

## Conservation of the mouse brain cCREs

To investigate the conservation of the gene regulatory landscapes in mouse brain cells, we compared the mouse brain cCREs defined in

this study with a separate study of single-cell chromatin accessibility in 42 human brain regions[23]. We first identified orthologues of mouse cCREs in the human genome by performing reciprocal homology searches and found 613,073 cCREs (58% of total mouse cCREs) defined in mouse brains to have orthologous sequences in the human genome (more than 50% of bases lifted over to the mouse genomes) (Fig. 5a and Extended Data Fig. 13a). The percentage of orthologous cCREs is significantly higher than the random expectation (32% orthologous

for randomly shuffled cCREs). Among these orthologous cCREs, 39% (22% of total mouse cCREs) were identified as open chromatin regions in one or more cell types in the human brains (Extended Data Fig. 13a,b). We therefore defined the 22% of mouse cCREs with both DNA sequence similarity and open chromatin in the human brain cells as chromatin-accessibility-conserved cCREs. This modest rate of conservation may reflect the still incomplete annotation of cCREs in the human brain. Indeed, nearly 33% of the human brain cCREs defined in the other study have a homologous sequence in the mouse genome that also displays chromatin accessibility in one or more mouse brain cell types[23]. Nevertheless, the chromatin-accessibility-conserved cCREs appear to have constraints during evolution, and probably have important regulatory roles in mammalian brain cells. Consistent with a recent report[63], the fraction of cCREs that are classified as chromatin-accessibility conserved in the human brain vary significantly among different brain cell types. Furthermore, the chromatin-accessibility-conserved cCREs tend to be at promoter regions (Extended Data Fig. 13c) and accessible in a broader spectrum of cell types (Extended Data Fig. 13d–f).

## Mouse-specific cCREs are enriched for TEs

Notably, 42% of mouse cCREs defined in mouse brain cells lack orthologous genome sequences in the human genome (Fig. 5a). These mouse-specific cCREs show strong enrichment of TEs, especially the LINEs, SINEs and LTRs (Fig. 5b and Extended Data Fig. 14a). Notably, cCREs defined in 22 subclasses of excitatory neurons display an unusually high rate of overlap with TEs, and we refer to them as highTE-Glut subclasses (Fig. 5c and Extended Data Fig. 14b–e). In total, 20 out of 22 highTE-Glut subclasses were specifically found in the isocortex, OLF and HPF. Notably, the genes near the 115,772 TE-overlapping cCREs, including both mouse-specific and orthologous cCREs, and expressed in at least one of the highTE-Glut neuron subclasses were enriched for those involved in synaptic-related functions (Extended Data Fig. 14f–h). We found 14,619 genes whose expression was positively correlated with chromatin accessibility at 31,137 TE-overlapping cCREs (hereafter, TE-cCREs) across the different subclasses of brain cells, and found that they were also significantly enriched for synapse-related functions (Fig. 5d,e, Extended Data Fig. 14i and Supplementary Table 22). The large number of genes with nearby accessible TE-cCREs is unexpected. To further investigate the genes potentially subject to TE-derived regulatory cCREs, we performed differential chromatin accessibility (DCA) analysis between highTE-Glut and other cell subclasses, and uncovered 1,331 such TE-cCREs. Among them, accessibility profiles at 228 DCA TE-cCREs, including L1MB8, L2 and ORR1E, were correlated with expression of synaptic-related genes (Fig. 5f, Extended Data Fig. 14j and Supplementary Table 23). Motif analysis of these DCA TE-cCREs showed enrichment of many bHLH-family and bZIP-family TFs, such as NeuroG2, TCF4 and FRA1 (Fig. 5g and Supplementary Table 24). Examples of positively correlated TE-cCRE and synaptic-related gene pairs are shown in Fig. 5h and Extended Data Fig. 14k. Furthermore, we examined the superfamilies and families of the DCA TE-cCREs in highTE-Glut, comparing them to all TE-cCREs in highTE-Glut as the background. We observed a significant enrichment of DCA TE-cCREs in the LINE superfamily (FDR = $8.05 \times 10^{-36}$) and the L1 subfamily (FDR = $1.27 \times 10^{-38}$). L1, an actively retrotransposon in both mouse and human, has accumulated in mammalian genomes. It can serve as a source of evolutionary novelties by providing essential motifs[64].

On the basis of the analysis of variability of chromatin accessibility of TEs, we found 90 TEs that display variable patterns of chromatin accessibility across brain cell subclasses (Extended Data Fig. 15a,b). Most of them showed strong negative correlation with DNA CpG methylation signals in the matched cell subclasses. Many of them, such as LTR64, X2_LINE and MamTip1, also showed positive correlations with RNA expression signals in the matched cell subclasses, suggesting a potential role for these TEs in regulating gene expression. We further

performed motif analysis on those variable TEs that may have a regulatory role. We found that distal variable TEs in positive proximal–distal cCRE connections were enriched for many binding sites of TFs, including HF1-halfsite, RORγt and HNF1 (Extended Data Fig. 15c). In addition to the above variable TE families, a greater number of TEs showed invariable chromatin accessibility across brain cell types (Extended Data Fig. 15d).

## Deep-learning models for brain cCREs

Deep-learning models have shown great promise in the dissection of gene regulatory mechanisms[65–69]. Sequence-based predictors of gene expression or epigenetic features have been developed for large mammalian genomes using cell-type-specific epigenetic and transcriptional profiles as training data[65,67,70]. These models can help to annotate sequence motifs that drive regulatory element function, and to predict the influence of DNA variants on gene regulation. To develop sequence-based predictors of chromatin accessibility in different brain cell types (Fig. 6a and Methods), we adapted the deep-learning model architecture Basenji, which uses densely connected dilated convolution neural networks that are used in natural language processing tasks[65]. We generated training, validation and testing datasets (Methods) from the 275 subclasses (also referred to as cell types in this section) and evaluated the model on the 221 subclasses with at least 500 cells including 93 GABAergic and 111 Glut cell subtypes, and 17 non-neuronal types (Fig. 6b). The resulting model successfully predicted open chromatin regions across these cell types, with an average PCC of 0.825 between the predicted signals and true chromatin accessibility signals across cell types (Fig. 6c). To further improve the model performance in under-represented cell types, we introduced a weighted loss function to enable the model to better learn the cell-type-specific signals during training (Methods). To compare the peaks identified from experimental signals to the peaks called from predicted signals, we calculated the area under the receiver operating characteristic (AUROC) and demonstrated that the model can predict the open chromatin regions very well (from 0.72 to 0.94, and 0.85 on average) for different cell types (Fig. 6d and Supplementary Table 25). This high performance was comparable to the prediction of chromatin accessibility signals from the most advanced deep-learning model[67]. We further evaluated the model's ability to predict cell-type-specific chromatin accessibility at each cCRE across the diverse cell subclasses, achieving a median PCC of 0.59 for the variable cCREs (coefficient of variation > 1) in the testing set (Fig. 6e). To demonstrate the performance of our model, we visualized predictions in unseen test regions among 12 cell types representing diverse brain regions, cell classes and neurotransmitters (Fig. 6f). Our model not only recapitulated signals that were common across subclasses (*Nr4a2*), but also showed subclass-specific predictions. For example, signals around *Apoe* were specific in astrocytes (Astro-TE-NN and Bergmann-NN) and signals around *Ecel1* were specific in neurons.

While still poorly characterized, the grammar and syntax of gene regulatory elements are believed to be evolutionarily conserved[71]. We therefore tested how well the above-described deep-learning model trained using mouse single-cell chromatin accessibility data can predict cCREs in the matched human brain cell types with human sequences as inputs[23] (Fig. 6g and Extended Data Fig. 16). Satisfyingly, we found that the mouse deep-learning model can predict chromatin accessibility profiles in the matching human brain cell types fairly accurately (AUROC, 0.75 on average) (Fig. 6h). It achieves modest accuracy in predicting cell type specificity among cCREs (median PCC = 0.41) (Fig. 6i). The cell-type-specific distal cCREs, such as the ones close to marker genes *CUX2*, *GAD2*, *DRD1* and *OLIG1*, were well predicted (Fig. 6j). These results open a window to evaluate the influence of risk variants on regulatory activities across corresponding cell types in the human brain.

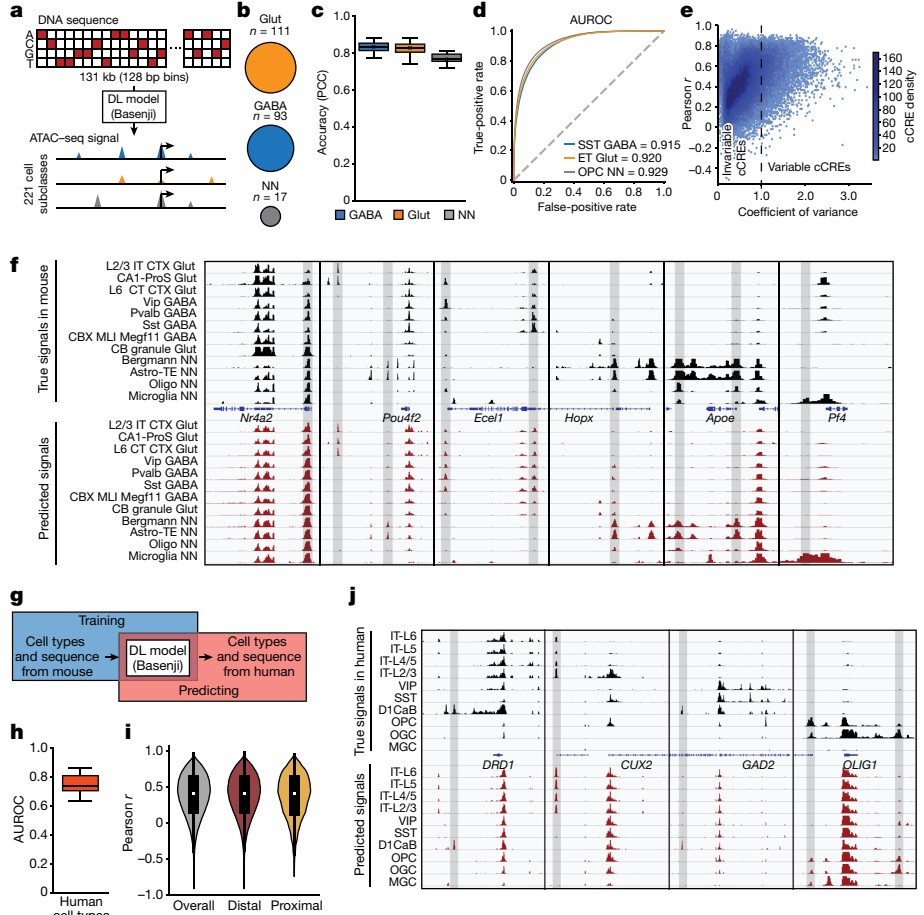

**Fig. 6 | Deep-learning models predict chromatin accessibility in different brain cell types from the DNA sequence. a**, Schematic of the deep-learning (DL) model Basenji for predicting chromatin accessibly. **b**, The number of subclasses of each cell class in the training dataset. **c**, The accuracy (Pearson correlation) of each class. $n = 93$ (GABA), $n = 111$ (Glut) and $n = 17$ (NN) subclasses. **d**, The AUROC was calculated for representative subclasses by comparing the peaks called from predicted genomic signals with the peaks called from real experimental signals. **e**, The model's ability to predict cell-type-specific patterns of open chromatin. The coefficient of variance (variance/mean) across cell types was compared with the Pearson $r$ calculated between true signals and the predicted signals across cell subclasses. Each dot represents one cCRE in the testing set. **f**, True signals from ATAC–seq data in mouse cell subclasses were compared with the predicted chromatin accessibility in the test set.

Representative loci near *Nr4a2, Pou4f2, Ecel1, Hopx, Apoe* and *Pf4* are shown. **g**, Schematic of predicting potential chromatin accessibility signals using human DNA sequence as inputs. **h**, The AUROC was calculated for matched human cell types. $n = 26$ cell types for the human brain dataset. **i**, The Pearson $r$ of true signals and the predicted signals across cell types for all tested cCREs, tested distal cCREs and tested proximal cCREs. The numbers of overall, distal and proximal cCREs are 452,531, 437,207 and 15,324, respectively. **j**, True signals captured from ATAC–seq analysis in human cell types and predicted chromatin accessibilities are shown at representative genomic loci near the genes *CUX2, GAD2, DRD1* and *OLIG1*. Cell-type-specific cCREs are highlighted in grey. For the box plots, the box limits span the first to third quartiles, the centre line denotes the median and the whiskers show 1.5× the interquartile range.

## Discussion

Here we describe a comprehensive cCRE catalogue of the mouse brain, through single-cell chromatin accessibility analysis of more than 2.3 million cells from 117 anatomical dissections in the adult mouse brain. This catalogue represents a comprehensive annotation of candidate gene regulatory elements of the mammalian brain. It greatly expands on the previous cCRE annotation of the mouse brain cells, adding more than 460,000 cCREs. This addition is enabled by the use of single-cell-resolution chromatin profiling, which enables the identification of chromatin accessibility in rare brain cell types that are under-represented in previous bulk assays and brain regions that were not surveyed in previous studies. Indeed, more than two-thirds of the new cCREs are detected in ten or fewer brain cell subtypes (Fig. 2d), with a median of six cell subtypes. By comparison, the cCREs reported in the previous catalogues[39,40] based on bulk tissue studies are typically detected as accessible in ten or more cell types, with a median of 28 cell subtypes. It is possible that additional mouse brain cCREs

remain to be discovered because many cell types defined by scRNA-seq or other molecular modalities are not currently represented in the snATAC-based cell clusters. Furthermore, the current catalogue was at the resolution of cell subclasses, and may not reflect subtle differences between cell types, subtypes and states defined in the companion single-cell transcriptomics or single-cell methylome studies[5,6,41].

We have attempted to reconstruct the GRNs in over 260 different brain cell subclasses by applying CellOracle[52] to the single-cell ATAC–seq and RNA-seq datasets collected from the adult mouse brain. The GRNs that we inferred for brain cells would be the first such GRNs characterized for the mammalian brain cells. We characterized the common network motifs in these cell types. Indeed, the GRN-based eigenvector centralities of TFs across the subclass (Fig. 4f) showed similar pattern in the scRNA-seq study[5]. There is a limitation to the GRNs inferred using the CellOracle strategy. For example, owing to the use of a regression model, CellOracle cannot infer autoregulatory loops. Besides, the double-negative network motif (A inhibits B and B inhibits A) was seldom predicted, potentially also due to the limitation of using a regression

model. In our opinion, instead of treating all of the cells in one population in such a static way, the pseudotime reconstruction models[72–75] from the single-cell data can be used to organize the cells in a dynamic manner, which would enable time-series-related models[76,77] to be used to predict the autoregulatory loops and the double-negative-network-motif-like structures. Indeed, a recent method, Dictys[78], uses stochastic process modelling to infer the feedback loops. Furthermore, to have more confident GRNs, from the computational view, multiple methods from different aspects can be combined to provide diverse evidence[43].

We investigated the sequence conservation of gene regulatory elements in the whole mouse brain by comparing the cCRE atlas in the mouse brain defined in the present study to a cCRE atlas obtained from a separate snATAC–seq analysis of 42 adult human brain regions in three adult male donors. We found that around 22% of cCREs defined in the current study are conserved in both sequence and in chromatin accessibility in the human brain. This modest number of conserved cCREs is probably due to the still incomplete cataloguing of cCREs in the human brain cells. Nevertheless, the cCREs showing conserved chromatin accessibility and sequence in both the mouse and human brains are clearly under evolutionary constraints and, therefore, probably possess functional importance. Consistent with previous reports, the chromatin-accessibility-conserved cCREs tend to be promoters or distal elements (probable enhancers) that display accessibility in a broader spectrum of cell types[24,63]. By contrast, the mouse-specific cCREs are strongly enriched for TEs, implicating a potential role of TEs in cell-type-specific gene expression patterns in the mouse brain. The finding is consistent with previous observations of TE reactivation in development and in various tissues[79]. Note that the strongest enrichment of TE in cCREs is observed especially in 20 Glut (excitatory) neurons from the isocortex, OLF and HPF. We speculate that TEs may contribute positively to transcriptional regulation and chromatin structure in these cells. In support of this possibility, nearly 1,300 TE-overlapping cCREs display positive correlation between chromatin accessibility and mRNA levels from potential target genes. Their putative target genes include those involved in synaptic function and synapse organization. Our results raise the interesting possibility that neural circuit diversity could be influenced by TEs during evolution.

By extracting the context information from DNA sequence, deep-learning methods have recently been used for the prediction of various genomic functional features, such as epigenetic modifications, 3D interactions and gene expression[65–69]. We adapted this approach to develop sequence-based models to predict the chromatin accessibility in 275 mouse brain cell subclasses. We achieved excellent performance comparable to the prediction of ATAC–seq signals from the most recent attention-based model architecture[67]. Although previous efforts have attempted to train deep-learning models simultaneously on multiple genomes[80], evaluation of how well the sequence-based predictors trained in one species can be applied to a different species is lacking for matched cell types between species. Our results demonstrate that deep-learning models trained using open chromatin landscapes in the mouse brain cell types generalize well in the corresponding human brain cell types.

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

## Methods

### Tissue preparation and nucleus isolation

All experimental procedures using live animals were approved by the SALK Institute Animal Care and Use Committee under protocol number 18-00006. Adult C57BL/6J male mice were purchased from Jackson Laboratories. Brains were extracted from 56–63-day-old mice and sectioned into 600 μm coronal sections along the anterior–posterior axis in ice-cold dissection medium[2,83]. Specific brain regions were dissected according to the Allen Brain Reference Atlas[26] (Extended Data Fig. 1) and nuclei were isolated as described previously[26]. For each region, dissected brain tissues were pooled from 2–31 (only 2 dissections from the mouse CB region had 2 animals for snATAC–seq library construction, all of the other samples had 4–31 animals) of the same sex to obtain enough nuclei for snATAC–seq for each biological replica, and two biological replicas were performed. We shared the same fluorescence-activated cell sorting (FACS) sequential gating/sorting strategy and the Sony SH800S software with our previous study[19].

### scATAC–seq analysis

snATAC–seq libraries were generated as described using version 2 indexing[19]. PCR amplification was performed for 11 or 12 cycles. A step-by-step-protocol for library preparation is available online (https://doi.org/10.17504/protocols.io.4zzgx76). Libraries were sequenced using the HiSeq 2500 (Illumina), a HiSeq 4000 (Illumina) or NovaSeq 6000 (Illumina) system with the following settings: 50 + 10 + 12 + 50 (read1 + index1 + index2 + read2).

### Processing and alignment of sequencing reads

Paired-end sequencing reads were demultiplexed and the cell index was transferred to the read name. Sequencing reads were aligned to the mm10 reference genome using bwa[84]. After alignment, we checked the fragment length contribution, which is characteristic for ATAC–seq libraries (Extended Data Fig. 2e) for each of the 234 samples. We then combined the sequencing reads to fragments using the make_fragment_file function of SnapATAC2[29] and, for each fragment, we applied the following quality control criteria: (1) retain only fragments with quality scores MAPQ > 30; (2) remove PCR duplicates. Reads were also sorted on the basis of cell barcodes in read names, and shifted +4 bp for positive strand and −5 bp for negative strand to correct the 9 bp duplication induced from Tn5 transposase[85] during processing.

### TSSe calculation

Enrichment of ATAC–seq accessibility at TSSs was used to quantify data quality without the need for a defined peak set. We followed a previously described procedure[86], and used the function filter_cells in SnapATAC2 to calculate TSS enrichment (TSSe). TSS positions were obtained from the GENCODE[87] database v.16. In brief, Tn5-corrected insertions (reads aligned to the positive strand were shifted +4 bp and reads aligned to the negative strand were shifted −5 bp) were aggregated ±2,000 bp relative (TSS-strand-corrected) to each unique TSS genome wide. This profile was then normalized to the mean accessibility ±1,900–2,000 bp from the TSS and smoothed every 11 bp. The maximum of the smoothed profile was taken as the TSSe.

### Nucleus filtering by quality control

Nuclei with ≥1,000 uniquely mapped fragments and TSSe ≥ 10 were filtered for each of 234 samples according to the ENCODE ATAC–seq data standards and process pipeline (https://www.encodeproject.org/atac-seq/). We used the filter_cells function of SnapATAC2 to achieve this.

### Doublet removal

We used a modified version of Scrublet[28] to remove potential doublets for every sample independently using SnapATAC2. First, we used the add_tile_matrix function to add the 500 bp genomic bin features, then used the select_features function to filter out the features with frequencies along the samples of lower than 0.5% or higher than 99.5%. We then applied the scrublet function of SnapATAC2 to get the doublet scores. The parameter expected_doublet_rate was set to 0.08, which is based on our previous experiment on the snATAC–seq pipeline[19]. Barcodes with scrublet scores of greater than 0.5 were treated as potential doublets and removed from our analysis.

We compared Scrublet with another recently published method named AMULET[30], which is used for doublet detection and removal in snATAC–seq data. We simulated datasets containing singlets and artificial doublets from eight samples in the primary motor area and evaluated the performances of the two methods using precision-recall curve (PRC) and area under PRC (AUPRC).

### Iterative cell clustering

After nucleus filtering by quantity control and doublet removal, we adapted a fourth-round iterative clustering using SnapATAC2 for later identification of cell-type-specific cCREs (Extended Data Fig. 4a). The following basic procedure was used. For the first round of clustering (L1-level clustering), we used all of the 2.3 million nuclei to perform the standard clustering. At the second round (L2-level), for each of the 37 clusters above, we performed independent clustering. At the third round (L3-level), for each of the 248 clusters above, we performed independent clustering again. At the fourth round of clustering (L4-level clustering), we performed only clustering for the L3-level clusters with number of cells no less than 400. The details are as follows.

**Feature selection.** We applied the function add_tlle_matrix from SnapATAC2 to extract the cell by genomic bin count matrix. The size of a consecutive genomic region was chosen as 500 bp. We filtered out any bins overlapping with the ENCODE blacklist and removed the top 0.5% and tail 0.5% bins based on the read coverage from the count matrix. Only chromosomes 1–19, X and Y were considered. For our L1-level clustering, we used all of the bin features (over 4 million) that passed the criteria above as non-neuronal cells and diverse neuronal cells were all included. For clustering of other levels, we chose the default top 500,000 features using the function select_features of SnapATAC2.

**Dimensionality reduction.** We applied the function of spectral from SnapATAC2 to convert the high-dimension sparse 500 bp genomic bin features per cell into low dimensional representations, which used spectral embedding of the normalized graph Laplacian defined by the cell-to-cell similarity matrix using cosine distance. For L1-level and L2-level clustering, we chose 50 as the dimension of the low-dimensional representation space as usually a large number of cells and potentially diverse cell types was involved in the two levels. We used 'elbow plot' to rank all of the principal components to make sure that the top 50 components were sufficient for our analysis. For later analysis, we chose 30 instead. The parameter 'weighted_by_sd' in the function spectral was set to be true for all dimensional reduction. We did not use the parameter 'sample_size' in the function spectral, so no approximation method was used for the spectral embedding. For 2.3 million cells, it took about 300 GB memory in our high-performance computing system[88].

**Graph-based clustering.** We then applied the function knn from SnapATAC2 to construct the *k*-nearest neighbour graph using the parameter n_neighbors = 50 and the parameter method was set to 'kdtree'. We next used the function leiden of SnapATAC2 for clustering with the parameter object_function set as modularity. The parameter resolution, which affected the number of clusters a lot, was selected from 0.1 to 2 with a step size 0.1 based on the silhouette coefficient[89] using the Python package Scikit-learn[90]. We also manually checked the UMAP[81] for each clustering result to make sure that the resolution was suitable corresponding to the top silhouette coefficient. UMAP

projections were calculated using the Python package umap with the parameters a as 1.8956, b as 0.8005 and init as spectral. All of the resolution parameters during clustering are provided in Supplementary Table 3. In our later analysis, we used the term subtypes to represent all of the final clusters from L3-level clustering and L4-level clustering.

### Integration analysis with scRNA-seq data

We performed integration analysis of the 1,482 subtypes with all of over 5,300 clusters reported in a companion scRNA-seq study of 4.5 million cells for the whole adult mouse brain[5]. Only cells from male mice were considered in the scRNA-seq data, which is over 2 million cells. The scRNA-seq data are mainly from 10x v.2 and 10x v.3 platforms, and only a few thousand cells are from snRNA-seq. On the basis of our integration analysis, we did not see significant differences between using 10x v.3 alone and using all of them. Very few cell clusters were found using the 10x v.2 but not using the 10x v.3 platform. We therefore used all of the cells without distinguishing their platform information in the later analysis.

We first imputed RNA expression levels according to the chromatin accessibility of the gene promoter (up to 2 kb to TSSs) and gene body as described previously[32] using the function make_gene_matrix in SnapATAC2. We next performed integration analysis using Seurat[32] for neuronal cells and non-neuronal cells separately. For neuronal cells, in the scRNA-seq data, we randomly selected 50 cells for each of over 5,100 clusters, and finally got more than 200,000 cells. To have a comparable number of cells in our snATAC−seq data, we randomly selected 150 cells for each of over 1,260 L4-level neuronal subtypes and got over 180,000 nuclei. For non-neuronal cells, we sampled 500 cells per cluster and got 35,000 cells in the scRNA-seq data. For the snATAC−seq, we sampled 300 cells per L4-level subtypes, and got over 57,000 nuclei.

For the variable features, we applied the >8,000 genes from differential expression analysis in the scRNA-seq study[5], and used their data as the reference. We next applied the canonical component analysis for integration using Seurat v.5. Canonical component analysis was recommended for the cross-modality integration, which indeed showed more promising results than reciprocal principal component analysis in our experiments. Seurat v.5 is specifically designed to handle large-scale datasets and is especially important for our scenario. We used the function FindTransferAnchors with the parameter k.anchor as 50 for single-cell level label transfer. k.anchor is important for large-scale data integration as mentioned in Seurat. The default k.anchor value is 5 for that function, and we tested k.anchor as 5, 10, 30, 50, 70, 100 and 120; a k.anchor value of 50 showed more reliable results compared with others. For UMAP visualization, we used the FindIntegrationAnchors function of Seurat, and then calculated UMAP based on the co-embedding space. It was also recommended by Seurat to perform integration in this manner. The transfer label scores for a given L4-level subtype in our snATAC−seq data is a numeric vector, where each element is the number of cells annotated as the corresponding cluster in the scRNA-seq data divided by the number of cells in that L4-level subtype. For each L4-level subtype, we used the corresponding top 3 clusters in the scRNA-seq data as the candidate annotations, then mapped the three clusters to the subclasses defined in the scRNA-seq data, and manually checked whether they were consistent on mouse brain major regions and gene markers.

### Identification of reproducible peak sets in each cell cluster

We performed peak calling according to the ENCODE ATAC−seq pipeline (https://www.encodeproject.org/atac-seq/) on 1,482 L4-level subtypes and used the same procedure to filter the peaks at both the bulk and single-cell level (Extended Data Fig. 9a) as in our previous study[19]. Before calling peaks, we merged clusters with the number of cells less than 200 if they shared the same cell cluster annotation based on the integration analysis before and were in the same L3-level cluster. Next, 1,463 subtypes (including merged ones) were used.

For every cell cluster above, we combined all properly paired reads to generate a pseudobulk ATAC−seq dataset for individual biological replicates. Moreover, we generated two pseudoreplicates comprising half of the reads from each biological replicate. We called peaks for each of the four datasets and a pool of both replicates independently. Peak calling was performed on the Tn5-corrected single-base insertions using MACS2[36] with the following parameters: --shift -75 --extsize 150 --nomodel --call-summits --SPMR -q 0.01. Finally, we extended peak summits by 250 bp on either side to a final width of 501 bp for merging and downstream analysis. If the number of cells in any of the pseudobulk ATAC−seq from either individual biological replicates or individual pseudoreplicates is fewer than 200, we did not run MACS2 for it. We did this to reduce the potential false negatives during the next filtering step induced by the limited number of cells in the replicates.

To generate a list of reproducible peaks, we retained peaks that (1) were detected in the pooled dataset and overlapped ≥50% of peak length with a peak in both individual replicates or (2) were detected in the pooled dataset and overlapped ≥50% of peak length with a peak in both pseudoreplicates.

We found that, when the cell population varied in read depth or number of nuclei, the MACS2 score varied proportionally due to the nature of the Poisson distribution test in MACS2[19]. Ideally, we should perform a reads-in-peaks normalization but, in practice, this type of normalization is not possible because we do not know how many peaks we will get. To account for differences in the performance of MACS2 based on read depth and/or number of nuclei in individual clusters, we converted MACS2 peak scores ($-\log_{10}[q]$) to SPM[37]. We filtered reproducible peaks by choosing a SPM cut-off of 5.

We then retained only reproducible peaks on chromosome 1–19 and both sex chromosomes and filtered ENCODE mm10 blacklist regions. A union peak list for the whole dataset was obtained by merging peak sets from all of the cell clusters using BEDtools[91].

Finally, as snATAC−seq data are very sparse, we selected only elements that were identified as open chromatin in a significant fraction of the cells in each cluster. To this end, we first randomly selected the same number of non-DHS regions from the genome as background using the shuffleBed function of BEDtools, and calculated the fraction of nuclei for each cell type that showed a signal at these sites. We next fitted a zero-inflated $\beta$-model, and empirically identified a significance threshold of FDR < 0.01 to filter potential false positive peaks. Peak regions with FDR < 0.01 in at least one of the clusters were included in downstream analysis. Given one cell subclass, we treat all of the peaks from the subtypes mapped to this subclass as the peaks for the subclass.

### Identification of *cis*-regulatory modules

We used NMF[42] to group cCREs into cis-regulatory modules on the basis of their relative accessibility across major clusters. We adapted NMF (Python package sklearn[90]) to decompose the cell-by-cCRE matrix $V$ ($N \times M$, $N$ rows: cCRE, $M$ columns: cell clusters) into a coefficient matrix $H$ ($R \times M$, $R$ rows: number of modules) and a basis matrix $W$ ($N \times R$), with a given rank $R$[19]:

The basis matrix defines module-related accessible cCREs, and the coefficient matrix defines the cell cluster components and their weights in each module. The key issue to decompose the occupancy profile matrix was to find a reasonable value for the rank $R$ (that is, the number of modules). Several criteria have been proposed to decide whether a given rank $R$ decomposes the occupancy profile matrix into meaningful clusters. Here we applied two measurements, Sparseness[92] and Entropy[42], to evaluate the clustering result. Average values were calculated from five NMF runs at each given rank with a random seed, which ensures that the measurements are stable (Extended Data Fig. 9f).

We next used the coefficient matrix to associate modules with distinct cell clusters. In the coefficient matrix, each row represents a module, and each column represents a cell cluster. The values in the matrix

indicate the weights of the clusters in their corresponding module. The coefficient matrix was then scaled by column (cluster) from 0 to 1. Subsequently, we used a coefficient > 0.1 (~95th percentile of the whole matrix) as a threshold to associate a cluster with a module.

Moreover, we associated each module with accessible elements using the basis matrix. For each element and each module, we derived a basis coefficient score, which represents the accessible signal contributed by all clusters in the defined module. We also implemented and calculated a basis-specificity score called feature score for each accessible element using the kim method[42]. The feature score ranges from 0 to 1. A high feature score means that a distinct element is specifically associated with a specific module. Only features that fulfil both following criteria were retained as module specific elements: (1) feature score greater than median + 3s.d.; (2) the maximum contribution to a basis component is greater than the median of all contributions (that is, of all elements of $W$).

## Inference of *cis*-co-accessible cCREs

*Cis*-co-accessibility cCREs are predicted for all open regions in each of the 275 cell subclasses separately using Cicero for Monocle 3[72,93] with the default parameters and the mouse mm10 genome, scanning the mouse genome with a window size of 500 kb. For each subclass, we randomly selected 5,000 nuclei, and used all of the nuclei for cell clusters with <5,000 nuclei. Only one subclass failed during running Cicero, which was annotated as 'Hypendymal_NN' with 92 nuclei in total, and showed the smallest number of peaks (less than 5,000) of all of the subclasses. To find an optimal co-accessibility threshold for each subclass, we randomly shuffled the columns of the cell-by-cCREs matrix (that is, the cCREs) in the cells as the background and identified co-accessibility regions from this shuffled matrix. A normal distribution is then used to fit the co-accessibility scores from the shuffled background using the R package fitdistrplus[94]. Co-accessibility cCREs were filtered out only if their co-accessibility scores were significantly larger than the background (FDR < 0.001 using Benjamini–Hochberg adjustment). CCREs outside of ±1 kb of TSSs in GENCODE mm10 version 23, were treated as distal cCREs, others as proximal ones. All of the *cis*-co-accessibility cCREs were then grouped into three classes: proximal-to-proximal, distal-to-distal and distal-to-proximal pairs. In our study, we focused only on distal-to-proximal pairs.

## Enrichment analysis of FIREs

We called frequently interacting regions (FIREs) in the mouse cortex[38] by applying the criteria in our group's FIRE paper[95]. The result showed that most FIREs (3,158 out of 3,169) overlap with cCREs in the mouse brain, and a fraction of the cCREs (71,626 out of 1,053,811) overlap with FIREs (Extended Data Fig. 10e).

We next tested whether cCREs are enriched at FIREs through permutation analysis. In brief, we shuffled the mouse genome 1,000 times, each time generating 1,053,811 random regions with equivalent sizes as the cCREs. We then calculated the number of overlaps between the randomly generated regions and the FIREs during each shuffle. We found that cCREs are significantly enriched at FIREs ($P < 0.001$; Extended Data Fig. 10f), with the actual number of overlaps on FIREs substantially higher than expected.

## Motif enrichment

We performed both de novo and known motif-enrichment analysis using Homer[45].

## Enrichment analysis of chromatin conformation

We cross-referenced the dataset from the companion study[41], in which a comprehensive chromatin conformation/methylome joint profile throughout the adult mouse brain is described, and most of the subclass annotations (244 subclasses of 275 subclasses in our data) are shared between these two datasets.

To evaluate the confidence of identified subclass-specific cCRE–gene pairs, we randomly selected 11 major subclasses (Sst_GABA, Pvalb_GABA, CBX_MLI_Megf11_GABA, Vip_GABA, CA1-ProS_Glut, CB_granule_Glut, L6_CT_CTX_Glut, L2-3_IT_CTX_Glut, Astro-TE_NN, Microglia_NN, Bergmann_NN), and calculated the Hi-C signal enrichment (at 1 kb resolution) at the top 20% subclass-specific cCRE–gene pair anchors identified in this study. We found that there is statistically significant higher enrichment ($P = 0.004$) of chromatin interaction signal at the corresponding subclass-specific cCRE–gene pair anchors, compared with non-corresponding pair anchors (Extended Data Fig. 10g), suggesting that subclass-specific cCRE–gene pairs are more likely to interact in the cell types in which the cCREs are active.

Meanwhile, we selected the two peak modules that show global accessibility across the subclasses based on the NMF analysis (Fig. 2f (top left)). We then selected all of the proximal–distal connections with cCREs in the peak modules above and ranked the proximal–distal connections based on the highest Cicero scores they have. We treated them as global proximal–distal connections and performed the Hi-C signals by aggregating all of the Hi-C data. From the heat maps (Extended Data Fig. 10h), we observed the strong enrichment signals for the global proximal–distal connections.

## Predicting GRNs for each cell subclass

We adapted the recently published Python package CellOracle[52] on our data to infer GRNs for each cell subclass across the whole mouse brain based on our integration analysis between our snATAC–seq data and the scRNA-seq data[5]. Three steps were followed. First, we identified the co-accessibility distal-to-proximal pairs, which was described previously for each subclass. Second, we mapped the distal cCREs to TFs. Lastly, we identified the regulatory relationships between TFs and the potential target genes by fitting a regularized linear regression model using scRNA-seq data. For the second step, according to the CellOracle tutorial, we used the Python package gimmemotifs[53] for the TF-binding-motif scan with the mouse genome mm10 and the default motif database provided by CellOracle. The proximal cCREs were mapped to the genes based on GENCODE mm10 (v.23, the same as above). We used Seurat[32] to randomly sample 1,000 cells per subclass (all of the cells of a cell subclass were used if it had <1,000 cells). To select the variable features, we performed the FindVariableFeatures function of Seurat to select the top 3,000 genes, and then we manually added the 499 TFs (if any of them were missed in the previous 3,000 genes) that were reported in the scRNA-seq data of ref. 5. For each subclass, we performed CellOracle on the scRNA-seq data with the default parameters. We used $P < 0.001$ and the top 10,000 edges based on the absolute values of the weights to filter the predicted interactions between TFs and genes as suggested by CellOracle. Finally, 267 out of 275 subclasses successfully had the predicted GRNs.

## Sequence conserved, chromatin accessibility conserved and mouse-specific cCREs

The orthologous cCREs of the mouse brain in the human genome were identified by performing reciprocal homology searches using the liftover tool[96]. The mouse cCREs for which human genome sequences had high similarity (more than 50% of bases lifted over to the mouse genome) were defined as orthologous cCREs. We next compared these orthologous cCREs in the mouse brain with our previously identified cCREs in the human brain[23]. Those orthologous cCREs, which both were DNA sequence conserved across species and had open chromatin in orthologous regions, were defined as chromatin-accessibility-conserved cCREs. The other orthologous cCREs, which were only sequence conserved to orthologous regions but had not been identified as open chromatin regions in other species, were defined as chromatin-accessibility-divergent cCREs. Mouse-specific cCREs were those ones that were not able to find orthologous regions in the human genome.

## TE analysis

The TE annotation of cCREs was annotated using Homer[45] and UCSC mm10 refGene and RepeatMasker annotation. To define the high TE-cCREs fraction of subclasses, we fitted a mixture model for the TE-cCRE fraction across all subclasses using the R package mixtools[97] (v.2.0.0). The P value was calculated based on the null distribution.

To annotate the TE-cCREs, we used two strategies. One was based on the genomic regions. We mapped the TE-cCREs to genes within 3 kb flanking regions using the R package ChIPseeker[98] (v.1.34.1). Another method to link the gene to TE-cCREs was based on the cCREs and gene correlation. For each GO test, we also filtered unexpressed genes in defined subclasses based on the single-cell RNA-seq data (see the companion manuscript[5]). The DCA of TE-cCREs between groups was calculated using the Wilcoxon rank-sum test. Motif-enrichment analysis of TE-cCREs was performed using Homer software using the 'given size' parameter.

To analyse the TE-accessible variability with decreased noise, the TE signal was aggregated from the TE-cCREs. To calculate the correlation between chromatin accessibility and mCG methylation in TEs across subclasses, we averaged and normalized the TE-cCRE mCG signal for each TE in matched subclasses from the companion paper[41]. To calculate the correlation between chromatin accessibility and RNA expression, we aggregated RNA signals at TE-cCREs of each TE in matched subclasses from a previous study[99].

## GO enrichment

We performed GO enrichment analysis using R package clusterProfiler[100,101]. The background genes were selected on the basis of the enrichment analysis and described in text. The P value was computed using the Fisher exact test and adjusted for multiple comparisons using the Benjamini–Hochberg method.

## Deep-learning model

Our model was trained on all 275 subclasses annotated based on the integration with the scRNA-seq data. We generated aggregated genome signal tracks in bigwig format by running MACS2[36]. The training, validation, and testing datasets have been generated using the script basenji_data.py from Basenji[65] with the parameters: "-b mm10.blacklist.bed -l 131072 --local -p 16 -t 0.1 -v 0.1 -w 128".

The model architecture, layers and parameters are adapted from the mouse model from a previous study[80], with modification only in the last output head layer with parameter: "units": 275. To encourage the model to predict cCREs in under-represented cell types, we created one novel loss function:

$$w_{i,i} = \mathrm{cov}(y_{\mathrm{true}(i,i)})$$

$$w = \sum_{i=1}^{n} w_{ii}/n$$

$$\text{Poisson loss} = y_{\mathrm{pred}(i,j)} - y_{\mathrm{true}(i,j)} \times \log(y_{\mathrm{pred}(i,j)})$$

$$\text{loss function} = w \cdot \text{Poisson loss}$$

The $i$ represent the cell type, and $j$ represents genomic bins. The $y_{\mathrm{true}}$ represents the genomic bin-by-type matrix calculated from true signals. The $y_{\mathrm{pred}}$ represents the predicted genomic bin-by-type matrix. The pairwise covariance $w_{i,i}$ was calculated between cell types. We then sum the scores across rows and normalize the number of cell types as weights. Last, the weights $w$ was dot multiplied by the original poisson loss.

We trained the subclass-level deep-learning model on four NVIDIA A100 80 GB GPUs using the script basenji_train.py from Basenji[65].

For training, we set the parameter batch size to 32, epochs to 150 and patience to 30.

To evaluate the model's ability to identify cell-type-specific patterns of cCRE, we compared the Spearman correlation of model predictions to true accessibility across cell types in all peaks in the test set. We further compared cross-cell-type correlation to the coefficient of variation (the ratio of s.d. to mean) of each peak.

We also evaluated the model's accuracy when applied to human cell types. We first identified matched human cell types from a previous study[23]. For each subclass in human and mouse cCREs, we performed spearman correlation across orthologous cCREs (Extended Data Fig. 16). We next selected pairs based on correlation and annotation matching. We then used the model to predict chromatin accessibility in the paired human cell types, across all chromosomes. We further evaluated this prediction accuracy within and across cell types.

## External datasets

External datasets used were as follows: (1) ENCODE rDHS regions for both hg19 and mm10 are obtained from SCREEN database (https://screen.encodeproject.org)[39,40]. (2) ChromHMM[38,102] states for mouse brain are download from GitHub (https://github.com/gireeshkbogu/chromatin_states_chromHMM_mm9) and coordinates are LiftOver (https://genome.ucsc.edu/cgi-bin/hgLiftOver) to mm10 with the default parameters[96]. (3) PhastCons[103] conserved elements were downloaded from the UCSC Genome Browser (http://hgdownload.cse.ucsc.edu/goldenpath/mm10/phastCons60way/). (4) The ENCODE mm10 blacklist file was downloaded from http://mitra.stanford.edu/kundaje/akundaje/release/blacklists/mm10-mouse/mm10.blacklist.bed.gz. (5) Mouse mm10 genome information was downloaded from GENCODE (https://www.gencodegenes.org/mouse/).

## Statistics

No statistical methods were used to predetermine sample sizes. There was no randomization of the samples, and investigators were not blinded to the specimens being investigated. However, clustering of single nuclei based on chromatin accessibility was performed in an unbiased manner, and cell types were assigned after clustering. Low-quality nuclei and potential barcode collisions were excluded from downstream analysis as described above.

## Reporting summary

Further information on research design is available in the Nature Portfolio Reporting Summary linked to this article.

## Data availability

Demultiplexed FASTQ files are available at the NEMO archive (NEMO, RRID: SCR_016152) at https://assets.nemoarchive.org/dat-bej4ymm (the raw directory under the source data URL in this archive), and at the NCBI under GEO accession number GSE246791. Processed data are available at our web portal (http://www.catlas.org) and the same GEO accession number above.

## Code availability

Custom code and scripts used for analysis are available at GitHub (https://github.com/beyondpie/CEMBA_wmb_snATAC).

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

**Acknowledgements** We thank all of the other members of the Ren laboratory for their input. This study was supported by NIH grant U19MH114831 to J.R.E. and B.R., and NIH grant U19MH114830 to H.Z. J.R.E. is an investigator of the Howard Hughes Medical Institute. Zhaoning Wang is a DDBrown Awardee of the Life Sciences Research Foundation. Work at the Center for Epigenomics was also supported by the UC San Diego School of Medicine. This publication includes data that were generated at the UC San Diego IGM Genomics Center using an Illumina NovaSeq 6000 system that was purchased with funding from a National Institutes of Health SIG grant (S10 OD026929).

**Author contributions** Study supervision: B.R. Contribution to data analysis: S.Z., Y.E.L., K.W., E.A., S.M., Y.W., M.L.A., H.L., J.Z., H.Z. and J.S. Contribution to data generation and management: S.Z., S.P., Y.E.L., A.W., X.H., M.M., S.K., J.O., J.L., A.P.-D., M.M.B., H.Z., Z.Y., B.L., K.A.S., M.N., B.J., L.L., Q.Y. and S.L. Contribution to the web portal: Y.E.L. and S.Z. Contribution to data interpretation: S.Z., Y.E.L., K.W., E.A., S.P., B.R., J.R.E., M.M.B., B.T., H.Z., J.X., Zihan Wang, S.M., Y.X., K.Z. and A.C. Contribution to writing the manuscript: S.Z., Y.E.L., B.R., K.W., H.Z., B.T., S.P., M.M.B., J.X., Zhaoning Wang and S.M. All of the authors edited and approved the manuscript.

**Competing interests** B.R. is a co-founder and consultant of Arima Genomics and co-founder of Epigenome Technologies. J.R.E. is on the scientific advisory board of Zymo Research. H.Z. is on the scientific advisory board of MapLight Therapeutics.

**Additional information**
**Correspondence and requests for materials** should be addressed to Bing Ren.

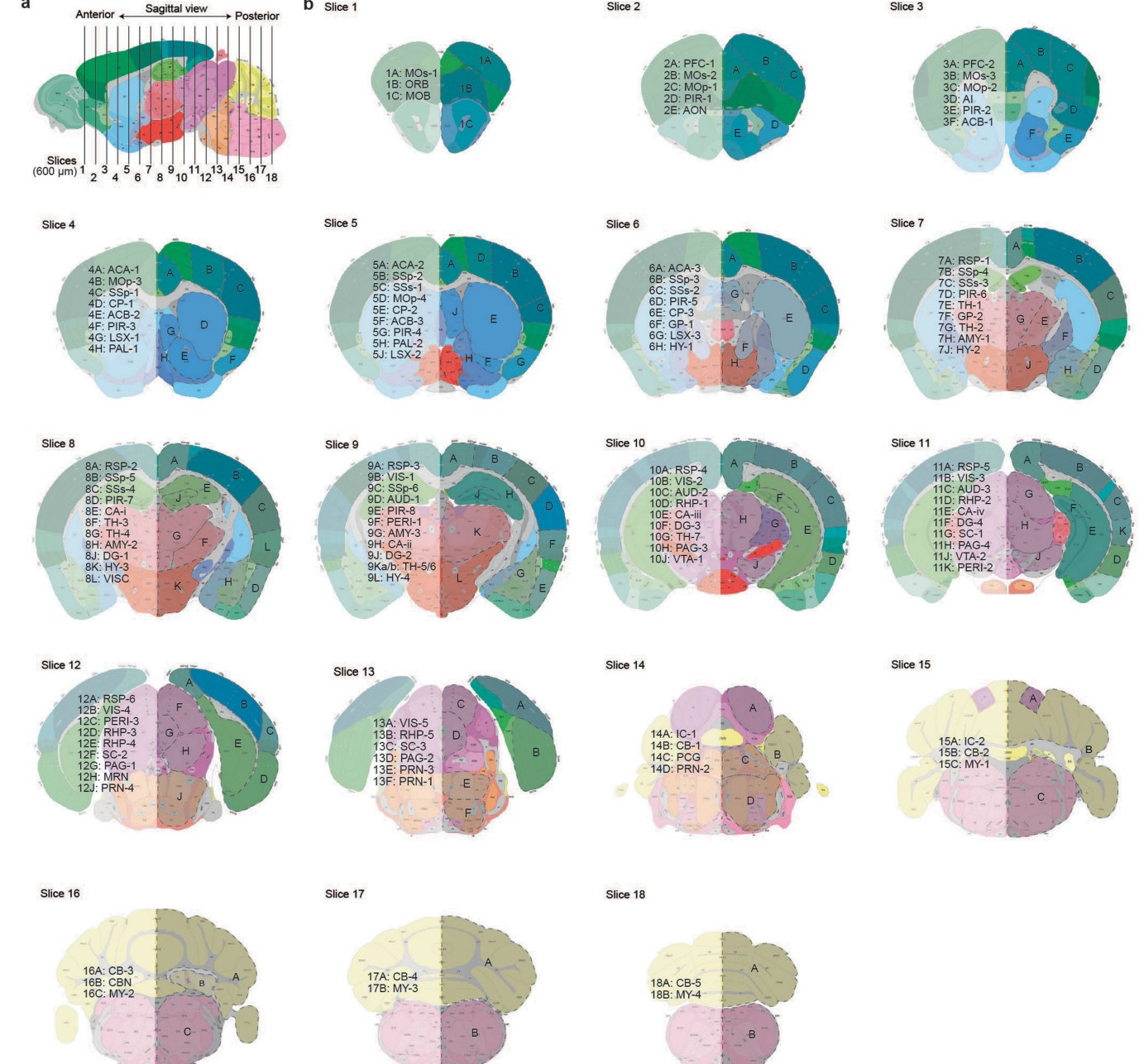

**Extended Data Fig. 1 | Maps of the 117 anatomical dissections of the adult whole mouse brain. a**, Schematic of brain tissue dissection strategy. Mouse brains were cut into 600-µm-thick coronal slices. **b**, These brain maps were generated using coordinates from the Allen Mouse Brain Common Coordinate Framework (CCF) v3 (ref. 26). Brain regions dissected from each coronal slice are marked according to the Allen Brain Reference Atlas[26]. The frontal view of each slice from slices 1–18 is shown, with the dissected regions alphabetically labelled on the left, and the anatomic labelling listed on the right. A detailed list of the dissected regions and the full anatomic labelling can be found in Supplementary Table 1.

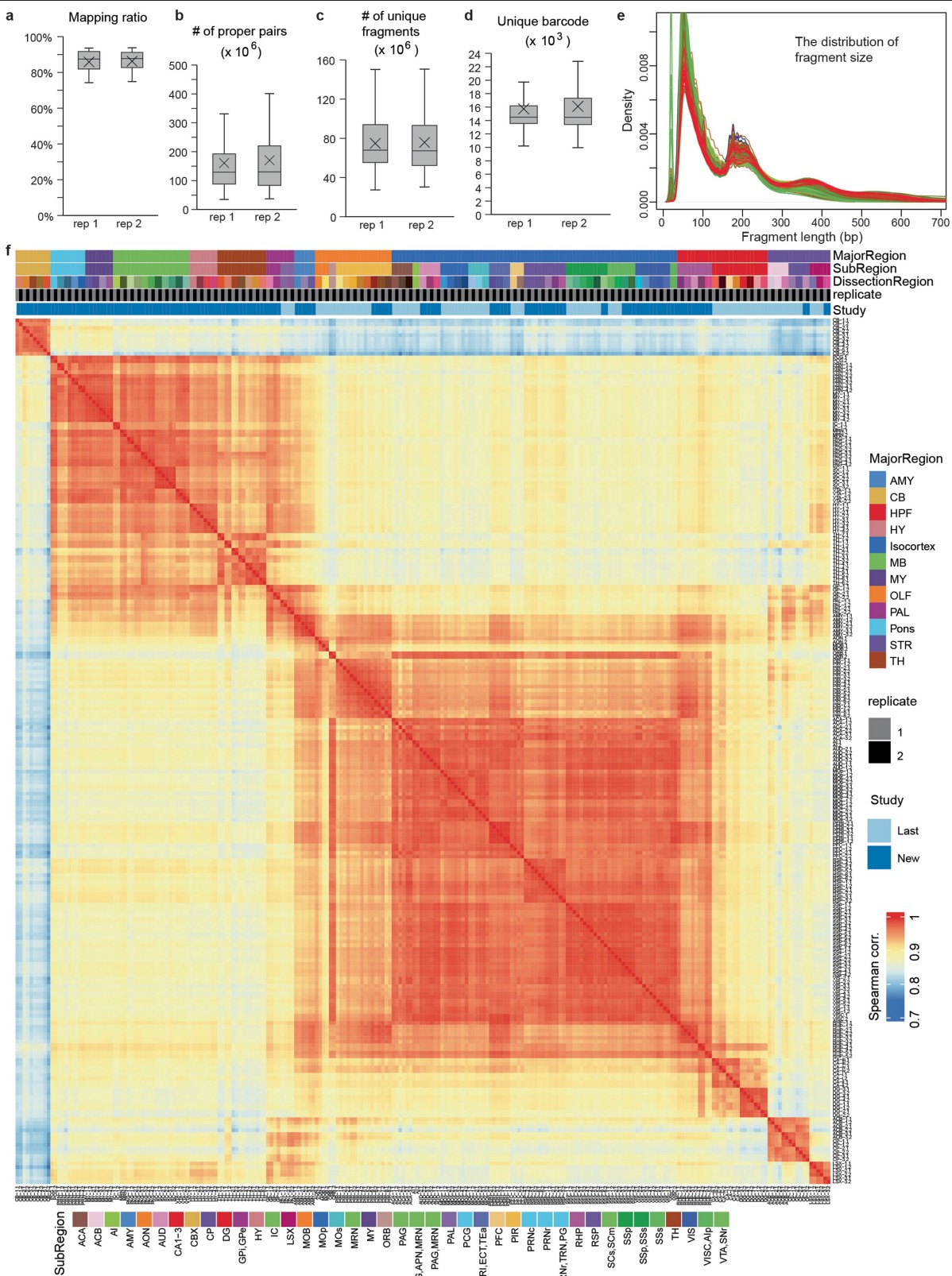

**Extended Data Fig. 2 | Quality control metrics of the snATAC-seq datasets at the bulk level. a**, Box plots showing the distribution of mapping ratios (the fraction of the mapped sequencing reads) in replicates (rep) 1 and 2 of the snATAC-seq experiments from each brain dissection. **b**, Box plots showing the distribution of the number of proper read pairs (reads are correctly oriented) in rep 1 and 2 of the snATAC-seq experiments. **c**, Box plots showing the distribution of numbers of unique chromatin fragments detected in rep 1 and 2 of the snATAC-seq experiments. **d**, Box plots showing the distribution of the number of unique barcodes captured in replicates 1 and 2 of snATAC-seq experiments. In **a-d**, the number per each boxplot (rep1 or rep2) is 117. In each boxplot, the box spans the first to third quartiles, the horizontal line denotes the median, and whiskers show 1.5x the interquartile range. **e**, Frequency distribution plot showing the fragment size distribution of each snATAC-seq sample or datasets (234 samples/datasets in total). **f**, Heat map showing the pairwise Spearman correlation coefficients of the mapping correlations of the bam files between the snATAC-seq datasets. The column and row names consist of two parts: brain region name and replicate label. Study represents dissections covered by our previous study (Last) or updated in the current study (New).

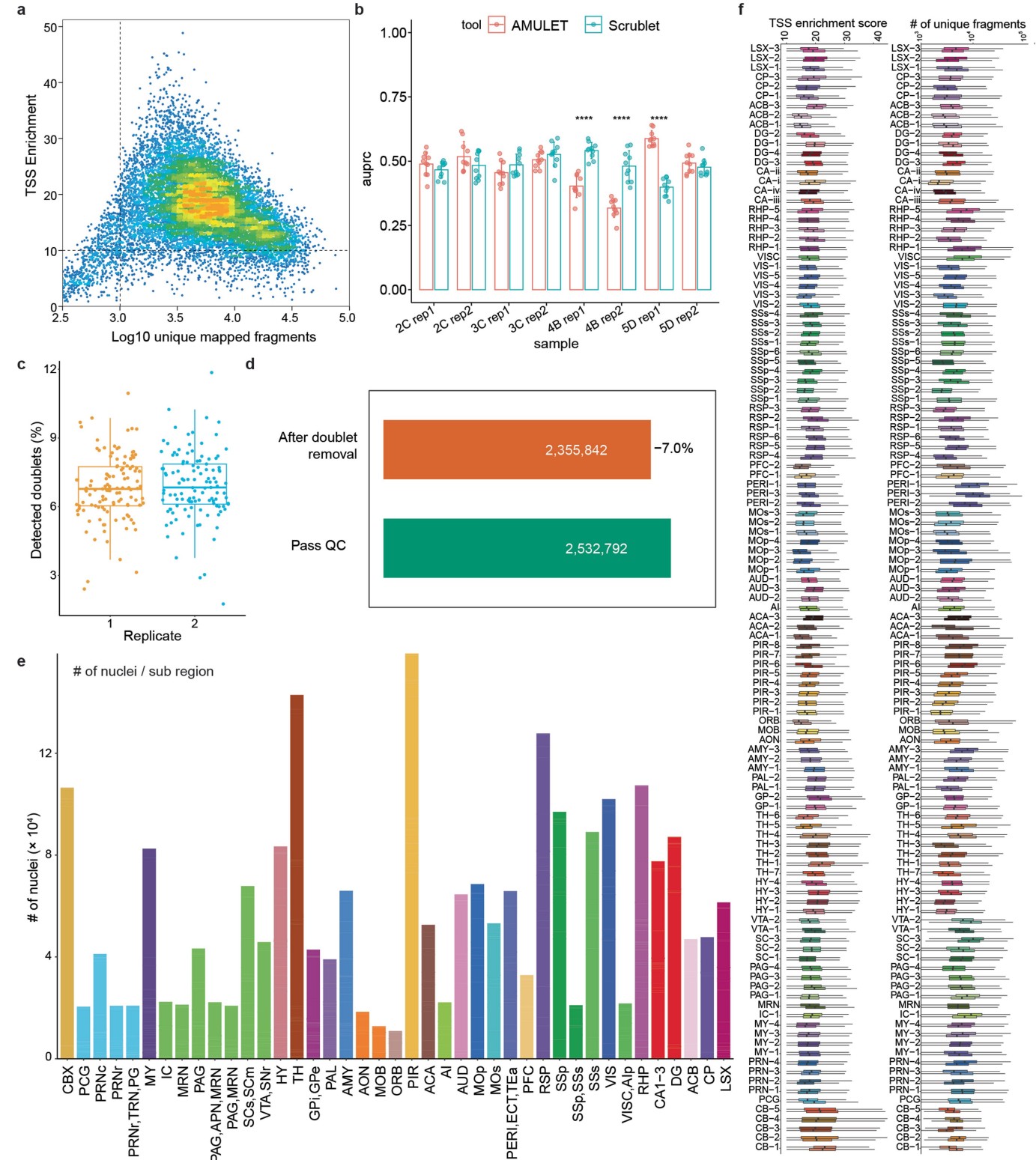

**Extended Data Fig. 3 | Quality control metrics of the snATAC-seq datasets at the single-cell level. a**, Dot plot illustrating fragments per nucleus and individual TSS enrichment. Nuclei in the top right quadrant were selected for analysis (TSS enrichment > 10 and > 1,000 fragments per nucleus). **b**, Box plots showing the AUPRCs of AMULET[30] and Scrublet[28] on the simulated data sets from the corresponding samples labelled in x axis. Each bar represents the mean value of 10 random experiments with 1x standard deviation as the error bar. Two-sided t-tests were used, and *** means P-value < 0.0001. **c**, Box plots showing the doublet rates across the samples. Samples were grouped based on their replicate information. n = 117 biologically independent samples for each replicate 1 and 2. **d**, Number of nuclei retained after each step of quality control. **e**, Bar plots showing the numbers of nuclei passing quality control for subregions. **f**, Box plots showing the TSS enrichments and unique fragments per nuclei for the replicates in different mouse brain regions. The smallest sample size is ORB region replicate 1 with n = 4,943 cells, while the largest is PAL-2 replicate 1 with n = 12,464 cells. In **c** and **f**, boxes span the first to third quartiles, horizontal line denotes the median, and whiskers show 1.5x the interquartile range.

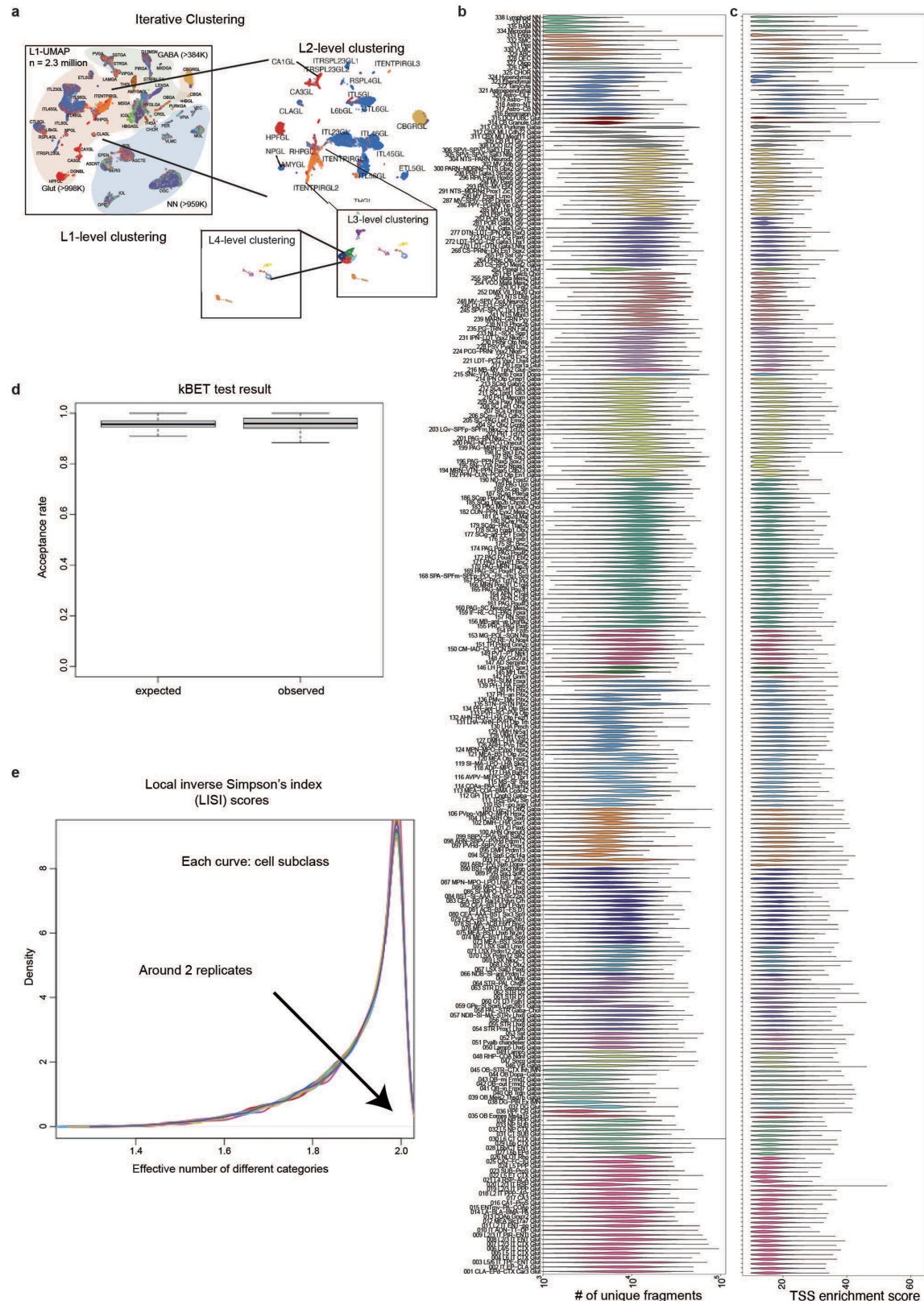

**Extended Data Fig. 4 | Iterative clustering for the snATAC-seq data.**
**a**, A multi-stage cell clustering pipeline is organized for all the nuclei passing our quality control. **b**, Violin plots showing the number of unique fragments per nucleus in each cell subclass. **c**, Violin plots showing the TSS enrichment in each nucleus of each cell subclass. **d**, Boxplots of acceptance rates from k-nearest neighbour batch effect test[34] (kBET) for the 275 subclasses. Boxes span the first to third quartiles, horizontal line denotes the median, and whiskers show 1.5× the interquartile range. Two-sided t-tests showed no significant P-values between the values from the two boxes. **e**, Distribution of the local inverse Simpson's index[35] (LISI) scores for cells in each subclass.

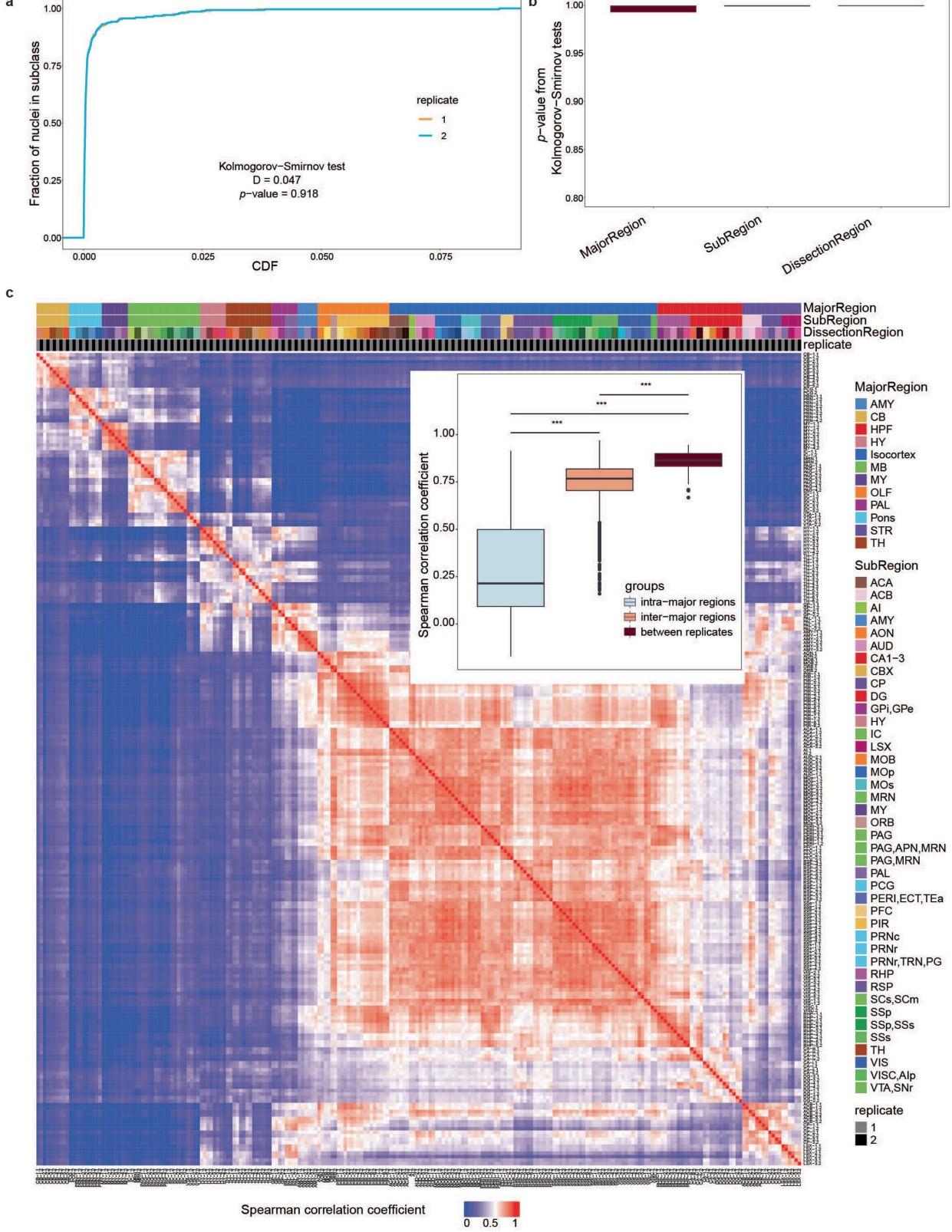

**Extended Data Fig. 5** | See next page for caption.

**Extended Data Fig. 5 | Quality and reproducibility of the cell clusters. a**, CDF plot showing the consistency of the estimated fraction of each cell subclass between the biological replicates. Two-sided Kolmogorov-Smirnov test shows no significant difference between the biological replicates. **b**, Box plots of the *P* values of two-sided Kolmogorov-Smirnov tests illustrate consistent results between the two biological replicates for each subclass across major brain regions, sub-regions and brain dissections tested. n = 12 comparisons for major regions, n = 41 comparisons for sub-regions and n = 117 comparisons for dissection regions. **c**, Heat map showing the pairwise Spearman correlation coefficients of cell subclass composition between each replicate of brain dissections. The column and row names consist of two parts: brain region name and replicate label. For example, CB-1.1 represents the replicate 1 of the first brain dissection of the cerebellum (CB-1). The embedded box plot shows the distribution of Spearman correlation coefficients between two biological replicates, replicates from intra-major brain regions and inter-major brain regions. Significance is denoted as ***P < 2.2e-16, determined by one-sided Wilcoxon rank-sum test. n = 22720 pairs for "intra-major regions" group, n = 4424 pairs for "inter-major regions" group, n = 117 for "between replicates" group. Boxes span the first to third quartiles, horizontal line denotes the median, and whiskers show 1.5x the interquartile range.

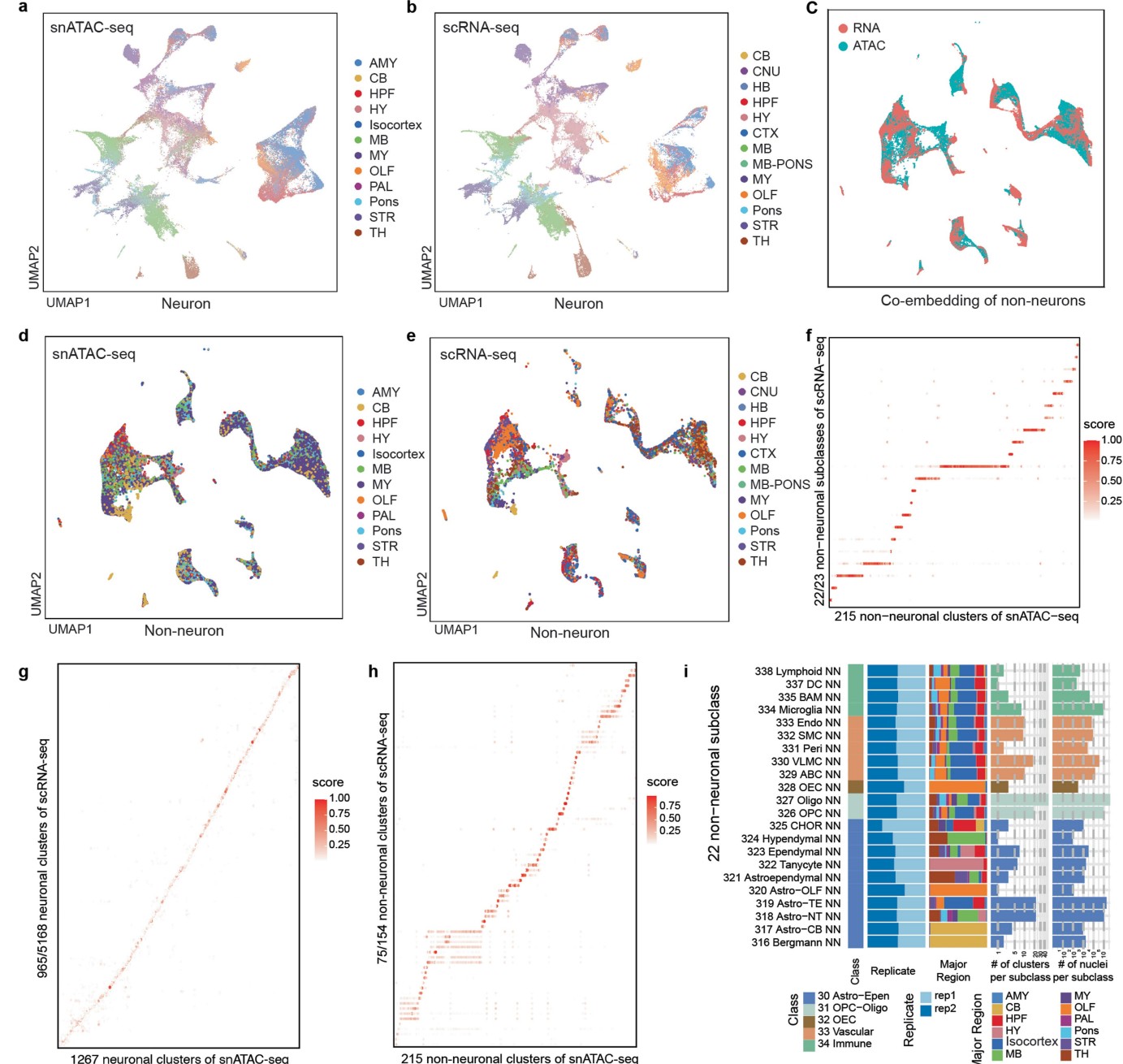

**Extended Data Fig. 6 | Integration analysis between the snATAC-seq and the scRNA-seq data for neurons and non-neurons separately.** UMAP on the co-embedding space of neurons from the snATAC-seq data (**a**) and scRNA-seq data (**b**). Colours as major regions. **c**, The co-embedding UMAP embedding of non-neuronal cells from the scRNA-seq data and the snATAC-seq data on the same space coloured by the two modalities. UMAP on the co-embedding space of non-neurons from snATAC-seq data (**d**) and scRNA-seq data (**e**). Colours as major regions. **f**, Consensus scores (i.e., transfer-label scores) between non-neuronal subclasses from the scRNA-seq data and L4-level non-neuronal

clusters from the snATAC-seq data. **g**, Consensus scores between neuronal clusters from the scRNA-seq data of Allen Institute and L4-level neuronal clusters from the snATAC-seq data. **h**, Consensus score between non-neuronal clusters from the scRNA-seq data and L4-level non-neuronal clusters from the snATAC-seq data. **i**, The 22 non-neuronal subclasses matched to the non-neuronal subclasses in the scRNA-seq. From left to right, the bar plots represent class, biological replicate distribution of nuclei, major region distribution of nuclei, number of clusters, and number of nuclei.

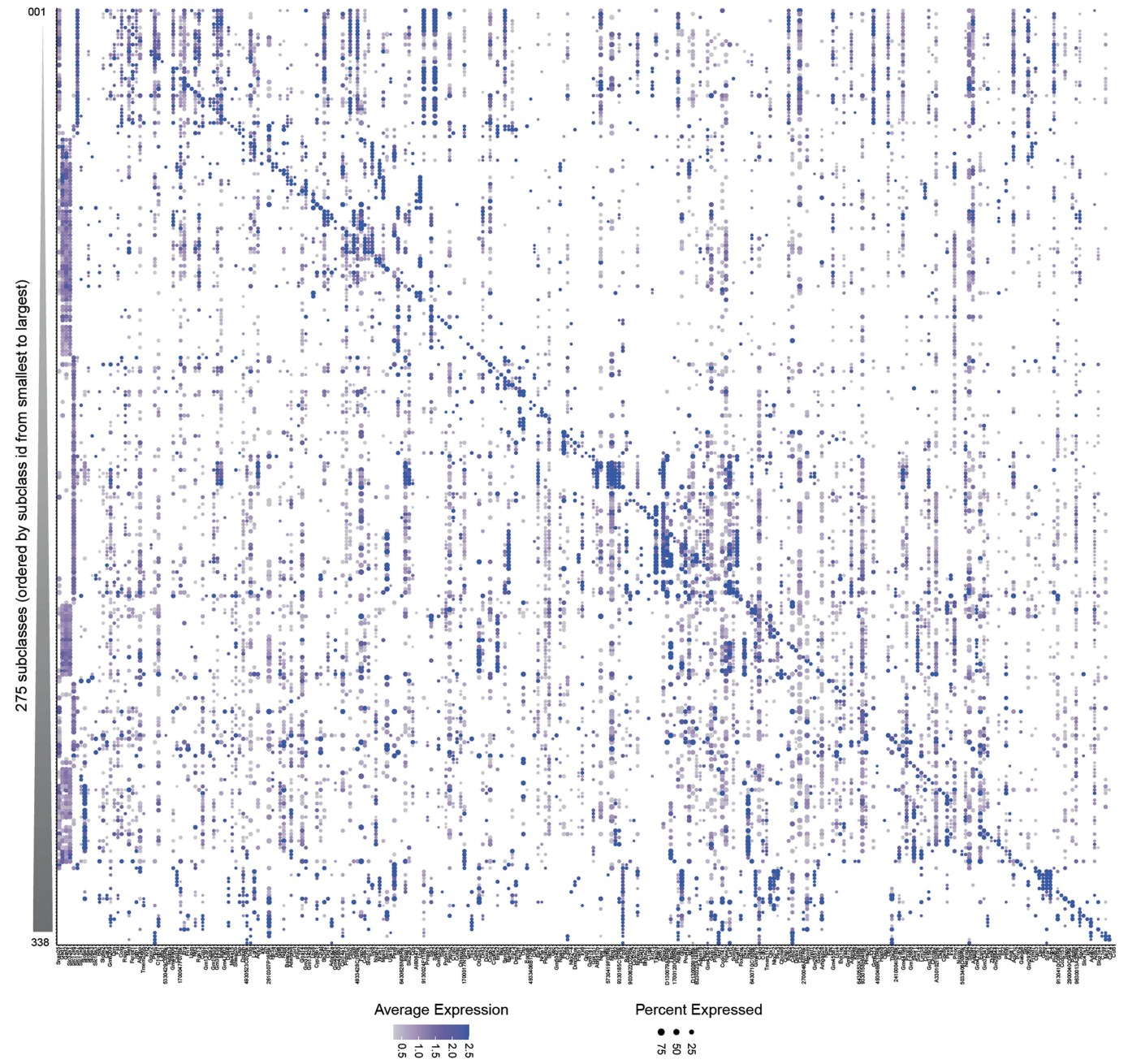

Average Expression
0.5 1.0 1.5 2.0 2.5

Percent Expressed
75 50 25

**Extended Data Fig. 7 | Marker genes for the subclasses after integration in the snATAC-seq data using the imputed gene expressions.** Dotplot showing the snATAC-seq gene activity scores of the marker genes (columns) used for identification of the scRNA-seq data across the cell subclasses[5]. The first 13 columns correspond to major neuronal cell type marker genes including neurotransmitter genes as follows: *Snap25* (Neuron), *Gad1* (GABA), *Gad2* (GABA), *Slc32a1* (GABA), *Slc17a6* (Glut-subcortical), *Slc17a7* (Glut-cortical), *Slc17a8* (Glut), *Slc6a5* (Gly-GABA), *Slc6a4* (Glut-Sero), *Slc6a3* (Dopa), *Slc18a3* (Chol), *Hdc* (Hist), *Slc6a2* (Nora). The subsequent columns are the most occurring marker gene reported within each Allen Institute subclass designation corresponding to each subclass annotation (row) of the snATAC-seq data.

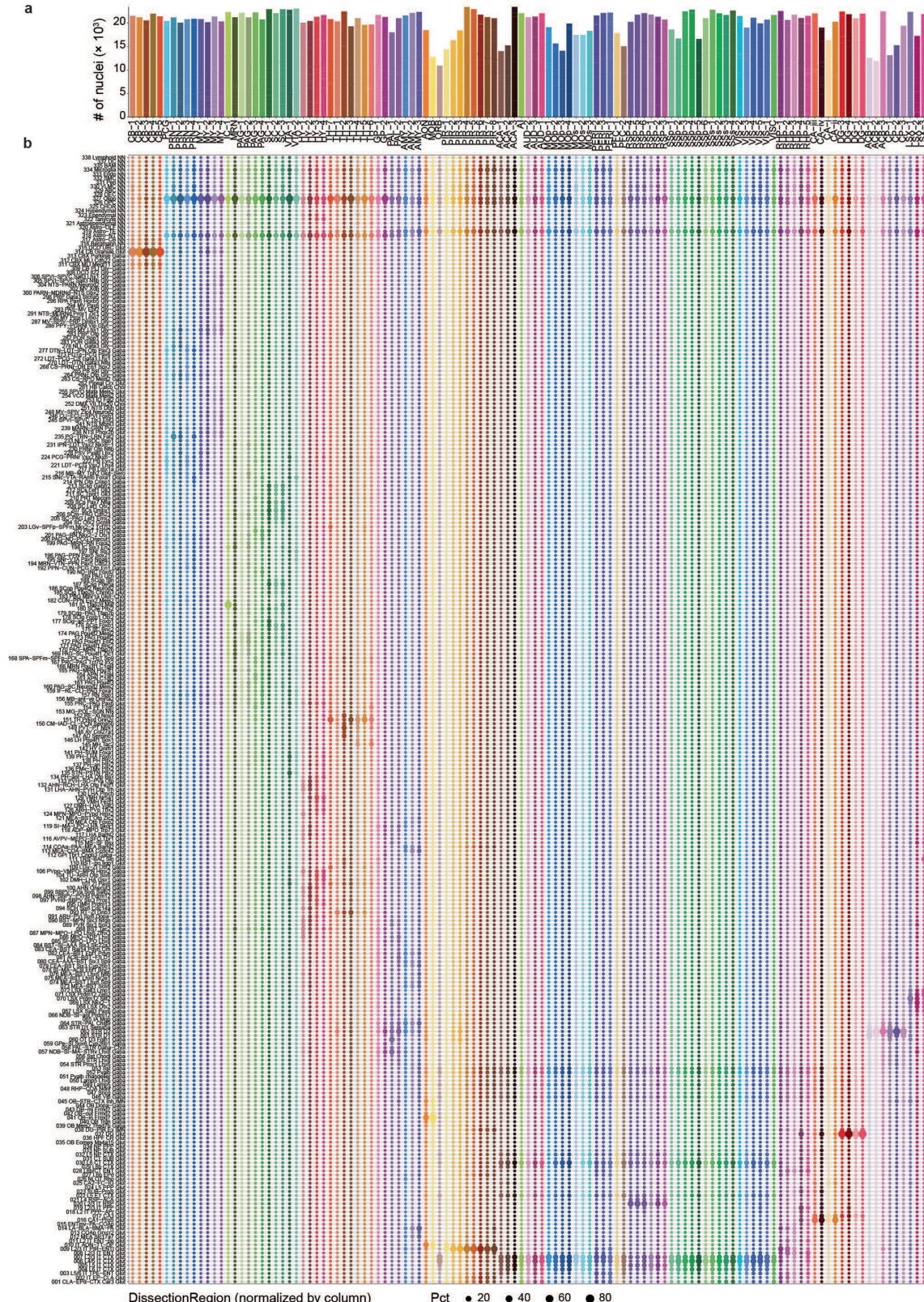

**Extended Data Fig. 8 | Cellular composition of brain dissections for cell subclasses. a**, Bar plot shows the total number of nuclei sampled for each brain dissection region. **b**, Normalized percentages (pct) of each subclass in all the dissected regions are shown as different sized dots. The sizes of dots correspond to the percentage and the colours of the dots indicate the brain dissections.

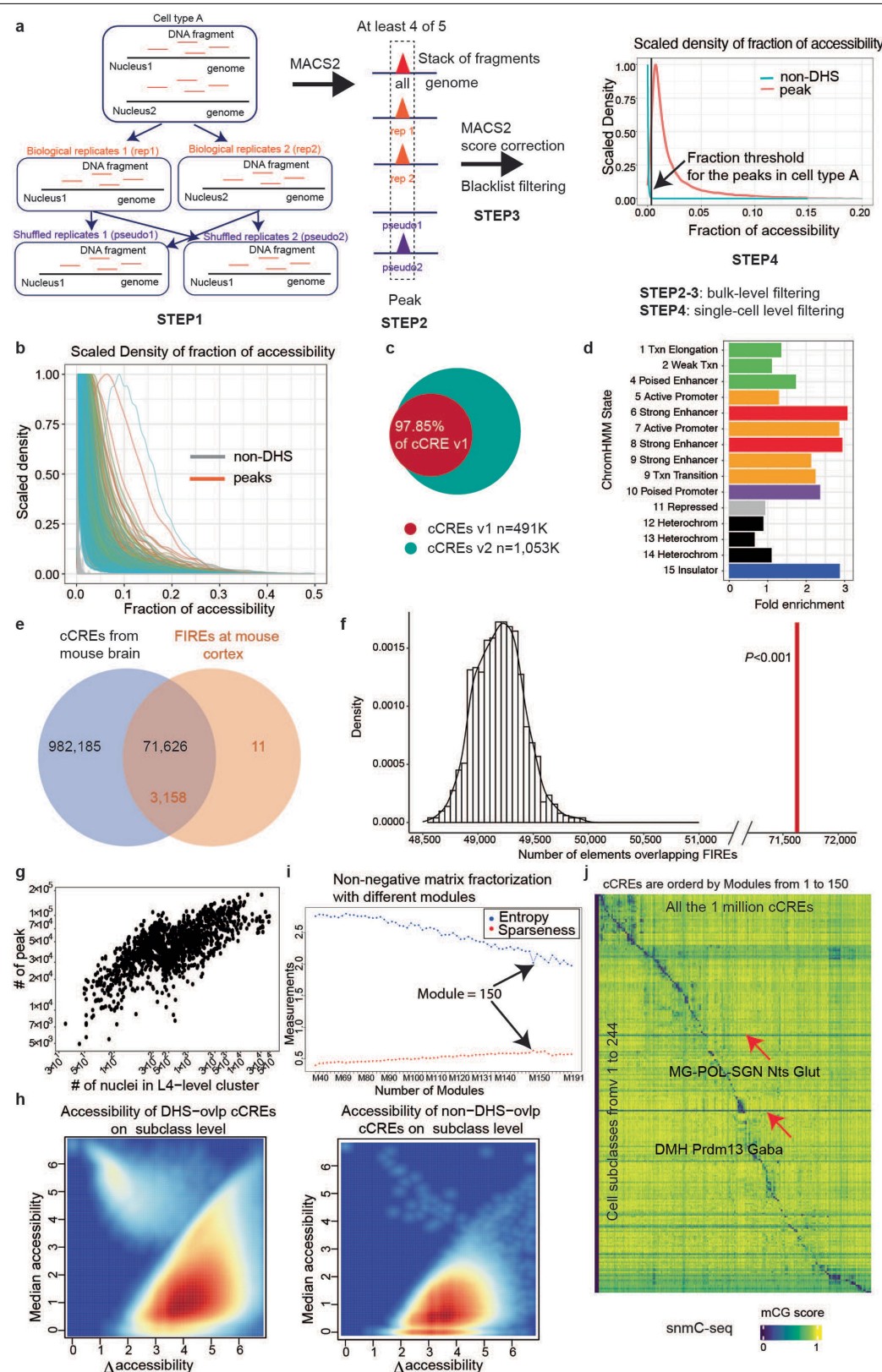

**Extended Data Fig. 9** | See next page for caption.

**Extended Data Fig. 9 | Statistics of peak calling on snATAC-seq data for each cell subtype. a**, Schematic of peak calling and filtering pipeline. **b**, Density distribution plot showing the fraction of cells per cell type in which a peak was accessible and a corresponding background for each cell type. For each cell type, the background is defined as the non-DHS and non-peak regions randomly picked from the genome. **c**, Venn plot showing the overlapping between the peaks from the whole mouse brain and the ones from the cerebral regions[19]. **d**, Enrichment analysis of the peak sets with a 15-state ChromHMM model in the mouse brain chromatin[102]. **e**, Density map comparing the median and maximum variation of chromatin accessibility at each cCRE across cell subclasses. The left density map refers to the cCREs overlapping with the ENCODE DHSs, and the right one refers to the cCREs having no overlaps with the ENCODE DHSs. **i**, Scatter plot showing entropy (blue) and sparseness (red) trends when increasing the number of modules used for non-negative matrix factorization. When the module number is 150, we can see a significant drop in entropy and a significant increase in sparseness. **j**, The red arrows point to the two subclasses with lowest number of cells in the snmC-seq data[41].

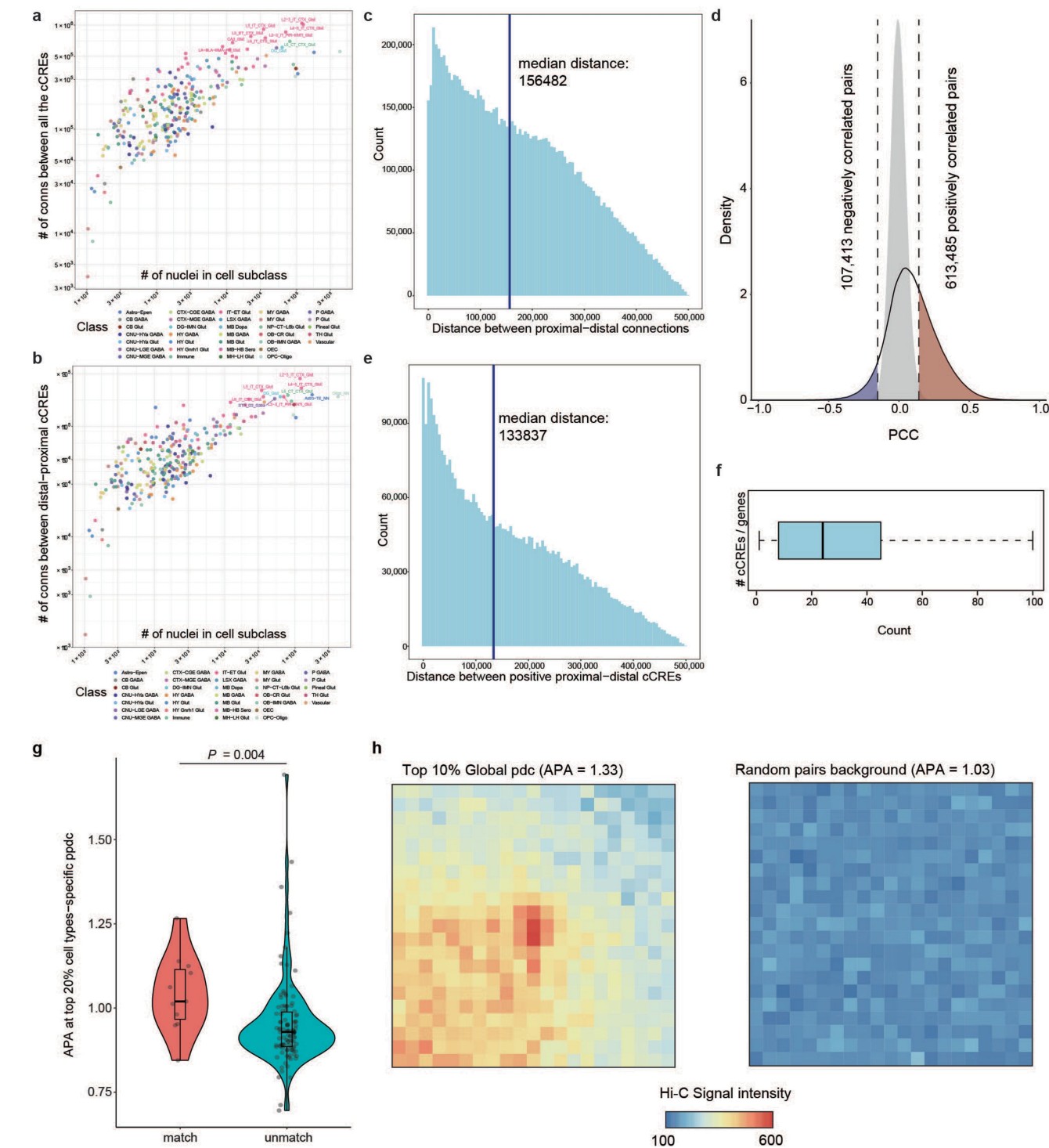

**Extended Data Fig. 10 | Characterization of predicted cCRE-target gene pairs. a**, Scatter plot showing the number of identified connections between all the cCREs pairs within 500k bp along with the number of nuclei for each cell subclass identified based on the integration analysis. **b**, Scatter plot showing the number of proximal-distal cCREs along with the number of nuclei for each cell subclass. **c**, Histogram showing the distances along the genome for each proximal-distal cCREs. **d**, Histogram showing the distances along the genome for each pair of enhancer and targeted gene's promoter (positive proximal-distal cCREs) inferred by the correlation study (Fig. 3b). **e**, In total, 613,485 positively correlated proximal-distal cCREs and 107,413 negatively correlated proximal-distal cCREs were identified. **f**, Boxplot showing the identified potential

enhancers for each of 20,703 gene in the positively correlated pairs. **g**, Boxplots of the enrichment scores (1 kb resolution) of aggregate peak analysis (APA) for the top 20% positive proximal-distal connections (ppdc) from several represented subclasses. Match, the subclass's Hi-C data[41] used for the same subclasses. Unmatch, the subclass's Hi-C data used for other subclasses as a random background. 11 data points were included in the match group and 110 points in the unmatched groups. P value was calculated by the one-sided Wilcoxon rank sum test. In **f** and **g**, boxes span the first to third quartiles, horizontal line denotes the median, and whiskers show 1.5x (**f**) and 2x (**g**) the interquartile ranges. **h**. Heatmaps of enrichment signals for the top 10% global proximal-distal connections (pdc) and enrichment signals for the random pairs.

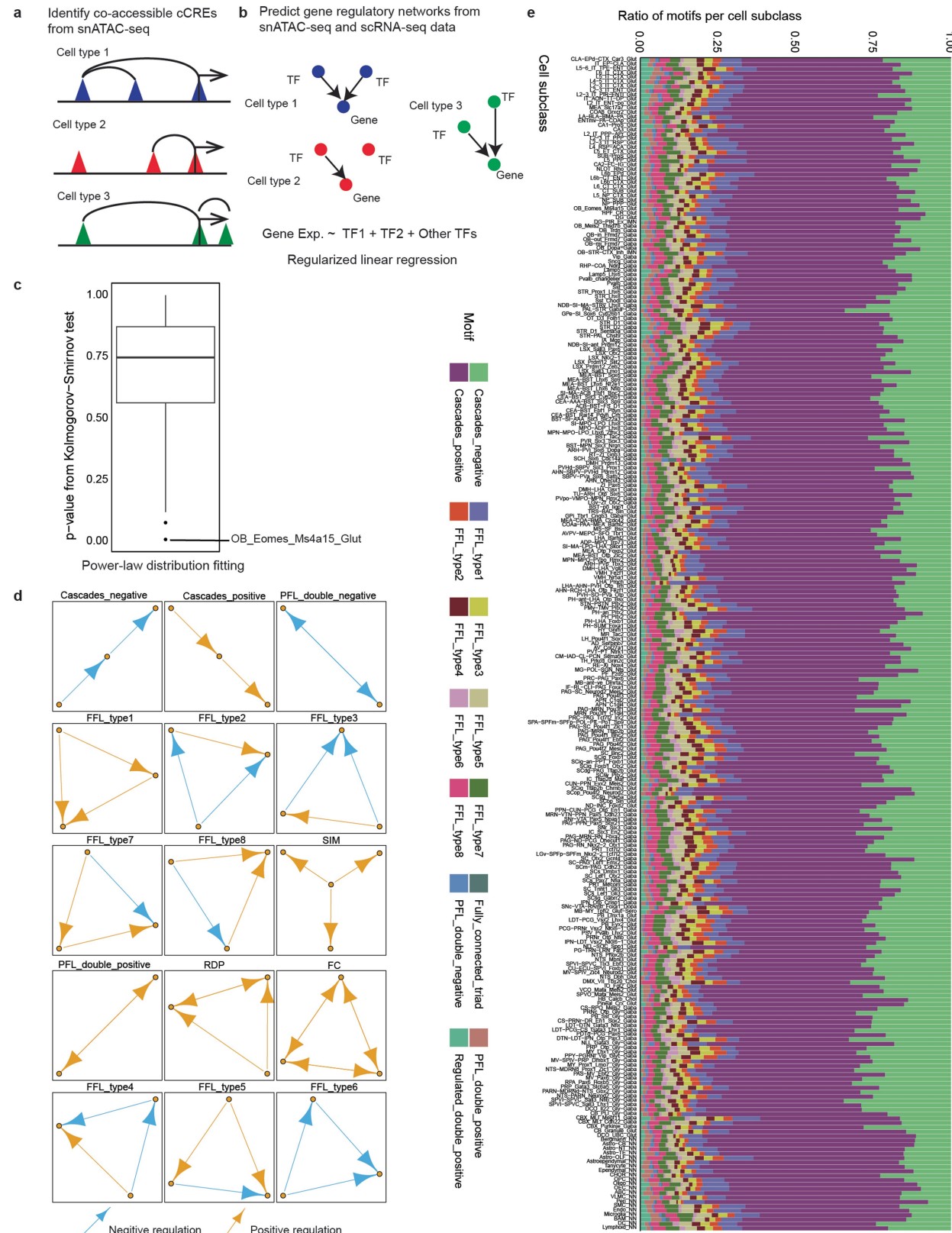

**Extended Data Fig. 11 | Inference of gene regulatory networks (GRNs) at cell subclass level across the whole mouse brain. a**, Schematic of identifying co-accessible cCREs for each cell subclass using Cicero[44]. **b**, Schematic view of inference of GRNs from predicting the putative target genes' expression with the corresponding transcription factors (TFs) for each cell subclass using CellOracle[52]. **c**, Boxplot of 267 *P* values from two-sided Kolmogorov-Smirnov test to check power-law distributions of the nodes' degrees from GRNs. Only one cell subclass (OB_Eomes_Ms4a15_Glut) did not pass this examination with the *P* values smaller than 0.05. The box spans the first to third quartiles, the horizontal line denotes

the median, and whiskers show 1.5x the interquartile range. **d**, 15 commonly used network motifs[56] used in our analysis. Each node is a TF or a gene, and edges describe the regulation directions, i.e., arrows pointed to the ones that were regulated by the source nodes or TFs. The blue colour means the negative regulation (TFs inhibit target gene expressions), while the orange colour means the positive regulation (TFs upregulate target gene expressions). PFL, positive-feedback loops; RDP, regulated double-positive; FC, fully connected triad; FFL, feedforward loops. SIM, single-input module. **e**, Stacked bar plots of the ratio of the network motifs above in each subclass. Each column responds to one cell subclass.

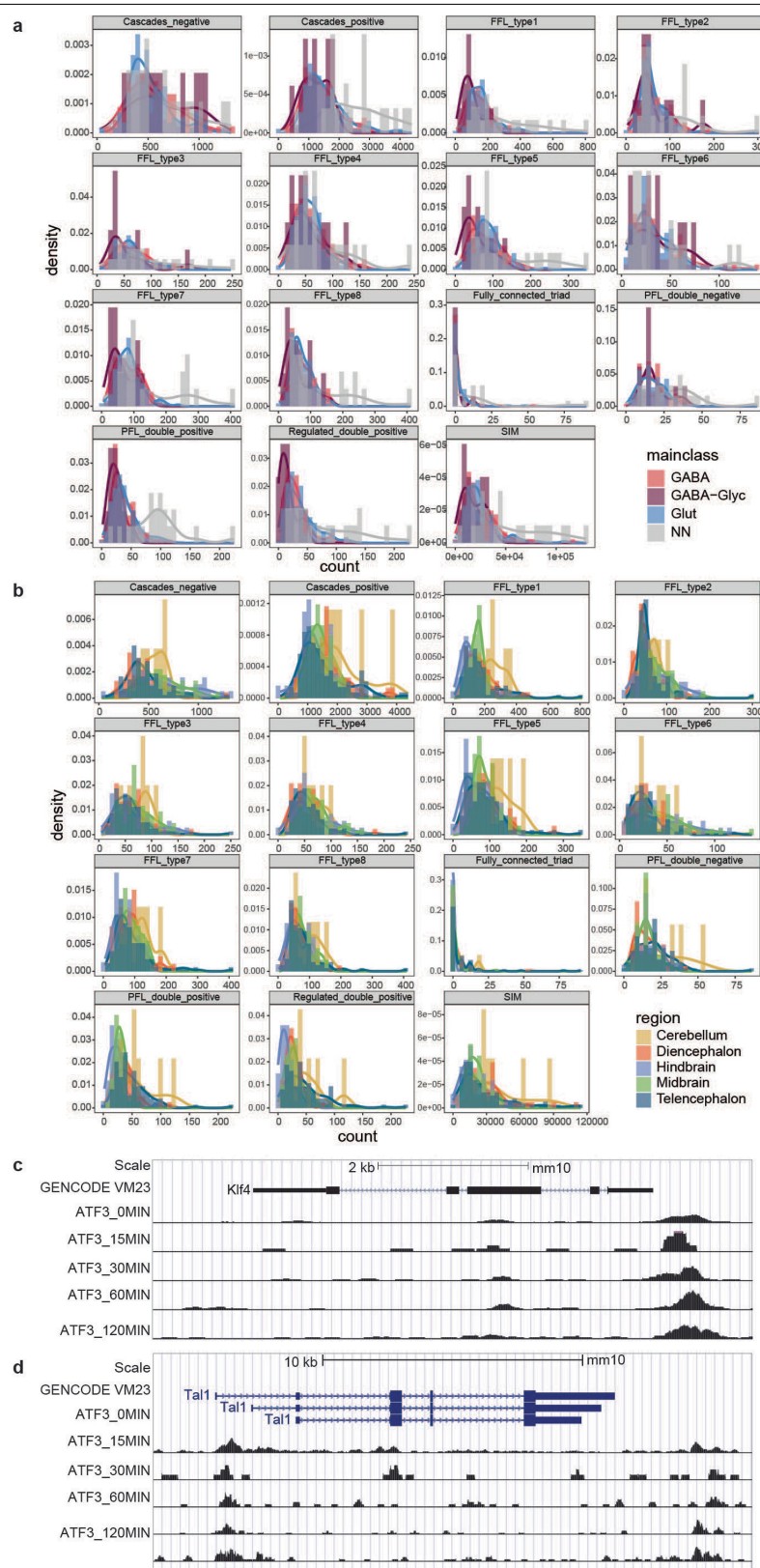

**Extended Data Fig. 12 | Histograms of the counts of the network motifs in each subclass's gene regulation network (GRN) grouped by main class (a) or regions (b).** The names of the network motifs are the same ones in Extended Data Fig. 11d. Only the class with at least 3 subclasses were shown here. For each histogram, we added the corresponding density plot. The telencephalon region includes isocortex, olfactory bulb, hippocampus, striatum, pallidum, and amygdala; the diencephalon region includes thalamus and hypothalamus; the hindbrain includes pons and medulla. **c**, Normalized signals of Atf3 ChIP-seq at *Klf4* in bone marrow-derived macrophages (BMM) showing *Klf4* is likely to be a putative target of Atf3. **d**, Normalized signals of Atf3 ChIP-seq at *Tal1* in bone marrow-derived macrophages (BMM) showing *Tal1* is likely to be a putative target of Atf3.

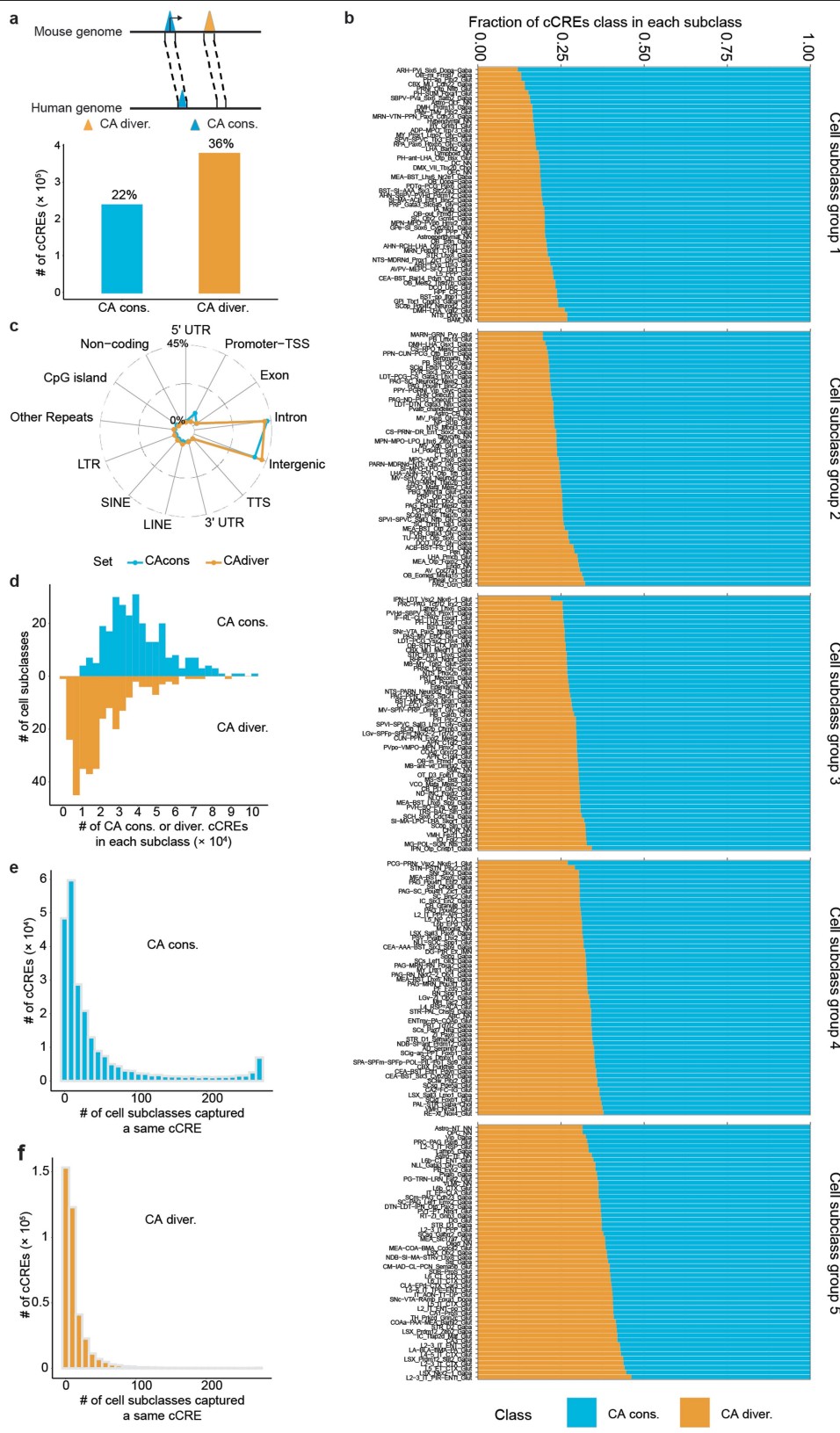

**Extended Data Fig. 13 | Comparison of chromatin accessibility (CA) conserved and divergent cCREs between mouse and human. a**, A schematic of CA conserved and divergent cCREs. The CA-conserved cCREs are the cCREs in our snATAC-seq data that are conserved across species and have open chromatin in orthologous regions. The CA divergent cCREs are sequence conserved to orthologous regions but have not been identified as open chromatin regions in other species. The bar plot shows the numbers of CA-conserved and CA-divergent cCREs. **b**, Bar plot showing the relative fraction of CA conserved and divergent cCREs across subclasses. **c**, Radar chart showing the fraction of genomic distribution of CA-conserved and CA-divergent cCREs. The CA-conserved cCREs show an increase in percentage in Promoter-TSS regions. **d**, Histograms showing the number of CA-conserved and CA-divergent cCREs in subclasses. The number of CA-conserved cCREs is higher than CA-divergent cCREs. **e**, Histograms showing the CA-conserved cCREs captured by the number of cell subclasses. A fraction of CA-conserved cCREs are captured by more than 200 cell subclasses. **f**, Histograms showing the CA-divergent cCREs captured by the number of cell subclasses. Most CA-divergent cCREs are captured by less than 50 cell subclasses.

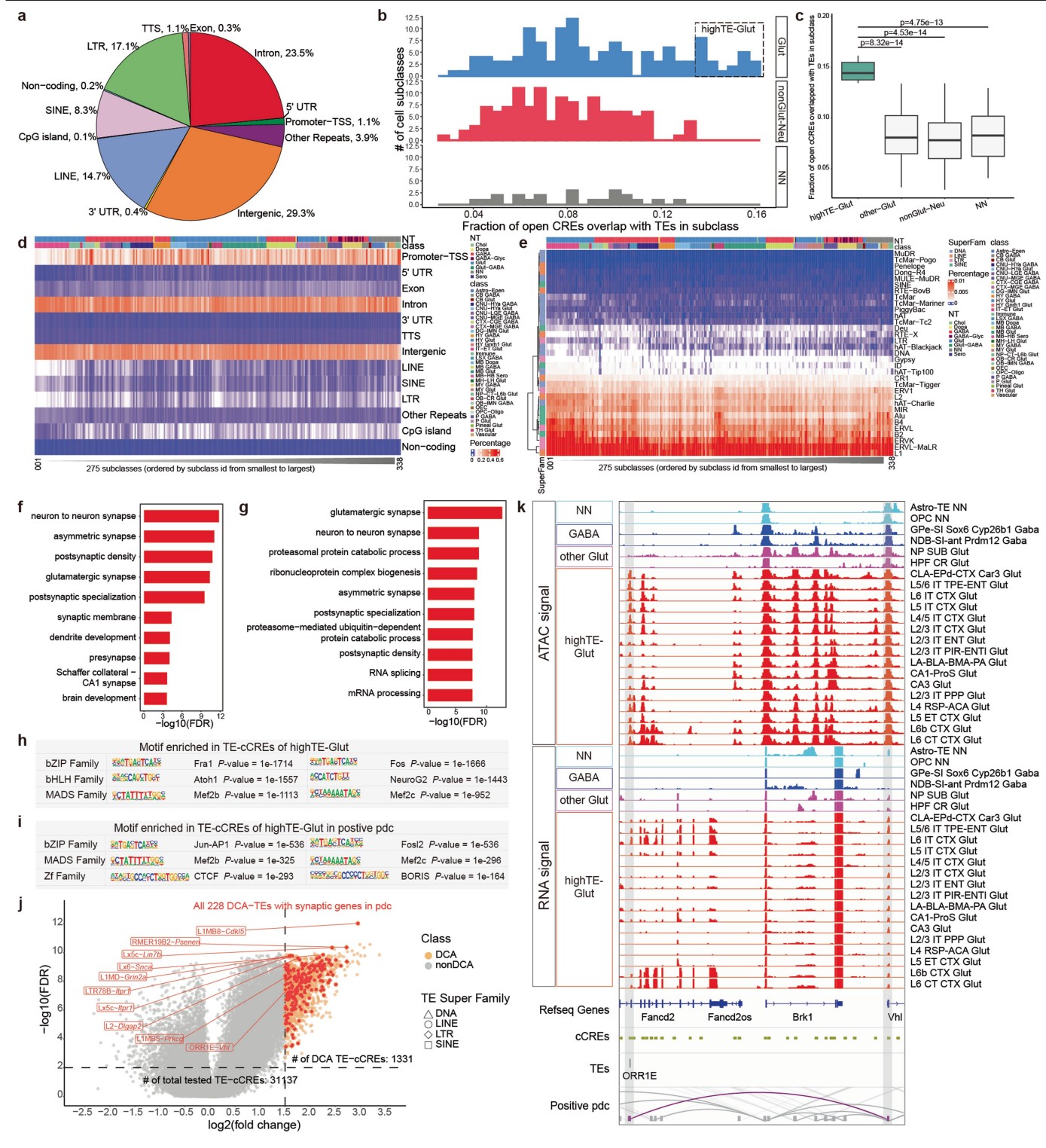

**Extended Data Fig. 14** | See next page for caption.

**Extended Data Fig. 14 | Analyses of chromatin accessibility at transposon elements (TEs) of cCREs. a**, Pie charts showing the genomic distribution of mouse-specific cCREs. **b**, Histograms showing the fraction of cCREs overlap with TEs in subclasses of glutamatergic neurons (Glut), non-glutamatergic neurons (nonGlut-Neu), and non-neurons (NN). **c**, Boxplot showing the fraction of cCREs overlap with TEs in highTE-Glut, other-Glut, nonGlut-Neu, and NN subclasses. The *P* values are calculated by the one-sided Wilcoxon rank-sum test. Boxes span the first to third quartiles, horizontal line denotes the median, and whiskers show 1.5× the interquartile range. There are n = 22 subclasses in the "highTE-Glut" group, n = 108 subclasses in the "other-Glut" group, n = 123 subclasses in the "nonGlut-Neu" group, and n = 22 subclasses in the "NN" group. **d**, Heatmap showing the fraction of genomic distribution of cCREs in each cell subclass. **e**, Heatmap showing the fraction of TE family distribution of cCREs in each cell subclass. **f**, GO analysis showing genes near TE-cCREs in highTE-Glut versus genes near TE-cCREs in all subclasses are enriched for neuronal specific functions. **g**, GO analysis showing genes near TE-cCREs in highTE-Glut versus genes near all cCREs in highTE-Glut are enriched for neuronal specific functions. **h**, Top3 motif families enriched in the TE-cCREs in highTE-Glut. The unadjusted P-values were calculated using a two-sided Fisher's exact test. **i**, Top3 motif families enriched in the TE-cCREs which showed positively correlated with genes and occurred in highTE-Glut. The unadjusted P-values were calculated using a two-sided Fisher's exact test. **j**, Volcano plot showing differential chromatin accessibility (DCA) TE-cCREs in highTE-Glut subclasses compared to other subclasses. The red colour labelled all DCA TE-cCREs which correlated with synaptic related genes. **k**, Genome browser tracks of aggregate chromatin accessibility profiles for NN, GABA, highTE-Glut, and other Glut subclasses at selected DCA TE-cCREs and gene pairs. RNA signals shown here were collected from previous study[99].

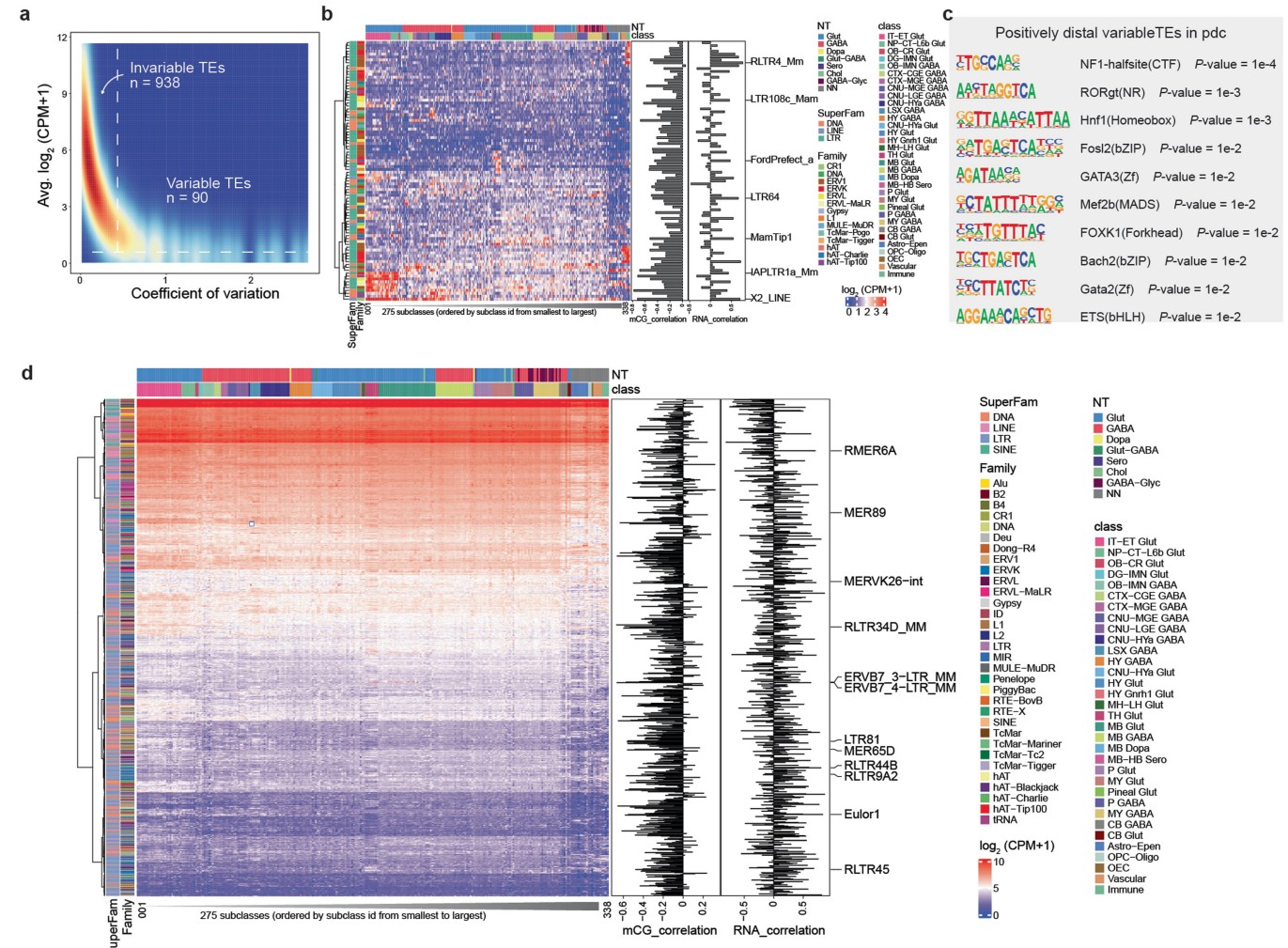

**Extended Data Fig. 15 | Accessible variability at transposon elements (TEs) across cell subclasses. a**, Density scatter plot comparing the averaged accessibility and coefficient of variation across cell subclasses at each transposon element. Variable TEs are defined on the upper right side of dash lines, invariable TEs are defined on the upper left of dash lines. **b**, Normalized accessibility at variable TEs in different cell subclasses. The middle bar plot showing correlation between mCG level and accessibility at variable TEs across subclasses. The right bar plot shows correlation between expression level and accessibility at variable TEs across subclasses. **c**, Top 10 motifs enrich in positively distal cCREs overlapped with variable TEs. The unadjusted P-values were calculated using a two-sided Fisher's exact test. **d**, Normalized accessibility at invariable TEs in different cell subclasses. The middle bar plot showing correlation between mCG level and accessibility at invariable TEs across subclasses. The right bar plot showing correlation between expression level and accessibility at invariable TEs across subclasses.

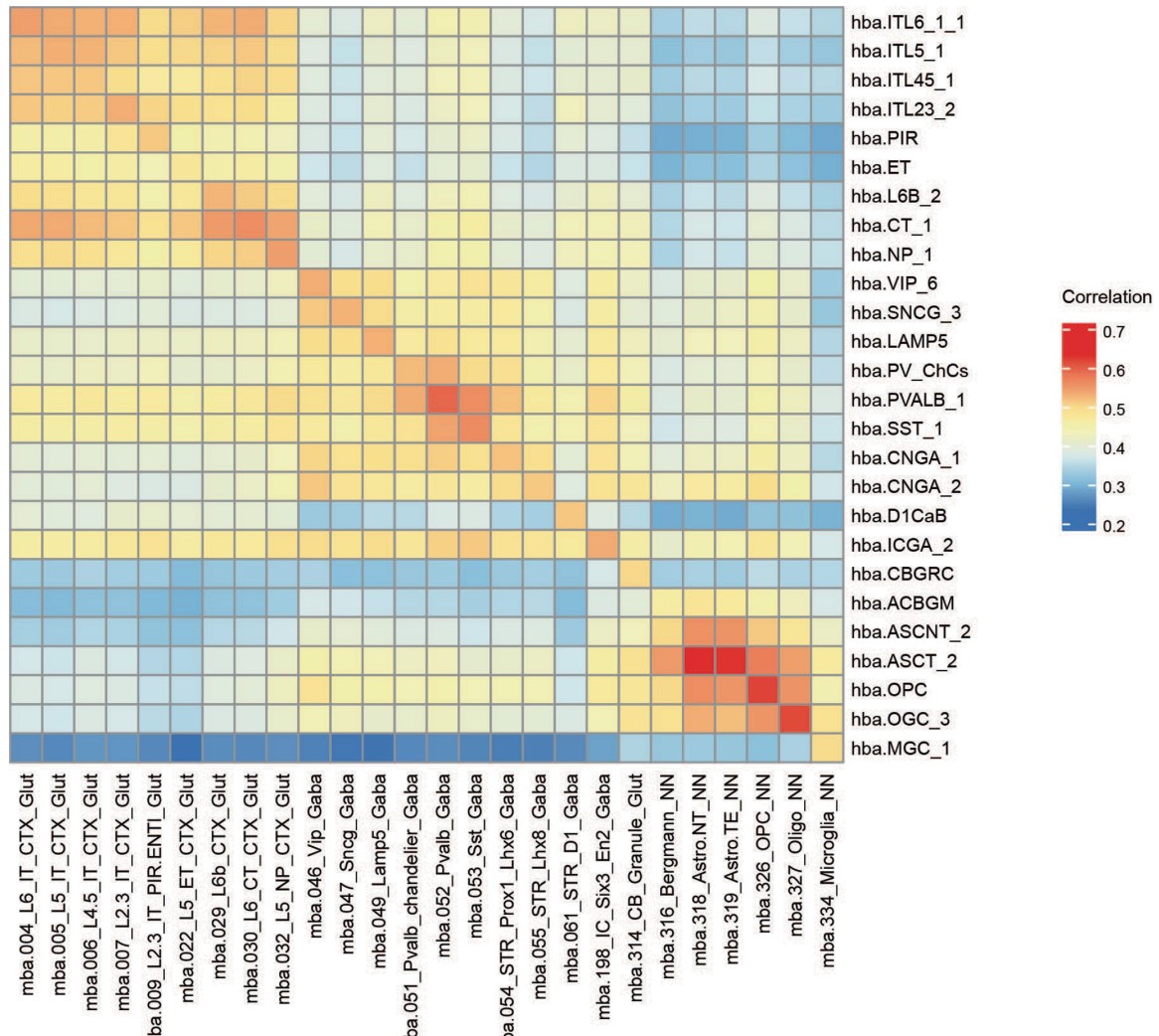

**Extended Data Fig. 16 | Spearman correlation across orthologous cCREs between all paired human and mouse subclasses (mba: mouse brain atlas; hba: human brain atlas).**

| | |
|---|---|

# Reporting Summary

## Statistics

For all statistical analyses, confirm that the following items are present in the figure legend, table legend, main text, or Methods section.

| n/a | Confirmed | |
|---|---|---|
| ☐ | ☒ | The exact sample size (*n*) for each experimental group/condition, given as a discrete number and unit of measurement |
| ☐ | ☒ | A statement on whether measurements were taken from distinct samples or whether the same sample was measured repeatedly |
| ☐ | ☒ | The statistical test(s) used AND whether they are one- or two-sided<br>*Only common tests should be described solely by name; describe more complex techniques in the Methods section.* |
| ☐ | ☒ | A description of all covariates tested |
| ☐ | ☒ | A description of any assumptions or corrections, such as tests of normality and adjustment for multiple comparisons |
| ☐ | ☒ | A full description of the statistical parameters including central tendency (e.g. means) or other basic estimates (e.g. regression coefficient) AND variation (e.g. standard deviation) or associated estimates of uncertainty (e.g. confidence intervals) |
| ☐ | ☒ | For null hypothesis testing, the test statistic (e.g. *F*, *t*, *r*) with confidence intervals, effect sizes, degrees of freedom and *P* value noted<br>*Give P values as exact values whenever suitable.* |
| ☒ | ☐ | For Bayesian analysis, information on the choice of priors and Markov chain Monte Carlo settings |
| ☐ | ☒ | For hierarchical and complex designs, identification of the appropriate level for tests and full reporting of outcomes |
| ☐ | ☒ | Estimates of effect sizes (e.g. Cohen's *d*, Pearson's *r*), indicating how they were calculated |

*Our web collection on statistics for biologists contains articles on many of the points above.*

## Software and code

Policy information about availability of computer code

| Data collection | Sony Cell Sorter Software v2.1.2-5, Biomek Software 5.1 (library preparation), Illumina HiSeq2500, HiSeq4000, and NovaSeq 6000 instrument control software (sequencing) |
|---|---|
| Data analysis | bwa (v.0.7.17), HOMER(v4.11), BEDTools (v2.25.0), MACS2 (v2.1.2), GNU parallel (20220822),<br>GNU R (v4.3.1), ggplot2(3.4.3), stringr (1.5.0), purrr(1.0.2), dplyr (1.1.3), Seurat v5,<br>Python (v3.10), SnapATAC2 (v2.4), Sklearn(v1.1.0), Cicero (v3.16), CellOracle (v0.15.0),<br>Sony SH800S software,<br>https://github.com/beyondpie/CEMBA_wmb_snATAC |

For manuscripts utilizing custom algorithms or software that are central to the research but not yet described in published literature, software must be made available to editors and reviewers. We strongly encourage code deposition in a community repository (e.g. GitHub). See the Nature Portfolio guidelines for submitting code & software for further information.

## Data

Policy information about availability of data

All manuscripts must include a data availability statement. This statement should provide the following information, where applicable:
- Accession codes, unique identifiers, or web links for publicly available datasets
- A description of any restrictions on data availability
- For clinical datasets or third party data, please ensure that the statement adheres to our policy

Demultiplexed FASTQ files are available at the NEMO archive (NEMO, RRID: SCR_016152 ) at https://assets.nemoarchive.org/dat-bej4ymm  (the raw directory under the source data URL in this archive), and at the NCBI under GEO accession number GSE246791 . Processed data are available at our web portal (http://www.catlas.org) and the same GEO accession number above.

## Research involving human participants, their data, or biological material

Policy information about studies with human participants or human data. See also policy information about sex, gender (identity/presentation), and sexual orientation and race, ethnicity and racism.

| Reporting on sex and gender | N/A |
|---|---|
| Reporting on race, ethnicity, or other socially relevant groupings | N/A |
| Population characteristics | N/A |
| Recruitment | N/A |
| Ethics oversight | N/A |

Note that full information on the approval of the study protocol must also be provided in the manuscript.

# Field-specific reporting

Please select the one below that is the best fit for your research. If you are not sure, read the appropriate sections before making your selection.

☒ Life sciences          ☐ Behavioural & social sciences          ☐ Ecological, evolutionary & environmental sciences

For a reference copy of the document with all sections, see nature.com/documents/nr-reporting-summary-flat.pdf

# Life sciences study design

All studies must disclose on these points even when the disclosure is negative.

| Sample size | No statistical methods were used to predetermine sample size. For each of 117 regions from the mouse brain, dissected brain tissues were pooled from 2-31 (only 2 dissections from the mouse cerebellum region had 2 animals for snATAC-seq library construction, all the other samples had 4-31 animals) of the same sex to obtain enough nuclei for single nucleus ATAC-seq for each biological replica, and two biological replicas were performed. In total, 234 samples were included. |
|---|---|
| Data exclusions | No samples were excluded.<br>For analysis, only nuclei with >1,000 fragments / nucleus and transcription start site enrichment > 10 were selected. |
| Replication | Experiments were performed for 2 biological replicates for each of 117 dissection regions. All the replicates were successfully collected. |
| Randomization | There was no randomization of the samples. For each dissection region, dissected brain tissues were pooled from 2-31 (only 2 dissections from the mouse cerebellum region had 2 animals for snATAC-seq library construction, all the other samples had 4-31 animals) of the same sex. The 117 dissection regions were designed before experiments for analyzing the whole mouse brain in a comprehensive way. |
| Blinding | Investigators were not blinded to the specimen being investigated based on our experimental design above. |

# Reporting for specific materials, systems and methods

We require information from authors about some types of materials, experimental systems and methods used in many studies. Here, indicate whether each material, system or method listed is relevant to your study. If you are not sure if a list item applies to your research, read the appropriate section before selecting a response.

## Materials & experimental systems

| n/a | Involved in the study |
|-----|----------------------|
| ☒ | ☐ Antibodies |
| ☒ | ☐ Eukaryotic cell lines |
| ☒ | ☐ Palaeontology and archaeology |
| ☐ | ☒ Animals and other organisms |
| ☒ | ☐ Clinical data |
| ☒ | ☐ Dual use research of concern |
| ☒ | ☐ Plants |

## Methods

| n/a | Involved in the study |
|-----|----------------------|
| ☒ | ☐ ChIP-seq |
| ☐ | ☒ Flow cytometry |
| ☒ | ☐ MRI-based neuroimaging |

# Animals and other research organisms

Policy information about studies involving animals; ARRIVE guidelines recommended for reporting animal research, and Sex and Gender in Research

| | |
|---|---|
| Laboratory animals | Adult (P56) C57BL/6J male mice were purchased from Jackson Laboratories at seven weeks of age and maintained in the Salk animal barrier facility under a 12-h light/12-h dark cycle in a temperature-controlled room with ad libitum access to water and food until euthanasia. The temperature in the animal facility was maintained within the range of 20 to 22.2C, while the humidity levels varied between 35 and 60%. |
| Wild animals | No wild animals were used in this study. |
| Reporting on sex | Only male mice were used. |
| Field-collected samples | No filed-collected samples were used in this study. |
| Ethics oversight | All experimental procedures using live animals were approved by the SALK Institute Animal Care and Use Committee under protocol number 18-00006. |

Note that full information on the approval of the study protocol must also be provided in the manuscript.

# Plants

| | |
|---|---|
| Seed stocks | N/A |
| Novel plant genotypes | N/A |
| Authentication | N/A |

# Flow Cytometry

## Plots

Confirm that:

☒ The axis labels state the marker and fluorochrome used (e.g. CD4-FITC).

☒ The axis scales are clearly visible. Include numbers along axes only for bottom left plot of group (a 'group' is an analysis of identical markers).

☒ All plots are contour plots with outliers or pseudocolor plots.

☒ A numerical value for number of cells or percentage (with statistics) is provided.

## Methodology

| | |
|---|---|
| Sample preparation | Nuclei were stained with DRAQ7 (#7406, Cell Signaling) |
| Instrument | Sony SH800 |
| Software | Sony SH800S software |

| Cell population abundance | Cell populations within each sample were determined using snATAC-seq as described in the manuscript. See Methods and Supplementary table 2 and 3 for details. |
| Gating strategy | Potential nuclei were first identified using FSC-Area and BSC-Area. Next doublets were removed based on BSC and FSC signal width. DRAQQ7 postive nuclei with 2n count were sorted. |

☒ Tick this box to confirm that a figure exemplifying the gating strategy is provided in the Supplementary Information.

