## [Peer Review File · Nature]

Manuscript Title: Single-cell analysis of chromatin accessibility in adult mouse brain

Reviewer Comments & Author Rebuttals

Reviewer Reports on the Initial Version:

Referees' comments:

Referee #1:

Remarks to the Author:

Overview

In this study, Zu, Li, and colleagues collected adult mouse brains and profiled 2.3 million nuclei from 117 anatomical dissections, resulting in identification of 1.3 million candidate cis-regulatory DNA elements (cCREs) from 602 distinct brain cell clusters. They performed several comprehensive analyses, including integration of sc-ATAC-seq data with sc-RNA-seq data and sc-Methyl-seq data (from other studies) from corresponding regions of mouse brain, identifying of the mouse-specific CREs and conserved CREs, inferring cell-type-specific gene regulatory networks, and using deep learning models to predict the activities of gene regulatory elements.

Overall, it's an impressive body of work. The data seem remarkable, the analyses generally well done, and the paper is generally very clearly well written. We have the following suggestions, offered in a constructive spirit to improve the manuscript.

Major comments

1. Since the structure of the 2D UMAP in Fig. 1e is very complex, it's challenging to judge whether the integration works or not by eye. We would suggest performing some quantitative analysis to evaluate the alignment. Moreover, it would be helpful to provide a supplementary figure or table, showing which dissection regions were collected as well as the fractions of cells were recaptured from each region, for each of three studies, even if you are relying on external datasets, just so that they can be compared in the context of a single table (snATAC-seq, scRNA-seq, and snmC-seq).
2. Most of the following analyses were based on 215 subclasses rather than 602 L3-level clusters (e.g. in the gene regulatory networks section). Does it mean that grouping cells into the subclasses is more informative than L3-level clusters? I imagine some analysis showing heterogeneity of a subclass with different L3-level clusters would be helpful. In addition, any comments on the three L3-level clusters which are not mapped to any subclasses defined in the RNA-seq data? More details on the choices and limitations here would be helpful.
3. Out of the new 663,167 cCREs, how many CREs are specific to the cell clusters corresponding to the newly collected major regions in this study?
4. In Fig. 2, panels c-e showed clear difference between overlapped cCREs and non-overlapped cCREs, however, panel f shows all the cCREs (top) vs. non-overlapped cCREs (bottom). It's confusing why authors present it in a different way and what the reader is supposed to learn from panel f.
5. In Fig. 3a-b, the co-accessible cCREs-gene connections were identified from individual cell subclasses using Cicero. However, the correlated cCRE-gene connections were identified across ~200 subclasses, by which I imagine some cell-type-specific connections are not detected. Could the authors comment on it? In addition, it would be better to show one or two real examples of

correlated cCRE-gene connections across subclasses by scatter plots.

6. The number of positively correlated pairs of cCREs and genes ($n = 337K$) is much more than the number of negatively correlated pairs ($n = 100K$), could the author briefly comment or benchmark the finding? Is this expected? Is an even stronger bias towards positive pairs expected?

7. Line 326-336, the authors indicated several interesting examples of TFs with motifs which are enriched in each cCRE module (the right panel in Fig. 3c). Could they also discuss some target genes as well (the middle panel in Fig. 3c)?

8. The authors applied two strategies to identify the cell-type-specific TF-gene pairs, the first strategy is performing correlation analysis between CRE and genes across all the ~ 200 subclasses followed by calling TF motifs on cell-type-specific CRE modules (Fig. 3c), and the second strategy is performing linear regression between TFs and genes (identified by co-accessible pairs) across individual cells in the sc-RNA-seq data to build a gene regulatory network (Fig. 4a). The two strategies seem a little redundant to me (or at least a little confusing), could the authors comment on whether these approaches are complementary or redundant?

9. The conclusion that the mouse-specific cCREs are enriched for transposable elements needs further consideration. I think there are some background signals which need to be normalized. In other words, I suppose the sequences of intron/intergenic are more conserved than the sequences of TE. Then, the current finding could be due to TE itself being less conserved rather than the CREs in TE being less conserved.

10. In Fig. 6c, could the author comment on the reason why the PCC doesn't change for those GABA & GLUT subclasses after using weighted poisson loss function?

11. I am curious, if the authors trained the DL model only using mouse-specific CREs or only using conserved CREs, what would the predictive performance on human sequence be?

12. A very general comment: the current dataset is very specific to the brain, so it would be very interesting to show some brain-specific signals when comparing to non-brain data, e.g. how many of the discovered cCREs are brain-specific?

13. The author showed that 599 clusters defined in sciATAC-seq mapped to 215 subclasses in scRNA-seq. Have the authors looked more closely at the differences between the sciATAC-clusters that mapped to the same scRNA-seq subclasses to see if there are interesting chromatin states in these subclasses?

14. Fig. 4d: I have a hard time understanding the figure. What do the bar and line means in the figure?

15. Line 363-372: The author compared frequency of different modes of GRNs in cell subclasses and found enrichment of some GRN modes in specific cell subclasses. What are some biological implications related to such enrichment? Have these differences been reported in a different study?

16. You state a certain percentage around line 415 for which orthologous sequence cannot be found, but it's not super clear if this is higher or lower than random expectation (and if higher or lower, how much?). If you choose random sequences, what % can you find orthologous human sequence for by the same procedures? It is well established that there is enormous turnover between mammalian genomes since the mouse-human split, and my recollection of the mouse genome paper from 2002 is that 45% is pretty close to what you would expect by chance (i.e. no enrichment or depletion) but you should test this using your procedures systematically. It's still interesting, but important to calibrate the observation.

17. On a related point, with the recent release of the Cactus alignments as a publication (though they have been available for some time), it seems there is a missed opportunity here to perhaps go a little deeper on the cCREs that are in sequences for which there is no orthologous human sequence that can be found. Are these sequences that are being lost in the human lineage and preserved in mouse lineage, or are they new (due to TE activity) in mouse? I don't want to be too prescriptive here, but given that the alignments are all set up, some basic analysis of what is going on here evolutionarily (perhaps connected to response to point 16) would be very helpful, particularly as this is one of the more unique parts of the paper, analytically speaking.

18. Going a little deeper, I would have loved more detail (and sorry if I missed it) re: what TE classes (i.e. specifically what subfamilies of what type of TE) these cCREs seem greatly enriched in. For example, taking the highTE GLUT subclasses, is this a specific subtype of TE that you can say expanded in the mouse lineage (and when?). Is the ancestral sequence of that TE subfamily "pre-enriched" for the motifs you cite around line 435?

19. It's a little bit weird how in both the HOMER and GRN analyses (around lines 318-384), but also in the DL example of oligodendrocyte), the examples you discuss in the text are primarily non-neuronal cell types. Given how focused the overall paper and "value add" of going deeper relates to neuronal sub-classes, could you try to either balance this, or if anything have most of your examples that get discussed in the text derive from neuronal subclasses?

20. I don't find it all that surprising that only 2% of elements overlap w/ promoters, nor that promoter grammar appears different (closing observation of discussion). For the latter, as you increase the number of distal elements, the number of promoters stays fixed, so with more discovery this is exactly what you'd expect. For the former point, because promoters are much less specific than enhancers (all other things being equal) one expects a different grammar and I haven't seen anyone claiming otherwise. I think there are stronger points that you could close on.

21. Please review the entire paper to avoid the pitfall of false precision. Don't say 24.37% of the mouse genome. Just say 24% or perhaps 24.4%. 2-3 sig digits (I prefer 2 but can live with 3) is the most you should have unless the additional precision is meaningful. Similarly, don't say ~36.7%. No tildas before percentages, only when you are rounding whole numbers. The tilda is implied...

Minor comments

- The three terms "sciATAC-seq" and "single cell ATAC-seq" and "snATAC-seq" have been used at different places of the manuscript. It would be better to just use one consistently.

- Line 148 - 150, what type of features (i.e. bin, imputed gene, or peak) were used to perform cell embedding and clustering? It needs to be briefly mentioned here.

- A little confused by panel b in Fig.1 - a) the size scale (i.e. #cell) is not clear enough to present the difference; b) the two legends of "#dissection" and "#total cell" are confusing; c) some colors of major regions are hard to distinguish (e.g. MY and STR, AMY and CTX); d) are the colors of major regions matched to the panel (a)?

- In the last column of Fig. 1d, I don't think the term "Class" has been formally introduced in the main text.

- In Fig. 1e, it's necessary to tell which groups of cells from either dataset were included in the integration analysis in the legend.

- In Fig. 1f, if I understand it correctly, the red color scale of consensus score was manually shifted

to right (more close to 1) to avoid a “noisy” heatmap. I think that’s okay, but still want to see how it looks with a regular color scale.

- I suggest putting the Extended Data Fig. 10e in the main figure (i.e. replacing Fig. 3b)? The plot shows both positively and negatively correlated pairs, which is more informative.

- Line 421, do the 134,765 TE-overlapping cCREs only include mouse-specific CREs?

- In Fig. 5f, the shapes of different TE super family are not distinguishable. In addition, it needs to explain the meaning of labels in the red rectangle (e.g. L1MB8~Cdkl5) by the figure legend.

- Line 434 - 438, I guess the citations of Fig. 5 should be f - h rather than e - g, right?

- The meaning of Fig. 6e is not very clear, would the authors give more explanation?

- Fig. 1: The color schemes of 1a,c and 1b,c are different. It could confuse the reader of which brain region they are looking at.

- Line 188-206: For L3 clustering, the authors performed integration and scRNA-seq and snmC-seq. They selected the 3 closest sub-classes in each integration as candidates. It is unclear to me how they define the ‘3 closest sub-classes’. Further, the steps they took to finalize the annotation from the candidate subclasses with marker genes are not described in detail.

- Line 207-211: The author first claimed that 511 cell subtypes map to both modalities and 59 subtypes map to only scRNA-seq, which will be 570 cell subtypes in total. Then the author claimed that “a total of 599 of 602 L3-level clusters were mapped to the subclasses defined in the whole mouse brain study using the scRNA-seq”. I am a little confused of the inconsistent number of cell clusters that mapped to scRNA-seq.

- Line 255-257: The authors claimed that they have identified additional cCREs that are “enriched for active chromatin states or potential insulator protein binding sites mapped in bulk mouse brain tissues”. Which dataset(s) did they compare to?

Referee #2:

Remarks to the Author:

In this manuscript, Zu, Li. et al generated an atlas of cell-types specific chromatin accessibility at single-cell resolution across 117 regions of the mouse brain expanding on their previous study Li et al. 2021. They identified candidate cis-regulating DNA elements across 602 brain cell populations and integrated single-cell RNA-seq (scRNA-seq) and single-cell DNA methylation from matched regions of the mouse brain reported in companion manuscripts. The integrative analysis of scATAC-seq and scRNA-seq also identified cell subclass-specific Gene regulatory elements. In addition, they have utilized deep learning models to predict cell type-specific cis-regulatory elements using DNA sequences as input. Intriguingly, the authors also note an enrichment of cis-regulatory elements on transposable elements, suggesting a potential mechanism for functional somatic neuronal diversity. Overall, this study provides a valuable resource to the field. However, the manuscript can be improved by addressing the following comments below.

Major comments:

1. The manuscript, while expanding on the depth and regional diversity, bears similarity to the group's previous study from 2021, “An atlas of gene regulatory elements in adult mouse cerebrum”. Besides the increased cell number, this work also includes parts of the midbrain and hindbrain. The authors should emphasize key differences in cCREs and gene-regulatory networks

associated with the newly added regions compared to the previous study.

2. The 663,167 new cCREs identified in this study: are they mainly derived from cell types enriched in the midbrain and hindbrain regions? Or is this increase more a function of the augmented cell number across all brain regions?
3. For each cluster, the signal was aggregated to identify cCREs. Can the authors also calculate the cell-cell variability of the cCREs for each cluster and investigate the difference between highly variable vs highly conserved cCREs?
4. In Figure 2f, please comment on cell subclasses that are hypomethylated across all cCREs that appear as lines on the heatmap.
5. The authors could leverage previously published single-cell Hi-C (and in-silico bulk) (PMID: 33484631) or bulk Hi-C from sorted neurons from publicly available data to check the interaction frequency between cCREs and target genes. This would enhance the confidence in the cCREs calls. For instance, the aggregate Hi-C interaction frequency between cCRE-putative target gene pairs can be compared to random pairs of equidistant locations to test if cCRE-gene pairs have increased proximity.
6. The group previously described Frequently Interacting Regions (FIREs). Do cCREs overlap with FIREs?

Minor comments:

1. In Extended Data Figure 2, a legend delineating datasets from the previous study versus datasets generated in the current study would be beneficial. Are the two large clusters seen in Extended Figure 2f an artifact from two different cohorts or is it biological?
2. In Lines 139-144: Please provide the number of single cells already reported in the previous study vs the additional cells added in the current study.
3. In Figure 2, panel g is not labeled.
4. There seems to be an error in Line 390. The cited reference (19) does not appear to report human ATAC-seq data. Please rectify this.

Author Rebuttals to Initial Comments:

We are grateful to both reviewers for their positive remarks and constructive comments on our previous submission! Their thoughtful comments and suggestions have indeed been tremendously helpful. In the revision, we have added several new dimensions to the study. The major changes are briefly highlighted below, followed by point-by-point responses to the reviewers' comments. The reviewer comments are pasted below followed by our responses in blue font. We also use the red font to indicate new texts added to the revised manuscript.

Summary of major changes:

1. We have redone the clustering analysis using the new SnapATAC2 pipeline (Zhang et al. in press, Nature Method¹), which resulted in identification of 1482 clusters, up from previous 602 clusters. The new clustering results showed much improved alignment with that described in Yao et al. brain cell atlas companion paper.
2. With the new clustering, we repeated the analysis of chromatin accessibility in each cell cluster using a more conservative threshold than the previous submission, finding a combined total of ~1 million peaks. This number is smaller than the previous submission, but in our opinion is more robust as indicated by a higher degree of sequence conservation.
3. We also repeated all the analyses with the new list of cCREs and revised all the figures.
4. We also repeated deep learning to train a sequence-based predictor of chromatin accessibility in the new cell sub-classes. The resulting deep learning models performed equivalently to the previous submission.
5. All these changes are highlighted in the revised manuscript.

Referee #1 (Remarks to the Author):

Overview

In this study, Zu, Li, and colleagues collected adult mouse brains and profiled 2.3 million nuclei from 117 anatomical dissections, resulting in identification of 1.3 million

candidate cis-regulatory DNA elements (cCREs) from 602 distinct brain cell clusters. They performed several comprehensive analyses, including integration of sc-ATAC-seq data with sc-RNA-seq data and sc-Methyl-seq data (from other studies) from corresponding regions of mouse brain, identifying of the mouse-specific CREs and conserved CREs, inferring cell-type-specific gene regulatory networks, and using deep learning models to predict the activities of gene regulatory elements.

Overall, it's an impressive body of work. The data seem remarkable, the analyses generally well done, and the paper is generally very clearly well written. We have the following suggestions, offered in a constructive spirit to improve the manuscript.

Response: Thank you for your enthusiasm about our work. We appreciate your careful review and insightful comments that have greatly helped us improve the manuscript.

Major comments

1. Since the structure of the 2D UMAP in Fig. 1e is very complex, it's challenging to judge whether the integration works or not by eye. We would suggest performing some quantitative analysis to evaluate the alignment. Moreover, it would be helpful to provide a supplementary figure or table, showing which dissection regions were collected as well as the fractions of cells were recaptured from each region, for each of three studies, even if you are relying on external datasets, just so that they can be compared in the context of a single table (snATAC-seq, scRNA-seq, and snmC-seq).

Response: Thank you for the comments! During revision, we carefully re-analyzed our data with our latest pipeline SnapATAC²¹, and obtained over 1480 clusters. We then updated the entire integration analysis with the reference scRNA-seq data², which in their latest annotation contains over 5,300 clusters. The 5,300 clusters are organized into 338 subclasses in the scRNA-seq data. During our integration, after mapping to the clusters based on the transfer-label scores (**Response Fig.1c,d**), we then linked our clusters to the subclasses in Yao et al. (2023). Here the transfer-label score is defined, on a given pair of one snATAC-seq cluster and one

cluster/subclass in their data, as the ratio of cells in the snATAC-seq data mapped to the corresponding cluster/subclass using Seurat CCA TransferLabel method.

Response Fig.1: Heatmaps of transfer-label scores on different levels (cluster and subclass) with different cell groups (neuronal and non-neuronal cells). Each panel is from the figure in our manuscript, and we labeled where they come from. **a**, transfer-label scores between 1267 L4-level clusters from our snATAC-seq data and the mapped 253 out of 315 subclasses in the scRNA-seq data. **b**, transfer-label scores between 215 non-neuronal clusters from our snATAC-seq data and 22 out of 23 mapped subclasses in the scRNA-seq data. **c,d** on the cluster level of the scRNA-seq data respectively.

We also updated the text related with the integration analysis in our manuscript as below (line 180-227).

“To annotate the cell type identity of the 1482 subtypes, we performed integration analysis with the data reported in a companion single-cell RNA-seq study of 2 million cells (over 5,300 clusters) from adult male mouse brains⁵. We first calculated gene expression scores in each nucleus using SnapATAC2 with the fragments mapped to the gene promoter (up to 2kb to transcription starting sites, TSS) and gene body

regions as described before^{31,32}. Then we performed integration analysis using the software Seurat^{32,33} separately for neuronal cells and non-neuronal cells (see **Methods**). The co-embedding of both the scRNA-seq and the snATAC-seq neuronal cells showed excellent overlap between the two modalities (**Fig. 1d**) and the mouse brain major regions (**Extended Data Fig. 6a,b**). We also observed the same result for non-neuronal cells (**Extended Data Fig. 6c,d,e**). The consensus matrix calculated based on the ratio of transferred labels from the scRNA-seq data to our snATAC-seq data showed excellent correspondence between the two datasets, suggesting the robustness of the cell type identification based on either transcriptome or chromatin accessibility (**Fig. 1e, Extended Data Fig. 6f,g,h, Supplementary Table 5**). For each snATAC-seq based subtype, we used the top ranked cluster label transferred from the scRNA-seq data to represent its scRNA-seq cluster-level annotation. In total, 1,267 neuronal subtypes in snATAC-seq data were mapped to 965 scRNA-seq clusters. In the scRNA-seq data, the 5,300 clusters were grouped into 338 cell subclasses, the most representative layer for cell type analysis. In order to annotate our data more robustly, we then mapped our cell subtypes into this layer using hierarchical relationship between cell cluster and cell subclass defined in the scRNA-seq data. The heatmap of the consensus matrix between our subtypes and the scRNA-seq subclasses showed excellent correspondence (**Fig. 1e, Supplementary Table 5**). In order to reduce the potential annotation bias induced by different numbers of cells in the clusters, for each of our 1482 subtypes, we manually checked the major regions of the top 3 cluster-related subclasses, and the gene markers for some subclasses using the bigwig data and gene expression scores (**Extended Data Fig. 7**) generated by SnapATAC2. Finally, 275 out of 338 subclasses were annotated to our 1482 subtypes. This includes 253 out of 315 neuronal subclasses, covering 28 neuronal classes and 7 neurotransmitter types, as well as 22 out of 23 non-neuronal subclasses, covering 5 non-neuronal classes (**Supplementary Table 5**). We confirmed that the matched subclasses in our snATAC-seq data were robust to variations of sequencing depth, signal-to-noise between brain regions and replicates (**Extended Data Fig. 4b,c**) by performing the K-nearest neighbor batch effect test (kBET)³⁴ and local inverse Simpson's index (LISI) analysis³⁵ (**Extended Data Fig. 4d,e**) and by comparing the ratio of biological replicates across multiple subclasses (**Extended Data Fig. 5**). The unmatched 63 subclasses correspond to mainly rare cell populations, accounting for a total of 1.7% of the scRNA-seq data. For example, the

only unmatched non-neuronal subclass is monocytes with 21 cells. Other unmatched subclasses correspond to rare cell subclasses mainly from MB, Pons, and MY regions, where the subtle differences between cell types may hinder their identification using chromatin accessibility profiles alone⁵. Nevertheless, the general agreement between the open chromatin-based clustering and transcriptomics-based clustering laid the foundation for integrative analysis of cell-type specific gene regulatory programs in the mouse brain, as for the mouse cerebral region¹⁹. In the following text, we focused on the snATAC-seq subclasses and the subtypes within each subclass based on the above integrative analysis.”

The transfer-label scores are provided in our **Supplementary Table 5**.

2. Most of the following analyses were based on 215 subclasses rather than 602 L3-level clusters (e.g. in the gene regulatory networks section). Does it mean that grouping cells into the subclasses is more informative than L3-level clusters? I imagine some analysis showing heterogeneity of a subclass with different L3-level clusters would be helpful. In addition, any comments on the three L3-level clusters which are not mapped to any subclasses defined in the RNA-seq data? More details on the choices and limitations here would be helpful.

Response: We primarily performed our bioinformatic analysis at the subclass level, where cell clusters are sufficiently and robustly resolved. At this level, cells within each subclass exhibit a noteworthy consistency in open chromatin landscapes, reflecting similarities in gene activity that align with the subclasses defined in the reference scRNA-seq dataset. This approach strikes a good balance, providing both generalizable insights into gene regulatory elements across various cell types and a comprehensive characterization of cell types and subtypes, all without overwhelming the essence of our analysis with an excessive number of clusters. In the meanwhile, the scRNA-seq papers^{2,3} and the snmC-seq paper⁴ also mainly discussed the biological analysis on the subclass level, so that we can easily compare the same subclass across multiple modalities, such as **Fig. 2f**.

L3-level or L4-level (added during revision) clustering, on the other hand, can provide further details on the cellular heterogeneity of subclasses, revealing unique open chromatin features that define distinguishing transcriptional features of cellular subtypes and transcriptional states.

Below text has been added to the revised manuscript to clarify. (line 292-299)

“We primarily performed our bioinformatic analysis at the subclass level for the later analysis, where cell clusters are sufficiently resolved. At this level, cells within each subclass exhibit a noteworthy consistency in open chromatin landscapes, reflecting similarities in gene activity that align with the subclasses defined in the reference scRNA-seq dataset. This approach strikes a good balance, providing both generalizable insights into gene regulatory elements across various cell types and a comprehensive characterization of cell types and subtypes, all without overwhelming the essence of our analysis with an excessive number of clusters.”

3. Out of the new 663,167 cCREs, how many CREs are specific to the cell clusters corresponding to the newly collected major regions in this study?

Response: Thank you for bringing up this important question. In our revision, we have identified a total of 446,606 new cCREs. To provide a more comprehensive perspective on the specificity of these cCREs to the cell types associated with the newly collected regions, we established specific criteria. A region-specific cell subclass was defined when over 60% of the cells in that subclass belonged to one major region. Subsequently, we conducted an analysis to determine how many of the newly discovered cCREs were specifically present in these region-specific subclasses.

Here is the distribution of specific cCREs in our analysis:

6,920 cCREs are specific to Amygdala-related subclasses.

4,279 cCREs are specific to Cerebellum-related subclasses.

27,253 cCREs are specific to Diencephalon-related subclasses.

27,698 cCREs are specific to Hindbrain-related subclasses.

40,383 cCREs are specific to Midbrain-related subclasses.

In addition, we identified:

86,347 cCREs that are specific to Telencephalon (excluding Amygdala) -related subclasses.

253,726 cCREs that are not specific to any region-related subclasses. These new Telencephalon-specific or non-specific cCREs could result from an increased cell number in the previous Telencephalon region or broader regions, illustrating the complexity of the dataset.

4. In Fig. 2, panels c-e showed clear difference between overlapped cCREs and non-overlapped cCREs, however, panel f shows all the cCREs (top) vs. non-overlapped cCREs (bottom). It's confusing why authors present it in a different way and what the reader is supposed to learn from panel f.

Response: We showed in the upper panel heatmap all the cCREs to demonstrate highly cell-type-specific patterns of these cCREs across all the cell subclasses defined in the study, which are further supported by cell-type-specific DNA hypomethylation patterns revealed by an independent technology (snmC-seq). Based on this result, it is expected that overlapping cCREs would show a similar level of consistency (which is indeed supported by our analysis). However, if there are any inconsistencies in non-overlapping cCREs, this could potentially be masked by the overlapping cCREs. Thus, we separated them from the whole set of cCREs and performed the same analysis (lower panel), which showed similar cell-type-specific patterns, further supporting a potential role of these non-overlapping cCREs in gene regulation.

5. In Fig. 3a-b, the co-accessible cCREs-gene connections were identified from individual cell subclasses using Cicero. However, the correlated cCRE-gene connections were identified across ~200 subclasses, by which I imagine some cell-type-specific connections are not detected. Could the authors comment on it? In addition, it would be better to show one or two real examples of correlated cCRE-gene connections across subclasses by scatter plots.

Response: Thank you for the comments! Indeed, the overall logic in **Fig. 3** is to identify the potential enhancers by observing the pseudo-bulk level correlations. We explored the single-cell level identification of cell-type-specific connections in **Fig. 4**. We add the comments in our manuscript to avoid potential confusions. (line 353-359)

“The analysis above was to relate distal cCREs with their putative target genes in a pseudo-bulk level and the correlations were calculated along all the subclasses. Next, we explored the inference of gene regulatory network⁵⁰ (GRN) in a single-cell level analysis. A GRN is typically represented as a static graph, where the nodes are either TFs or non-TF genes, and the edges are weighted and directed from TFs to their putative target genes. Negative edges suggest potential repressors, and positive ones potential activators.”

Thanks for the suggestions of using scatter plots to show several examples. In our paper, we use **Fig.3c** to show the correlations between the potential enhancers and their putative target gene in each row.

6. The number of positively correlated pairs of cCREs and genes (n = 337K) is much more than the number of negatively correlated pairs (n = 100K), could the author briefly comment or benchmark the finding? Is this expected? Is an even stronger bias towards positive pairs expected?

Response: That is correct. This is expected as open chromatin regions enrich positive cis-regulatory elements (enhancers) that enhance gene expression.

7. Line 326-336, the authors indicated several interesting examples of TFs with motifs which are enriched in each cCRE module (the right panel in Fig. 3c). Could they also discuss some target genes as well (the middle panel in Fig. 3c)?

Response: Thanks for your suggestion! In **Fig.3**, the motif analysis is basically on each peak module (in total we have 54 modules). Based on your comments, we put more discussion of the TFs and the corresponding target genes in the gene regulatory networks in **Fig.4** since this would be more interesting to discreetly look at the subclass-level specific regulations. We attached the corresponding modifications made to the manuscript regarding the TF analysis below. (line 416-429)

“TFs such as Jun, Junb, Fos have high importance scores across multiple neuronal and non-neuronal subclasses. TFs of the bHLH family such as Neurod1, Neurod2, Neurod6, Bhlhe22 have high importance scores for many types of neurons such as

the glutamatergic neurons in the isocortex region. Our analysis also indicated potential regulations of gene expressions in GABAergic neurons by TFs such as Arx, Sp8 and Sp9 in the telencephalon regions; whereas TFs such as Gata2, Tal1 and Gata3 showed high importance scores for GABAergic neurons in the MB and pons regions. Tcf7l2, Shox2, and Ebf1 had high importance scores associated with glutamatergic neurons specifically in the TH region. Additionally, the Tcf7l2 gene exhibited high importance in the MB region. Next we observed TFs Foxa1, Foxa2 pointed to specific association to the glutamatergic neurons in the MB region. Hox family TFs displayed high importance scores in both GABAergic and glutamatergic neurons in the MY region. Last, Maf and Mafb showed high importance scores in GABAergic neurons in the cortex region.”

Fig.4 has also been revised as shown below.

“Fig. 4: Inference of subclass-specific gene regulatory networks (GRNs) across the whole mouse brain. a, Example of the GRN inferred in telencephalon-region astrocyte (ASC-TE NN) using CellOracle⁵¹. Edges are weighted and directed to reflect the putative regulation strength and mode (inhibition or activation). **b,** Graph degree distribution of the GRN in **a**. $P(k)$, the probability of a node having k degree in the GRN. The degree of one node is the number of other nodes having links with it. **c.** Boxplots of the number of transcription factors (TFs), the number of genes, the number of regulated TFs per gene, and the number of genes regulated by the TFs among the GRNs for each cell subclass. **d,** Normalized histograms of the number of the regulated double positive⁵⁵ network motifs for each main cell class. The lines are the kernel-based density curves fitted for different histograms. **e,** Histograms of the two network

motifs for five mouse brain regions. Telencephalon: Isocortex, olfactory bulb (OLF), hippocampal formation (HPF), striatum (STR), pallium (PAL) and amygdala (AMY); Diencephalon: thalamus (TH) and hypothalamus (HY); Midbrain: midbrain (MB); Hindbrain: medulla (MY) and pons; Cerebellum: cerebellum (CB). **f**, Heatmap of eigenvector-based centralities or importance scores of TFs in each of the subclass-specific GRNs. Each row represents a TF, and each column a subclass. Orders of the TFs and subclasses are based on the Yao et al, 2023⁵ for the similar heatmap but using the scRNA-seq data.”

8. The authors applied two strategies to identify the cell-type-specific TF-gene pairs, the first strategy is performing correlation analysis between CRE and genes across all the ~200 subclasses followed by calling TF motifs on cell-type-specific CRE modules (Fig. 3c), and the second strategy is performing linear regression between TFs and genes (identified by co-accessible pairs) across individual cells in the scRNA-seq data to build a gene regulatory network (Fig. 4a). The two strategies seem a little redundant to me (or at least a little confusing), could the authors comment on whether these approaches are complementary or redundant?

Response: We apologize for the confusion that this might have caused. These two strategies are complementary to each other: the cCRE co-accessibility analysis using Cicero followed by motif analysis provides the gene regulatory programs in a pseudo-bulk analysis view, whereas the linear regression analysis using CellOracle highlights the regulatory relationships between TF motif occurrence and gene expression, providing gene regulatory networks in a TF-centered and single-cell analysis view. To avoid any potential confusion, we revised our text to highlight the unique biological insights that can be obtained from these complementary analyses. (line 353-359)

“The analysis above was to relate distal cCREs with their putative target genes in a pseudo-bulk level and the correlations were calculated along all the subclasses. Next, we explored the inference of gene regulatory network⁵⁰ (GRN) in a single-cell level analysis. A GRN is typically represented as a static graph, where the nodes are either TFs or non-TF genes, and the edges are weighted and directed from TFs to their putative target genes. Negative edges suggest potential repressors, and positive ones potential activators.”

9. The conclusion that the mouse-specific cCREs are enriched for transposable elements needs further consideration. I think there are some background signals which need to be normalized. In other words, I suppose the sequences of intron/intergenic are more conserved than the sequences of TE. Then, the current finding could be due to TE itself being less conserved rather than the CREs in TE being less conserved.

Response: Following your suggestion, we conducted an analysis of sequence conservation in introns, intergenic regions, and transposable elements (TEs) across the mouse and human genomes. Our findings are as follows: mouse-specific cCREs exhibit a significantly higher percentage of TEs; orthologous cCREs exhibit a significantly higher percentage of introns and intergenic regions. These results align with the idea that sequences within introns and intergenic regions tend to be more conserved than TE sequences. We have integrated these new findings into

Response Fig. 2 and in the revised the manuscript at line 640-643: “Comparing TE-overlapping cCREs to cCREs in introns and intergenic regions, we observed that cCREs in introns and intergenic regions are more conserved in the human genome. This suggests that TE elements themselves are less conserved, rather than the cCREs within TE elements being less conserved.”

Response Fig. 2: Barplots showing the fraction of genomic distribution of full set, mouse-specific, and orthologous cCREs. The mouse-specific cCREs show a significant enrichment for TE. The orthologous cCREs show a significant enrichment for introns and intergenic regions. Enrichment testing was conducted utilizing the hypergeometric distribution. *** denotes $FDR < 0.0001$.

10. In Fig. 6c, could the author comment on the reason why the PCC doesn't change for those GABA & GLUT subclasses after using weighted poisson loss function?

Response: There are modest improvements in accuracy for the neuronal subclasses based on our experiments. The nature of the loss function is it increases the “importance” of cell types which are more “independent” when training the model. In this case it means non-neuronal cell types have a larger contribution to the loss than neuronal cell types. Given the loss should predominantly encourage the model to improve accuracy in non-neuronal cell types, the gains among neuronal cell types suggest our model is overall more reflective of gene regulatory grammar.

11. I am curious, if the authors trained the DL model only using mouse-specific CREs or only using conserved CREs, what would the predictive performance on human sequence be?

Response: We thank the reviewer for their thoughtful question. While we agree this is biologically, and methodologically interesting, it is somewhat inappropriate for our model, which integrates information from across all sequences in a 131072 base pair region. Because almost all promoters have conserved sequences between the two species, we feel comfortable speculating signals which rely on promoter proximal sequence will be predicted much less accurately, but we cannot assume the impact on distal elements.

12. A very general comment: the current dataset is very specific to the brain, so it would be very interesting to show some brain-specific signals when comparing to non-brain data, e.g. how many of the discovered cCREs are brain-specific?

Response: In our revision, we identified that 56% of the cCREs in the current study overlapped with representative DNase hypersensitivity sites (rDHSs) in the SCREEN database (<https://screen.encodeproject.org/>). As you suggested, we conducted further analysis to assess the activity of the cCREs, as defined by this study, in other non-brain rDHS. To distinguish non-brain rDHS, we collected rDHSs from both mouse brain tissue and cerebellum tissue in the SCREEN database. Any rDHSs not associated with brain or cerebellum tissue were classified as non-brain rDHS.

Comparing our cCREs with the non-brain rDHS, we observed that 48% (508609/1053811) of the cCREs identified in the current study lack epigenomic features related to non-brain rDHS, indicating their brain-specific nature.

13. The author showed that 599 clusters defined in sciATAC-seq mapped to 215 subclasses in scRNA-seq. Have the authors looked more closely at the differences between the sciATAC-clusters that mapped to the same scRNA-seq subclasses to see if there are interesting chromatin states in these subclasses?

Response: Thank you for the comments! During revision, we reanalyze our integration analysis, and based on the results, all the clusters in our data are assigned to a subclass in the scRNA-seq data. We addressed this quantitative measurement in the first question.

14. Fig. 4d: I have a hard time understanding the figure. What do the bar and line means in the figure?

Response: Thank you for pointing this out. We apologize for the confusion. This is a histogram of the counts of the motifs in different groups. In order to compare different groups, we choose a normalized histogram in order to compare the results between groups. The lines are the fitting curve automatically generated based on the histogram in order to show the smoothed density based on the histogram. Based on your comments, we revised our figure legend for **Fig.4d**:

“d, Normalized histograms of the number of the regulated double positive⁵⁵ network motifs for each main cell class. The lines are the kernel-based density curves fitted for different histograms.”

15. Line 363-372: The author compared frequency of different modes of GRNs in cell subclasses and found enrichment of some GRN modes in specific cell subclasses. What are some biological implications related to such enrichment? Have these differences been reported in a different study?

Response: Thank you so much for pointing out this. We are not sure if there are some biological implications for such enrichment. But we do see some non-neuronal cells tend to have high motif enrichments. We did not see the reports on this before.

16. You state a certain percentage around line 415 for which orthologous sequence cannot be found, but it's not super clear if this is higher or lower than random expectation (and if higher or lower, how much?). If you choose random sequences, what % can you find orthologous human sequences for by the same procedures? It is well established that there is enormous turnover between mammalian genomes since the mouse-human split, and my recollection of the mouse genome paper from 2002 is that 45% is pretty close to what you would expect by chance (i.e. no enrichment or depletion) but you should test this using your procedures systematically. It's still interesting, but important to calibrate the observation.

Response: Thank you for your valuable suggestion. As a control, we employed bedtools (v2.30.0) to randomly shuffle cCREs with default parameters and conducted the same analysis. Our findings from the randomly shuffled cCREs are as follows: 32% (458,110 out of 1,427,178) of randomly shuffled cCREs were identified as orthologous. In contrast, 58% (613,073 out of 1,053,811) of real cCREs were classified as orthologous. This observed percentage of orthologous cCREs is significantly higher than random expectations, suggesting that cCREs may indeed exhibit greater conservation across species compared to other sequences. We have incorporated this information into the manuscript with the following statements:

“The percentage of orthologous cCREs is significantly higher than the random expectation (32% orthologous for randomly shuffled cCREs).” (line 439-441) and “We investigated the sequence conservation of gene regulatory elements in the whole mouse brain by performing a liftover to the human genome. In comparison to a random background, cCREs in the mouse brain exhibit greater conservation in the human genome.” (line 610-613).

17. On a related point, with the recent release of the Cactus alignments as a publication (though they have been available for some time), it seems there is a missed opportunity here to perhaps go a little deeper on the cCREs that are in sequences for which there is no orthologous human sequence that can be found.

Are these sequences that are being lost in the human lineage and preserved in mouse lineage, or are they new (due to TE activity) in mouse? I don't want to be too prescriptive here, but given that the alignments are all set up, some basic analysis of what is going on here evolutionarily (perhaps connected to response to point 16) would be very helpful, particularly as this is one of the more unique parts of the paper, analytically speaking.

Response: While we did not conduct a full-scale re-alignment to other species, we employed an efficient approach known as 'liftover' to map the coordinates of cCREs between assemblies using a corresponding mapping file. This approach, compared to full re-alignment methods, offers a quicker and more cost-effective solution⁵. In order to ascertain if these sequences were new to the mouse genome, we utilized the liftover approach to compare mouse-specific regions with the genomes of chicken (galGal6) and *X. tropicalis* (xenTro10). Our analysis revealed that merely 0.44% (1941/440,738) and 0.31% (1353/440,738) of these regions were orthologous in chicken and *X. tropicalis*, respectively.

18. Going a little deeper, I would have loved more detail (and sorry if I missed it) re: what TE classes (i.e. specifically what subfamilies of what type of TE) these cCREs seem greatly enriched in. For example, taking the highTE GLUT subclasses, is this a specific subtype of TE that you can say expanded in the mouse lineage (and when?). Is the ancestral sequence of that TE subfamily "pre-enriched" for the motifs you cite around line 435?

Response: We extended our investigation to assess the superfamilies and families of the differential chromatin accessibility (DCA) TE-cCREs within highTE GLUT. This was done in comparison to all TE-cCREs within highTE GLUT as the background. The results indicated a noteworthy enrichment of DCA TE-cCREs in the LINE superfamily (FDR = 8.05e-36) and the L1 subfamily (FDR = 1.27e-38). L1, a retrotransposon that is active in both mouse and human genomes, has accumulated in mammalian genomes over time. This makes L1 a valuable source of evolutionary novelties through the provision of essential motifs⁶. These findings have been incorporated into the manuscript (lines 484-489) for further clarification.

“Furthermore, we delved into the superfamilies and families of the DCA TE-cCREs in highTE GLUT, comparing them to all TE-cCREs in highTE GLUT as the background. We observed a significant enrichment of DCA TE-cCREs in the LINE superfamily (FDR = 8.05e-36) and the L1 subfamily (FDR = 1.27e-38). L1, an actively retrotransposon in both mouse and human, has accumulated in mammalian genomes. It can serve as a source of evolutionary novelties by providing essential motifs⁶³.”

19. It's a little bit weird how in both the HOMER and GRN analyses (around lines 318-384), but also in the DL example of oligodendrocyte), the examples you discuss in the text are primarily non-neuronal cell types. Given how focused the overall paper and “value add” of going deeper relates to neuronal sub-classes, could you try to either balance this, or if anything have most of your examples that get discussed in the text derive from neuronal subclasses?

Response: Thank you for the comment! Based on your comments, we add more discussions about neuronal cell related TFs in our **Fig. 4**. Here are the corresponding contents (line 405-429):

“Furthermore, we highlighted the importance of key transcription factors within these networks by calculating their eigenvector centrality scores using CellOracle. In **Fig.4f**, the 267 subclasses and 226 transcription factors were ordered in the same way as in Yao et al., 2023⁵ (**Supplementary Table 21**). Interestingly, we observed a similar pattern of importance scores for the transcription factors as seen in the scRNA-seq data, where normalized gene expression were shown. This consistency of the transcription factor signatures across modalities reinforced the fidelity of our gene regulatory network inferences. It also demonstrated how regulatory codes of transcription factors across the whole mouse brain could be revealed through integrated analysis of snATAC-seq and scRNA-seq data.

TFs such as Jun, Junb, Fos have high importance scores across multiple neuronal and non-neuronal subclasses. TFs of the bHLH family such as Neurod1, Neurod2, Neurod6, Bhlhe22 have high importance scores for many types of neurons such as

the glutamatergic neurons in the isocortex region. Our analysis also indicated potential regulations of gene expressions in GABAergic neurons by TFs such as Arx, Sp8 and Sp9 in the telencephalon regions; whereas TFs such as Gata2, Tal1 and Gata3 showed high importance scores for GABAergic neurons in the MB and pons regions. Tcf7l2, Shox2, and Ebf1 had high importance scores associated with glutamatergic neurons specifically in the TH region. Additionally, the Tcf7l2 gene exhibited high importance in the MB region. Next we observed TFs Foxa1, Foxa2 pointed to specific association to the glutamatergic neurons in the MB region. Hox family TFs displayed high importance scores in both GABAergic and glutamatergic neurons in the MY region. Last, Maf and Mafb showed high importance scores in GABAergic neurons in the cortex region.”

20. I don't find it all that surprising that only 2% of elements overlap w/ promoters, nor that promoter grammar appears different (closing observation of discussion). For the latter, as you increase the number of distal elements, the number of promoters stays fixed, so with more discovery this is exactly what you'd expect. For the former point, because promoters are much less specific than enhancers (all other things being equal) one expects a different grammar and I haven't seen anyone claiming otherwise. I think there are stronger points that you could close on.

Response: Thank you for your insightful comment. We have trained a new model. With this new model there is not a significant difference between cCRE accuracy, and promoter accuracy. Additionally, we showed in **Fig. 6e** that the model has improved performance across cell types where there are greater differences. Promoters also have less cell-type-specific activity than enhancers. In the face of this, it is difficult to suggest that a difference in model performance implies a difference in grammar between enhancers and promoters. We have removed the corresponding section from the text.

21. Please review the entire paper to avoid the pitfall of false precision. Don't say 24.37% of the mouse genome. Just say 24% or perhaps 24.4%. 2-3 sig digits (I prefer 2 but can live with 3) is the most you should have unless the additional

precision is meaningful. Similarly, don't say ~36.7%. No tildas before percentages, only when you are rounding whole numbers. The tilda is implied...

Response: Thank you for pointing this out. We have now revised our manuscript accordingly when describing percentages.

Minor comments

- The three terms “sciATAC-seq” and “single cell ATAC-seq” and “snATAC-seq” have been used at different places of the manuscript. It would be better to just use one consistently.

Response: Thank you for pointing this out. We have now consolidated the term into “snATAC-seq” throughout the revised manuscript.

- Line 148 - 150, what type of features (i.e. bin, imputed gene, or peak) were used to perform cell embedding and clustering? It needs to be briefly mentioned here.

Response: Bins were used as features to perform cell embedding and clustering. We revised our text to highlight this (line: 151-178), “We performed iterative clustering using SnapATAC²⁹ to classify the 2.3 million nuclei into distinct cell groups based on their pairwise similarity of chromatin accessibility profiles (**Extended Data Fig. 4, Extended Data Fig. 5, Supplementary Table 3, see Methods**). We performed four rounds of iterative clustering in order to further classify the cells into subclasses and cell subtypes (**Extended Data Fig. 4a**). During clustering, we used 500-base pair (bp) resolution for genomic bin features. Following the first iteration (hereafter referred to as L1-level clustering), we divided all the 2.3 million nuclei into 37 groups for L2-level clustering, using the over 4 million chromatin features. For each group, we then performed a second and a third round of clustering (L2-level, L3-level clustering) sequentially with the top 500,000 genomic bin features and identified a total of 248 subgroups and 899 subtypes of brain cells correspondingly (**Extended Data Fig. 4a**). 291 out of 899 L3-level subtypes consisted of more than 400 cells per subtype, and in total they captured 1.8 million cells. We finally performed a fourth round of clustering (L4-level clustering) for the

291 L3-level subtypes to further classify them into a total of 874 clusters. In summary, after L4-level clustering, we identified 1482 cell clusters (874 L4-level clusters and 608 L3-level clusters without L4-level clustering). The number of nuclei in each cluster ranges from 34 to 48,694, with a median number of 484 nuclei per cluster (**Supplementary Table 3,4**). We used the term “subtypes” to represent the 1482 clusters in the latter part of this manuscript.”

- A little confused by panel b in Fig.1 - a) the size scale (i.e. #cell) is not clear enough to present the difference; b) the two legends of “#dissection” and “#total cell” are confusing; c) some colors of major regions are hard to distinguish (e.g. MY and STR, AMY and CTX); d) are the colors of major regions matched to the panel (a)?

Response: We further revised our figures and legends to provide more clarity on this. #dissection represents the number of sample directions covered by our previous study (Last) and updated in the current study (New). #total cells represent the number of cells covered by our previous study (Last) and updated in the current study (New). **Fig. 1d** is currently updated to **Fig. 1f**. For major regions, we have revised and standardized the color schemes to ensure consistency. These updated figures should offer greater clarity in depicting the brain regions. However, please note that the colors in **Fig. 1a** differ from our standard major region colors. **Fig. 1a** serves as a schematic illustration of brain tissue dissection, and the distinct colors are used to delineate the boundaries of large brain regions. This schematic is shared with other teams involved in the project, and we intend to retain this color scheme as is.

“Fig. 1: Single-cell analysis of chromatin accessibility in the adult whole mouse brain. a, Schematic of sample dissection strategy. A detailed list of regions in **Supplementary Table 1.** **b,** Number of nuclei for 117 dissections after quality control and doublet removal. The dot size is proper to the size of cells and the gray colors are the dissections that are not covered by our previous study¹⁹. A detailed list of single nuclei in our snATAC-seq data in **Supplementary Table 2.** A to L on the left used as the dissection region labels on each slice (see **Extended Data Fig.1** for details). #dissection represents the number of sample directions covered by our previous study (Last) and updated in the current study (New). #total cells represent the number of cells covered by our previous study (Last) and updated in the current study (New). **c,** Uniform manifold approximation and projection (UMAP)⁸⁰ embedding and clustering analysis of snATAC-seq data. Light colors denote major cell classes. GABA, GABAergic neurons; Glut, glutamatergic neurons; NN, non-neuronal cells. Cells are

colored with major regions as in **a**. **d**, The co-embedding UMAP embedding of the neuronal cells from scRNA-seq data⁵ and the snATAC-seq data on the same space colored by the two modalities. **e**, Consensus score between neuronal subclasses from the scRNA-seq data above and L4-level neuronal clusters from our snATAC-seq data. **f**, The 253 neuronal subclasses in our snATAC-seq data matched to neuronal subclasses in the scRNA-seq above, and ordered based on the subclass ids (all our figures later were kept the same order if not specifically mentioned). From left to right, the bar plots represent class, major neurotransmitter (NT) type, biological replicate distribution of nuclei, major region distribution of nuclei, number of clusters, and number of nuclei. The detailed information about class, NT type, and subclass can be found in Yao, et al., 2023⁵.”

- In the last column of Fig. 1d, I don't think the term “Class” has been formally introduced in the main text.

Response: We revised our text/figure legends accordingly to further clarify this. The previous **Fig. 1d** now is updated to **Fig. 1f**.

“Finally, 275 out of 338 subclasses were annotated to our 1482 subtypes. This includes 253 out of 315 neuronal subclasses, covering 28 neuronal classes and 7 neurotransmitter types, as well as 22 out of 23 non-neuronal subclasses, covering 5 non-neuronal classes (**Supplementary Table 4**).” (line 208-211)

Fig. 1f, “The 253 neuronal subclasses in our snATAC-seq data matched to neuronal subclasses in the scRNA-seq above and ordered based on the subclass ids (all our figures later were kept the same order if not specifically mentioned). From left to right, the bar plots represent class, major neurotransmitter (NT) type, biological replicate distribution of nuclei, major region distribution of nuclei, number of clusters, and number of nuclei. The detailed information about class, NT type, and subclass can be found in Yao, et al., 2023⁵.” (line 677-683)

- In Fig. 1e, it's necessary to tell which groups of cells from either dataset were included in the integration analysis in the legend.

Response: The previous **Fig. 1e** is currently updated **Extended Data Fig. 6a,b**. We revised the text and legend accordingly (line: 1273-1276). **“Integration analysis between the snATAC-seq and the scRNA-seq data for neurons and non-neurons separately. UMAP on the co-embedding space of neurons from the snATAC-seq data (a) and scRNA-seq data (b). Colors as major regions.”** We also provide **Supplementary Table 5** and **4** to give the detail of consensus score and brain regions for integration analysis.

- In Fig. 1f, if I understand it correctly, the red color scale of consensus score was manually shifted to right (more close to 1) to avoid a “noisy” heatmap. I think that’s okay, but still want to see how it looks with a regular color scale.

Response: The previous **Fig. 1f** has now been updated to **Fig. 1e**. This adjustment involved shifting the scale towards scores close to 1 to emphasize the most correlated scATAC-seq/scRNA-seq cluster relationships. This change was made because additional clusters from cell types with similar transcriptome profiles could generate background noise on the heatmap. To provide clarity, we’ve incorporated a score legend bar to illustrate the heatmap’s scale and included **Supplementary Table 5**, which details the consensus scores for each cluster.

- I suggest putting the Extended Data Fig. 10e in the main figure (i.e. replacing Fig. 3b)? The plot shows both positively and negatively correlated pairs, which is more informative.

Response: Thank you for your suggestion. In the revision, we have emphasized the interpretation of positively correlated pairs, which is a key advantage of snATAC-seq technology. However, for comprehensive information, we have included a summary of both positively and negatively correlated pairs in **Supplementary Table 13**.

- Line 421, do the 134,765 TE-overlapping cCREs only include mouse-specific CREs?

Response: These cCREs encompass all cCRE classes, which include both mouse-specific and orthologous cCREs, although the mouse-specific cCREs constitute 80%

of this total. We have revised our text to provide clearer information. The updated sentence reads, “Notably, the genes near the 115,772 TE-overlapping cCREs, including both mouse-specific and orthologous cCREs, and expressed in at least one of the highTE GLUT neuron subclasses were enriched for those involved in synaptic related functions (**Extended Data Fig. 14f-h**.)” (line 466-469).

- In Fig. 5f, the shapes of different TE super family are not distinguishable. In addition, it needs to explain the meaning of labels in the red rectangle (e.g. L1MB8~Cdkl5) by the figure legend.

Response: Thank you for pointing this out. We further revised this figure and its legend to provide more clarity. “The red color labeled top 10 DCA TE-cCREs, which correlated with synaptic related genes. The top10 DCA TE-cCRE and gene pairs (e.g. L1MB8~Cdkl5) were displayed in the left red box. The super family of top10 DCA TE-cCREs were also labeled with different shapes in the left red box.” in line 765-768. Below is the revised figure with legend.

“Fig. 5: Analyses of chromatin accessibility at transposon elements (TEs) of cCREs. f, Volcano plot showing differential chromatin accessibility (DCA) at TE-cCREs in highTE GLUT subclasses compared to other subclasses. The red color labeled top 10 DCA TE-cCREs, which correlated with synaptic related genes. The top10 DCA TE-cCRE and gene pairs (e.g. L1MB8~Cdkl5) were displayed in the left red box. The super family of top10 DCA TE-cCREs were also labeled with different shapes in the left red box.”

- Line 434 - 438, I guess the citations of Fig. 5 should be f - h rather than e - g, right?

Response: That is right. Thank you for pointing this out and we apologize for our oversight during our internal revision process. The figure citations have been corrected in the revised version of the manuscript.

- The meaning of Fig. 6e is not very clear, would the authors give more explanation?

Response: Thank you for your suggestion. We change the text of the legend. We additionally improved the clarity of the plot by changing the style and adding a color bar. The new panel is shown below:

“Fig. 6e, Density scatter plot showing the models ability to predict cell-type specific patterns of open chromatin. The coefficient of variance (Variance/Mean) across cell types is compared to the Pearson r calculated between true signals and predicted signals across cell subclasses. Each dot represents one cCRE in the testing set.”

- Fig. 1: The color schemes of 1a,c and 1b,c are different. It could confuse the reader of which brain region they are looking at.

Response: We have revised and standardized the color schemes for major regions to ensure consistency. These updated figures should offer greater clarity in depicting the brain regions. However, please note that the colors in **Fig. 1a** differs from our standard major region colors. **Fig. 1a** serves as a schematic illustration of brain tissue dissection, and the distinct colors are used to delineate the boundaries of

large brain regions. This schematic is shared with other teams involved in the project, and we intend to retain this color scheme as is.

- Line 188-206: For L3 clustering, the authors performed integration and scRNA-seq and snmC-seq. They selected the 3 closest sub-classes in each integration as candidates. It is unclear to me how they define the '3 closest sub-classes'. Further, the steps they took to finalize the annotation from the candidate subclasses with marker genes are not described in detail.

Response: In the revision, we conducted an integration of L4 clusters. We have included additional details in our revised manuscript (line 203-208) to better explain the methods used in these analyses. In summary, we defined the top three subclasses based on the consensus score between our clusters and the subclasses in the scRNA-seq data. To identify the marker genes of the top three subclasses in our snATAC-seq data, we manually reviewed the bigwig data for each cluster to finalize the annotation.

“In order to reduce the potential annotation bias induced by different numbers of cells in the clusters, for each of our 1482 subtypes, we manually checked the major regions of the top 3 cluster-related subclasses, and the gene markers for some subclasses using the bigwig data and gene expression scores (Extended Data Fig. 7) generated by SnapATAC2.”

- Line 207-211: The author first claimed that 511 cell subtypes map to both modalities and 59 subtypes map to only scRNA-seq, which will be 570 cell subtypes in total. Then the author claimed that “a total of 599 of 602 L3-level clusters were mapped to the subclasses defined in the whole mouse brain study using the scRNA-seq”. I am a little confused of the inconsistent number of cell clusters that mapped to scRNA-seq.

Response: We apologize for any confusion this may have caused. The 511 cell subtypes and 59 subtypes mapped to both scATAC-seq and scRNA-seq are specifically referring to neuronal cell types. The 599 clusters mapped to the subclasses include both neuronal and non-neuronal cell types. In our revision, we conducted integration on our L4 clusters and scRNA clusters. This is described in

line 185-211, and we have provided **Fig. 1e, f**, and **Extended Data Fig. 6** to better illustrate the integration of both neuronal and non-neuronal lineages.

“Then we performed integration analysis using the software Seurat^{32,33} separately for neuronal cells and non-neuronal cells (see **Methods**). The co-embedding of both the scRNA-seq and the snATAC-seq neuronal cells showed excellent overlap between the two modalities (**Fig. 1d**) and the mouse brain major regions (**Extended Data Fig. 6a,b**). We also observed the same result for non-neuronal cells (**Extended Data Fig. 6c,d,e**). The consensus matrix calculated based on the ratio of transferred labels from the scRNA-seq data to our snATAC-seq data showed excellent correspondence between the two datasets, suggesting the robustness of the cell type identification based on either transcriptome or chromatin accessibility (**Fig. 1e**, **Extended Data Fig. 6f,g,h**, **Supplementary Table 5**). For each snATAC-seq based subtype, we used the top ranked cluster label transferred from the scRNA-seq data to represent its scRNA-seq cluster-level annotation. In total, 1,267 neuronal subtypes in snATAC-seq data were mapped to 965 scRNA-seq clusters. In the scRNA-seq data, the 5,300 clusters were grouped into 338 cell subclasses, the most representative layer for cell type analysis. In order to annotate our data more robustly, we then mapped our cell subtypes into this layer using hierarchical relationship between cell cluster and cell subclass defined in the scRNA-seq data. The heatmap of the consensus matrix between our subtypes and the scRNA-seq subclasses showed excellent correspondence (**Fig. 1e**, **Supplementary Table 5**). In order to reduce the potential annotation bias induced by different numbers of cells in the clusters, for each of our 1482 subtypes, we manually checked the major regions of the top 3 cluster-related subclasses, and the gene markers for some subclasses using the bigwig data and gene expression scores (**Extended Data Fig. 7**) generated by SnapATAC2. Finally, 275 out of 338 subclasses were annotated to our 1482 subtypes. This includes 253 out of 315 neuronal subclasses, covering 28 neuronal classes and 7 neurotransmitter types, as well as 22 out of 23 non-neuronal subclasses, covering 5 non-neuronal classes (**Supplementary Table 4**).”

- Line 255-257: The authors claimed that they have identified additional cCREs that are “enriched for active chromatin states or potential insulator protein binding sites mapped in bulk mouse brain tissues”. Which dataset(s) did they compare to?

Response: We apologize for any confusion. The dataset we used is the 15-state ChromHMM model for mouse brain chromatin from a previous study. We have clarified this in line 1314-1315: “Enrichment analysis of the peak sets with a 15-state ChromHMM model in the mouse brain chromatin¹⁰³.”

Referee #2 (Remarks to the Author):

In this manuscript, Zu, Li. et al generated an atlas of cell-types specific chromatin accessibility at single-cell resolution across 117 regions of the mouse brain expanding on their previous study Li et al. 2021. They identified candidate cis-regulating DNA elements across 602 brain cell populations and integrated single-cell RNA-seq (scRNA-seq) and single-cell DNA methylation from matched regions of the mouse brain reported in companion manuscripts. The integrative analysis of scATAC-seq and scRNA-seq also identified cell subclass-specific Gene regulatory elements. In addition, they have utilized deep learning models to predict cell type-specific cis-regulatory elements using DNA sequences as input. Intriguingly, the authors also note an enrichment of cis-regulatory elements on transposable elements, suggesting a potential mechanism for functional somatic neuronal diversity. Overall, this study provides a valuable resource to the field. However, the manuscript can be improved by addressing the following comments below.

Response: Thank you for your recognition of our work and your careful review of our manuscript. We appreciate your insightful comments and helpful suggestions, and have addressed all your comments and concerns below and in the revised manuscript.

Major comments:

1. The manuscript, while expanding on the depth and regional diversity, bears similarity to the group's previous study from 2021, “An atlas of gene regulatory elements in adult mouse cerebrum”. Besides the increased cell number, this work also includes parts of the midbrain and hindbrain. The authors should emphasize key differences in cCREs and gene-regulatory networks associated with the newly added regions compared to the previous study.

Response: Thank you for the comment! Here in our work, we performed gene-regulatory network (GRN) analysis for each subclass from the whole mouse brain, which was not done in our previous study, and during our revision, we added more descriptions about TFs in midbrain and other regions. Below is the updated **Fig. 4f** and attached the discussion about this. The key differences in cCREs with the newly added regions are discussed in the next question.

“Fig. 4f, Heatmap of eigenvector-based centralities or importance scores of TFs in each of the subclass-specific GRNs. Each row represents a TF, and each column a subclass. Orders of the TFs and subclasses are based on the Yao et al, 2023⁵ for the similar heatmap but using the scRNA-seq data.”

Here we add the discussions in our manuscript for **Fig.4f** in line:405-429:

“Furthermore, we highlighted the importance of key transcription factors within these networks by calculating their eigenvector centrality scores using CellOracle. In Fig.4f, the 267 subclasses and 226 transcription factors were ordered in the same way as in Yao et al., 2023⁵ (Supplementary Table 21). Interestingly, we observed a similar pattern of importance scores for the transcription factors as seen in the scRNA-seq data, where normalized gene expressions were shown. This consistency

of the transcription factor signatures across modalities reinforced the fidelity of our gene regulatory network inferences. It also demonstrated how regulatory codes of transcription factors across the whole mouse brain could be revealed through integrated analysis of snATAC-seq and scRNA-seq data.

TFs such as Jun, Junb, Fos have high importance scores across multiple neuronal and non-neuronal subclasses. TFs of the bHLH family such as Neurod1, Neurod2, Neurod6, Bhlhe22 have high importance scores for many types of neurons such as the glutamatergic neurons in the isocortex region. Our analysis also indicated potential regulations of gene expressions in GABAergic neurons by TFs such as Arx, Sp8 and Sp9 in the telencephalon regions, whereas TFs such as Gata2, Tal1 and Gata3 showed high importance scores for GABAergic neurons in the MB and pons regions. Tcf7l2, Shox2, and Ebf1 had high importance scores associated with glutamatergic neurons specifically in the TH region. Additionally, the Tcf7l2 gene exhibited high importance in the MB region. Next we observed TFs Foxa1, Foxa2 pointed to specific association to the glutamatergic neurons in the MB region. Hox family TFs displayed high importance scores in both GABAergic and glutamatergic neurons in the MY region. Last, Maf and Mafb showed high importance scores in GABAergic neurons in the cortex region. “

2. The 663,167 new cCREs identified in this study: are they mainly derived from cell types enriched in the midbrain and hindbrain regions? Or is this increase more a function of the augmented cell number across all brain regions?

Response: Thank you for raising this important question. In our revision, we redid the clustering using the latest SnapATAC2 software, which allowed us to resolve nearly twice as many clusters as in the previous submission. We also used a more stringent peak finding approach, which led to identification of a total of 446,606 new cCREs, fewer than the previous submission. To provide a more comprehensive perspective on the specificity of these cCREs to the cell types associated with the newly collected regions, we established specific criteria. A region-specific cell subclass was defined when over 60% of the cells in that subclass belonged to one

major region. Subsequently, we conducted an analysis to determine how many of the newly discovered cCREs were specifically present in these region-specific subclasses.

Here is the distribution of specific cCREs in our analysis:

6,920 cCREs are specific to Amygdala-related subclasses.

4,279 cCREs are specific to Cerebellum-related subclasses.

27,253 cCREs are specific to Diencephalon-related subclasses.

27,698 cCREs are specific to Hindbrain-related subclasses.

40,383 cCREs are specific to Midbrain-related subclasses.

In addition, we identified:

86,347 cCREs that are specific to Telencephalon (exclude Amygdala) -related subclasses.

253,726 cCREs that are not specific to any region-related subclasses. These new Telencephalon-specific or non-specific cCREs could result from an increased cell number in the previous Telencephalon region or broader regions, illustrating the complexity of the dataset.

3. For each cluster, the signal was aggregated to identify cCREs. Can the authors also calculate the cell-cell variability of the cCREs for each cluster and investigate the difference between highly variable vs highly conserved cCREs?

Response: Thank you for bringing up this important point. As limited by the scATAC-seq technology itself, the single cell open chromatin data is very sparse and suffers limited coverage per cell, making analysis of cell-cell variability of cCREs for each cluster challenging.

However we calculated the median and variation of the accessibility for cCREs across the subclasses (**Extended Data Fig. 9e**) for the cCREs overlapped with ENCODE rDHS (DHS-ovlp cCREs) and the cCREs not overlapped with ENCODE rDHS (non-DHS-ovlp cCREs). The scatter plot on the left showed that DHS-ovlp cCREs has a group of cCREs that had high accessibility on average but small variations across the subclasses. On the other hand, non-DHS-ovlp cCREs showed

high variations across the subclasses with low accessibility on average, which indicates these cCREs are more cell-type specific ones.

If we look further on the peak modules across the subclasses in **Fig.2f**, we can see there are two peak modules at top left that showed consistent high accessibility across the subclasses and they also showed consistent hypomethylation signals across the subclasses. This is also observed in our previous study on cerebral regions⁷, and they mainly correspond to promoters, CTCF and some other regulatory elements.

4. In Figure 2f, please comment on cell subclasses that are hypomethylated across all cCREs that appear as lines on the heatmap.

Response: Thank you for pointing out this. We checked the subclasses that show hypomethylation across all cCREs and highlighted them in the heatmap below (**Response Fig.3**). The right arrows pointed to the subclasses. We checked the information shared by the companion study⁴, and they are the subclasses with the lowest number of cells in their data. MG-POL-SGN Nts Glut (34 cells only) and DMH Prdm13 Gaba (32 cells only). The median size of the subclasses in snmC-seq data is 432. So it is potentially the lower number of sizes in the subclasses that induces the issue.

Response Fig.3 (Fig.2f, and only show the heatmap for all the cCREs)

Upper, heat map showing the chromatin accessibility at 150 cis-regulatory modules across the 244 shared cell subclasses in both snATAC-seq and snmC-seq data (see companion manuscript by Liu et al.⁴) in the mouse brain. Rows represent subclasses, and columns are representative cCREs sampled from each module. Lower, heat map showing the snDNA-methylation signals at the genomic locations of the corresponding cCREs for the same subclasses. The red arrows point to the two subclasses with lowest number of cells in the snmC-seq data.

5. The authors could leverage previously published single-cell Hi-C (and in-silico bulk) (PMID: 33484631) or bulk Hi-C from sorted neurons from publicly available data to check the interaction frequency between cCREs and target genes. This would enhance the confidence in the cCREs calls. For instance, the aggregate Hi-C interaction frequency between cCRE-putative target gene pairs can be compared to random pairs of equidistant locations to test if cCRE-gene pairs have increased proximity.

Response: Thank you for your suggestion! As advised we cross-referenced dataset⁴, where a comprehensive chromatin conformation / methylome joint profile throughout the adult mouse brain is described, and most of the subclass annotations (244 subclasses of 275 subclasses in our data) are shared between these two datasets.

To evaluate the confidence of identified subclass-specific cCRE-genes pairs, we randomly select eleven major subclasses (Sst_Gaba, Pvalb_Gaba, CBX_MLI_Megf11_Gaba, Vip_Gaba, CA1-ProS_Glut, CB_Granule_Glut, L6_CT_CTX_Glut, L2-3_IT_CTX_Glut, Astro-TE_NN, Microglia_NN, Bergmann_NN), and calculate the Hi-C signal enrichment (at 1 kb resolution) at the top 20% subclass-specific cCRE-gene pairs anchors identified in this study. We found that there is statistically significant higher enrichment (p value = 0.004) of chromatin interaction signal at the corresponding subclass-specific cCRE-gene pairs anchors, compared to non-corresponding pairs anchors, suggesting that subclass-specific cCRE-gene pairs are more likely to interact in the cell types where the cCREs are active.

In the meanwhile, we selected the two peak modules that show global accessibility across the subclasses based on the non-negative matrix factorization analysis (**Fig. 2f up left**). Then selected all the proximal-distal connections (pdc) with cCREs in the peak modules above and ranked the pdc based on the highest cicero scores they have. We treated them as global pdc and performed the Hi-C signals by aggregating all the Hi-C data. From the heatmaps (**Response Fig.4b**), we observed the strong enrichment signals for the global pdc.

Response Fig. 4: a. Boxplots of the enrichment scores (1kb resolution) of aggregate peak analysis (APA) for the top 20% positive proximal-distal connections (ppdc) from several represented subclasses. Match, the subclass's Hi-C data used for the same subclasses. Unmatch, the subclass's Hi-C data used for other subclasses as a random background. **b.** Heatmaps of enrichment signals for the top 10% global proximal-distal connections (pdc) and enrichment signals for the random pairs.

6. The group previously described Frequently Interacting Regions (FIREs). Do cCREs overlap with FIREs?

Response: We thank the reviewer for the insightful suggestion and we performed the analysis accordingly. We called FIREs in the mouse cortex⁸ applying the criteria

in our group's FIRE paper⁹. The result showed that the majority of FIREs (3,158 out of 3,169) overlap with cCREs in the mouse brain, and a part of the cCREs (71,626 out of 1,053,811) overlap with FIREs (**Response Fig. 5a**).

Next, we tested whether cCRE are enriched at FIREs through permutation analysis. Briefly, we shuffled the mouse genome 1000 times, each time generating 1,053,811 random regions with equivalent sizes as the cCREs. We then calculate the number of overlap between the randomly-generated regions and the FIREs during each shuffle. We found that cCREs are significantly enriched at FIREs ($p < 0.001$, **Response Fig. 5b**), with the actual number of overlaps on FIREs dramatically higher than expected.

Response Fig. 5: a. The mouse brain cCREs were compiled and merged from all mouse brain cell clusters. The FIREs at adult mouse cortex are 40Kb bins called with a FIRE score higher than 3.00 and lifted over from mm9 to mm10. **b.** Columns on the left denote the overlap resulting from permutation of cCREs. The red line on the right marks the actual number of overlap ($n=71,626$) between cCREs and FIREs. Empirical p value was calculated.

Minor comments:

1. In Extended Data Figure 2, a legend delineating datasets from the previous study versus datasets generated in the current study would be beneficial. Are the two large clusters seen in Extended Figure 2f an artifact from two different cohorts or is it biological?

Response: Thank you for your suggestion. We have included the sample information from the previous study in **Fig. 1b**. Now we also added this information in **Extended Fig. 2f** accordingly. It's important to note that the previous study only contained samples from the cortex region. When you refer to **Extended Fig. 2f**, the samples are primarily clustered based on their major regions, rather than the specific cohorts.

“**Extended Data Fig. 2f**, Heat map showing the pairwise Spearman correlation coefficients of the mapping correlations of the bam files between the snATAC-seq datasets. The column and row names consist of two parts: brain region name and replicate label. Study represents sample directions covered by our previous study (Last) or updated in the current study (New).”

2. In Lines 139-144: Please provide the number of single cells already reported in the previous study vs the additional cells added in the current study.

Response: We provided the number of cells from the previous study for each region in line 140-146 to highlight this information. “Among them, 817,655 were from Isocortex (including 370,841 from previous study), 201,113 from OLF (including 137,209 from previous study), 155,952 from STR (including 114,743 from previous study), 81,834 from PAL (including 38,960 from previous study), 271,933 from HPF (including 164,568 from previous study), 65,958 from AMY, 142,890 from TH, 83,321

from HY, 243,137 from MB, 82,488 from MY, 103,147 from Pons, and 106,414 from CB (Fig. 1a,b, Extended Data Fig. 3e,f).”

3. In Figure 2, panel g is not labeled.

Response: We apologize for our mistake in preparing the figures and now have corrected this in the revised manuscript.

4. There seems to be an error in Line 390. The cited reference (19) does not appear to report human ATAC-seq data. Please rectify this.

Response: We apologize for this mistake and have corrected the citation for this in the revised manuscript, which should be this one: Yang Eric Li, et al., *Science*, 2023 (DOI: 10.1126/science.adf7044).

Reference:

- 1 Zhang, K., Zemke, N. R., Armand, E. J. & Ren, B. SnapATAC2: a fast, scalable and versatile tool for analysis of single-cell omics data. *bioRxiv*, 2023.2009.2011.557221 (2023). <https://doi.org:10.1101/2023.09.11.557221>
- 2 Yao, Z. *et al.* A high-resolution transcriptomic and spatial atlas of cell types in the whole mouse brain. *bioRxiv*, 2023.2003.2006.531121 (2023). <https://doi.org:10.1101/2023.03.06.531121>
- 3 Zhang, M. *et al.* A molecularly defined and spatially resolved cell atlas of the whole mouse brain. *bioRxiv* (2023). <https://doi.org:10.1101/2023.03.06.531348>
- 4 Liu, H. *et al.* Single-cell DNA Methylome and 3D Multi-omic Atlas of the Adult Mouse Brain. *bioRxiv*, 2023.2004. 2016.536509 (2023).
- 5 Luu, P. L., Ong, P. T., Dinh, T. P. & Clark, S. J. Benchmark study comparing liftover tools for genome conversion of epigenome sequencing data. *NAR Genom Bioinform* **2**, lqaa054 (2020). <https://doi.org:10.1093/nargab/lqaa054>
- 6 Sookdeo, A., Hepp, C. M., McClure, M. A. & Boissinot, S. Revisiting the evolution of mouse LINE-1 in the genomic era. *Mob DNA* **4**, 3 (2013). <https://doi.org:10.1186/1759-8753-4-3>
- 7 Li, Y. E. *et al.* An atlas of gene regulatory elements in adult mouse cerebrum. *Nature* **598**, 129-136 (2021). <https://doi.org:10.1038/s41586-021-03604-1>
- 8 Shen, Y. *et al.* A map of the -regulatory sequences in the mouse genome. *Nature* **488**, 116-120 (2012). <https://doi.org:10.1038/nature11243>
- 9 Schmitt, A. D. *et al.* A Compendium of Chromatin Contact Maps Reveals Spatially Active Regions in the Human Genome. *Cell Reports* **17**, 2042-2059 (2016). <https://doi.org:10.1016/j.celrep.2016.10.061>

Reviewer Reports on the First Revision:

Referees' comments:

Referee #1 (Remarks to the Author):

Overall we are pleased with the revision and response. There are few remaining points:

1. We thank the authors for improving the integration analysis with scRNA-seq data and for carefully addressing the first part of our previous comment-1. However, we note that the second part of our comment-1 was fully addressed: "Moreover, it would be helpful to provide a supplementary figure or table showing which dissection regions were collected and the fractions of cells recaptured from each region for each of the three studies, even if you are relying on external datasets." This seems like a straightforward thing to generate and would allow for comparison of the three datasets (snATAC-seq, scRNA-seq, and snmC-seq) in a single table.
2. A typo in the response to our comment-10: In this case it means *non-neuronal* cell types have a larger contribution to the loss than *non-neuronal* cell types.
3. Comment 15 - you did not flesh out the potential biological implication of this particular analysis; more explanation would be helpful.
4. Comment 17 - we still think that it's a missed opportunity (esp. with the Cactus alignments out there) here to not perhaps go a little deeper on the cCREs that are in sequences for which there is no orthologous human sequence that can be found, but not the end of the world.

Referee #2 (Remarks to the Author):

The authors have addressed all my comments. However, for points 2 and 4 it's not clear what was changed in the manuscript so the readers don't have the same concern. In addition, the authors performed new analyses in response to points 5 and 6, but these figures are not incorporated into the manuscript.

Author Rebuttals to First Revision:

We are grateful again to both reviewers for their positive remarks and constructive comments on our previous submission! The reviewer comments are pasted below followed by our responses in blue font. We also use the red font to indicate new texts added to the revised manuscript.

Referee #1 (Remarks to the Author):

Overall we are pleased with the revision and response.

Remaining points:

1. We thank the authors for improving the integration analysis with scRNA-seq data and for carefully addressing the first part of our previous comment-1. However, we note that the second part of our comment-1 was fully addressed: "Moreover, it would be helpful to provide a supplementary figure or table showing which dissection regions were collected and the fractions of cells recaptured from each region for each of the three studies, even if you are relying on external datasets." This seems like a straightforward thing to generate and would allow for comparison of the three datasets (snATAC-seq, scRNA-seq, and snmC-seq) in a single table.

Response: We added the major region information for both the scRNA-seq and our snATAC-seq data in the **Supplementary Table 4**. For snmC-seq data, we did not use this information for our integration now, and the companion study¹ for the snmC-seq data provides the related information in their manuscript.

2. A typo in the response to our comment-10: In this case it means *non-neuronal* cell types have a larger contribution to the loss than *non-neuronal* cell types.

Response: We updated the commit 10 response here with the changed sentence labels as red color.

"There are modest improvements in accuracy for the neuronal subclasses based on our experiments. The nature of the loss function is it increases the "importance" of cell types which are more "independent" when training the model. **In this case it means non-neuronal cell types have a larger contribution to the loss than neuronal cell types.** Given the loss should predominantly encourage the model to improve

accuracy in non-neuronal cell types, the gains among neuronal cell types suggest our model is overall more reflective of gene regulatory grammar. ”

3. Comment 15 - you did not flesh out the potential biological implication of this particular analysis; more explanation would be helpful.

Response: We showcased some potential biological implication of the gene regulatory network analysis using the example of border-associated macrophages, where we have identified a regulatory module composed of Atf3, Klf4 and Zbtb16 that is supported by an independent study (lines 370-384). In terms of the observation that “non-neuronal cells showed higher numbers on several network motifs compared to glutamatergic neurons and GABAergic neurons”, we believe this is likely due to a representation bias of certain cell clusters. Based on the current cell type annotation, we have 22 non-neuronal subclasses that constitute almost half of the scATAC-seq profiles. On the contrary, more than 250 neuronal subclasses can be identified from the other half of the entire dataset. This representation bias of cellular composition could underlie the difference in the degree of motif enrichment. Although we tried to address this to our best capacity through a down-sampling approach to balance the subclass sizes when performing Cicero and CellOracle analyses, it can still affect the results when a query cell cluster has much less number of cells than the down-sampled non-neuronal clusters (see **Methods** for details).

4. Comment 17 - we still think that it's a missed opportunity (esp. with the Cactus alignments out there) here to not perhaps go a little deeper on the cCREs that are in sequences for which there is no orthologous human sequence that can be found, but not the end of the world.

Response: Thank you for mentioning the tool Cactus. We admit that it could be a good opportunity for future analysis. Unfortunately, due to time and space limitations, we were unable to incorporate such analysis into the current manuscript as it extends beyond the scope of the current work.

Referee #2 (Remarks to the Author):

The authors have addressed all my comments. However, for points 2 and 4 it's not clear what was changed in the manuscript so the readers don't have the same concern. In addition, the authors performed new analyses in response to points 5 and 6, but these figures are not incorporated into the manuscript.

Response: Thank you again for your recognition of our work and your careful review of our manuscript! We appreciate your insightful comments and helpful suggestions!

According to the suggestions, we added the following information to our manuscript:

- Point 2: due to the space limitation, we are unfortunately not able to include this analysis to the current manuscript. However, we are glad to **open peer review contents in public**, so readers still can have this information.
- Point 4: “Two subclasses were hypomethylated across most of the cCREs potentially due to the low number of cells in the companion study (**Extended Data Fig. 9j**)”
- Point 5: “Majority of the frequently interacting regions from the mouse cortex region² (3,158 out of 3,169) overlap with our cCREs (**Extended Data Fig. 9e,f, see Methods**). ”
- Point 6: “The top proximal-distal cCRE pairs and positive pairs showed enrichment signals using the chromatin conformation data from the companion study¹ (**Extended Data Fig. 10g,h, see Methods**). ”

REFERENCES

- 1 Liu, H. *et al.* Single-cell DNA Methylome and 3D Multi-omic Atlas of the Adult Mouse Brain. *bioRxiv*, 2023.2004.2016.536509 (2023).
- 2 Shen, Y. *et al.* A map of the regulatory sequences in the mouse genome. *Nature* **488**, 116-120 (2012).
<https://doi.org:10.1038/nature11243>